# Action Chunking and Exploratory Data Collection Yield Exponential Improvements in Behavior Cloning for Continuous Control

**Thomas T. Zhang**[*] **Daniel Pfrommer**[†] **Chaoyi Pan**[‡] **Nikolai Matni**[*] **Max Simchowitz**[‡]

[*]University of Pennsylvania  [†]Massachusetts Institute of Technology  [‡]Carnegie Mellon University

## Abstract

This paper presents a theoretical analysis of two of the most impactful interventions in modern learning from demonstration in robotics and continuous control: the practice of *action-chunking* (predicting sequences of actions in open-loop) and *exploratory augmentation* of expert demonstrations. Though recent results show that learning from demonstration, also known as imitation learning (IL), can suffer errors that compound *exponentially* with task horizon in continuous settings, we demonstrate that action chunking and exploratory data collection circumvent exponential compounding errors in different regimes. Our results identify control-theoretic stability as the key mechanism underlying the benefits of these interventions. On the empirical side, we validate our predictions and the role of control-theoretic stability through experimentation on popular robot learning benchmarks. On the theoretical side, we demonstrate that the control-theoretic lens provides fine-grained insights into how compounding error arises, leading to tighter statistical guarantees on imitation learning error when these interventions are applied than previous techniques based on information-theoretic considerations alone.

## 1 Introduction

Imitation learning (IL) is the problem of learning complex behaviors from data labeled with actions from an expert demonstrator policy. This methodology encompasses both some of the earliest examples and most recent state-of-the-art in control for autonomous robotic systems (Pomerleau, 1988; Ross & Bagnell, 2010; Bojarski et al., 2016; Teng et al., 2023; Zhao et al., 2023). Following the rise of large language models (LLMs), IL has also become increasingly prevalent in settings where an agent predicts *discrete tokens*, such as words in a sentence, lines in a proof, or positions on a chessboard (Chen et al., 2021). Such methods have also seen adoption in the context of both continuous and discretized-action control of continuous state-space dynamical systems in an autoregressive fashion.

The recent and dramatic successes of imitation learning in continuous control applications has coincided with a range of algorithmic interventions which appear essential to ensure strong performance: 1. the prediction of open-loop sequences, or "chunks" of actions by the control policy, called *action-chunking* (AC), 2. the careful curation of expert data to be imitated and 3. the adoption of *generative* neural architectures (e.g. conditional diffusion models (Chi et al., 2023)) as parameterizations of learned policies. While the benefits of 3. have been studied broadly, a precise understanding of how action-chunking and curated expert data improve behavior cloning performance remains elusive. Current hypotheses around AC foreground partial observability as the underlying mechanism, despite its clear benefits in fully-observable, state-based control (see e.g. Figure 2). Moreover, studies on active data collection, recent and classical, focus on multi-round interactive data collection to witness expert corrections, but do not isolate core mechanisms of how exploratory data can cover susceptibilities of behavior cloning, especially for single or few-shot dataset generation.

In this work, we provide the first theoretical guarantees justifying the practices of AC and exploratory data augmentation during expert data collection (defined formally below) in the **minimal** setting of imitation of an expert in a state-based continuous-control problem. Our point of departure is the finding in recent work (Simchowitz et al., 2025) that imitation learning in continuous settings— even those whose dynamics and expert demonstrator appear benign—can be considerably more

---

[*]{ttz2, nmatni}@seas.upenn.edu

[†]dpfrom@mit.edu

[‡]{chaoyip, msimchow}@andrew.cmu.edu

**Figure 1:** Visualization of the benefits of action-chunking (Practice 1) and noise-injection (Practice 2). **Left:** even on synthetic *globally EISS* (Definition 2.1) dynamics $f$, frequent feedback can cause exponential compounding error, which action-chunking mitigates. **Center:** HalfCheetah-v5 environment. We see sufficiently large white noise injection yields significant performance improvement, on par with more advanced iterative methods. **Right:** Humanoid-v5 environment. Iterative methods like DAGGER and DART can be suboptimal due to poor learned policy rollouts or aggressive noise-covariance shaping, while naive noise-injection reliably provides the necessary *local* exploration; error bars omitted for clarity. Experiment details in §F and Appendix G.

challenging than imitation in discrete settings, such as those encountered in language modeling, demonstrating compounding errors can grow exponentially with horizon, as opposed to polynomially (or none) (Foster et al., 2024). As Simchowitz et al. (2025) eliminates the possibility of a simple "fix" to the *learning procedure*, we instead consider how changes to either 1. the policy parameterization or 2. the data-collection process can circumvent this negative result. We thereby elucidate both the design space of "sound" offline learning methodologies and better understand the success of widely-deployed practices such as action-chunking (Zhao et al., 2023) and data-augmentation (Ross et al., 2011; Laskey et al., 2017; Ke et al., 2021; Hu et al., 2025).

**Contributions.** We provide the first theoretical guarantees in continuous state-action IL for **interventions that provably prevent compounding error without iterative expert feedback.** Whereas previous work (Ross et al., 2011; Laskey et al., 2017; Pfrommer et al., 2022) require either iterative interaction with the expert or knowledge of the underlying system, we establish our results without access to such oracles, using near-"vanilla" behavior cloning. We study two key practices:
**Practice 1: Action-Chunking.** When the environment is benign, we show the algorithmic modification of *action-chunking*, i.e., predicting and playing open-loop sequences of actions, mitigates compounding errors without requiring any modification to the expert data (Theorem 1).
**Practice 2: Exploratory Data Collection via Expert Noise-Injection.** When the environment is less benign, some alteration of the expert data distribution is necessary. We demonstrate *noise-injection*, i.e., adding noise while executing expert actions, is a simple and practical tool for avoiding compounding errors (Theorem 2).

**Surprising Takeaways.** While Practice 1 and 2 are reflective of popular practices at the intersection of (reinforcement) learning and control, our analysis additionally uncovers phenomena that contrast with the common perspectives of both literatures. In particular:
**Rationale for action-chunking.** Action-chunking has been motivated by both enabling larger policy latency, and encouraging long-horizon planning and stronger policy representations (Chi et al., 2023; Liu et al., 2025). We illuminate an orthogonal rationale: **action-chunking encourages control-theoretic stability of policies learned**, mitigating possibly exponential compounding errors from unstable closed-loop interactions.

Moreover, our analysis of expert-noise injection reveals that, from a theoretical perspective, adaptive iterative interaction or queries with an expert is not needed.
**Sufficiency of Non-Interactive Exploration.** Various prior works have approached the compounding errors problem by adaptively querying the expert policy at possibly suboptimal states (Ross et al., 2011; Laskey et al., 2017), with the overarching goal of *witnessing how the expert policy recovers from errors*. We expose the key directions of (additional) supervision around expert trajectories required to achieve stable imitation, and **we attain this supervision through noising expert actions at a fixed, isotropic scale**. In doing so, we establish that—unlike what an online learning perspective may suggest—**iterative, adaptive interaction with an expert is unnecessary to achieve near-optimal expert imitation**.

Finally, our results reveal the inadequacy of existing theoretical tools for describing or mitigating compounding errors in continuous state spaces.
**Towards New Notions of "Coverage" in Continuous State Space.** The Simchowitz et al. (2025) lower bounds shows that standard notions of "coverage" in theoretical reinforcement learning (Jiang

& Xie, 2024) (which is *maximal* when learning from expert data) is insufficient for mitigating compounding error in continuous behavior cloning. Our work reveals that that **novel, stronger notions of coverage realized via noise injection do suffice in continuous state spaces,** and lead to guarantees that are sharper than coverage-based arguments which leverage exploratory data.

**Towards New Notions of "Excitation" in Continuous State Space.** Despite requiring stronger coverage, our bounds do not require the injected noise to produce "persistent excitation" from control theory (Bai & Sastry, 1985), i.e. uniform variation across all state directions. Rather *naive exploration* (Simchowitz & Foster, 2020) via white noise suffices, even when the underlying system is not controllable (Kailath, 1980). This is because the **error-directions which are susceptible to compounding error are those along which noise injection (Practice 2) provides supervision.**

**Related Work.** Imitation learning from expert demonstrations has emerged as a dominant technique for learning performant models across many sequential decision-making applications. As such, the compounding error phenomenon is well-documented, dating back even to the introduction of IL (Pomerleau, 1988). In discrete state-action settings, compounding errors appears more benign (Ross & Bagnell, 2010; Ross et al., 2011), where recent work by Foster et al. (2024) demonstrates that just modifying the loss may result in performance that has *no* adverse dependence on horizon. However, these settings are ill-suited for continuous control, where the expert policy must be estimated in information-theoretic distances that are not feasible, e.g., even for deterministic policies. A complementary line of work has attempted to understand the theoretical foundations of imitating in continuous settings. Tu et al. (2022) parameterize a scale of "incremental stability" (see Definition 2.1) and study its impact on the statistical generalization of IL. Pfrommer et al. (2022) proposes sufficient conditions for benign compounding errors in a similar setting. However, the resulting algorithms have exceedingly strong requirements, e.g., stability oracles or $\partial$input/$\partial$state derivative sketching, respectively. Rounding off this line of work, Simchowitz et al. (2025) offers definitive evidence that exponential compounding errors cannot be avoided by altering the learning procedure, motivating the interventions we study. We restate the relevant lower bounds in Theorem A. In addition to the works discussed above, we provide extended background in Appendix B.

## 2 PRELIMINARIES

We consider a discrete-time, continuous state-action control system with states $\mathbf{x}_t \in \mathcal{X} = \mathbb{R}^{d_x}$ and inputs[1] $\mathbf{u}_t \in \mathcal{U} = \mathbb{R}^{d_u}$, where dynamics deterministically evolve according to $\mathbf{x}_{t+1} = f(\mathbf{x}_t, \mathbf{u}_t)$. A deterministic policy $\pi$ maps histories of states, inputs, and the current time step to a control input $\mathbf{u}_t = \pi(\mathbf{x}_{1:t}, \mathbf{u}_{1:t-1}, t)$. We assume the initial state is drawn $\mathbf{x}_1 \sim D$ for some distribution $D$ fixed throughout. We say $\pi$ is Markovian and time-invariant if we can simply express $\mathbf{u}_t = \pi(\mathbf{x}_t)$. In this case, we define the closed-loop dynamics $f^\pi(\mathbf{x}, \mathbf{u}) \triangleq f(\mathbf{x}, \pi(\mathbf{x}) + \mathbf{u})$, and $f^\pi(\mathbf{x}) = f^\pi(\mathbf{x}, 0)$. We let $\mathbb{E}_\pi$ (resp. $\mathbb{P}_\pi$) denote expectation (resp. law) under $\mathbf{x}_1 \sim D$, the dynamics $f$, and inputs selected by the policy $\pi$. Given two deterministic policies $\pi_1, \pi_2$, we let $\mathbb{E}_{\pi_1, \pi_2}$ denote the expectation of sequences $(\mathbf{x}_t^{\pi_i}, \mathbf{u}_t^{\pi_i})_{t \geq 1}$ under the dynamics $f$, coupled so that $\mathbf{x}_1^{\pi_1} = \mathbf{x}_1^{\pi_2} \sim D$. We consider estimation of **deterministic, Markovian** expert policies $\pi^\star : \mathcal{X} \to \mathcal{U}$ given a problem horizon $T$. Our aim is to learn some policy $\hat{\pi}$ which accumulates low squared-**trajectory error**:

$$\mathbf{J}_{\text{TRAJ},T}(\hat{\pi}) \triangleq \mathbb{E}_{\hat{\pi}, \pi^\star}\left[ \sum_{t=1}^{T} \min\left\{ 1, \|\mathbf{x}_t^{\hat{\pi}} - \mathbf{x}_t^{\pi^\star}\|^2 + \|\mathbf{u}_t^{\hat{\pi}} - \mathbf{u}_t^{\pi^\star}\|^2 \right\} \right]. \tag{2.1}$$

Above, the practice of taking a minimum with 1 accounts for the possibility of unbounded trajectories on rare events; 1 can be replaced by an arbitrary constant. Upper bounds on $\mathbf{J}_{\text{TRAJ},T}$ imply upper bounds on the difference in expected Lipschitz costs, see e.g., Appendix D. Let $\mathbb{P}_{\text{demo}}$ denote a probability distribution over demonstrations $(\mathbf{x}_t, \mathbf{u}_t)_{1 \leq t \leq T}$. We set $\mathbb{P}_{\text{demo}} = \mathbb{P}_{\pi^\star}$ in Practice 1, but consider modifications beyond the expert distribution for Practice 2. For a given candidate imitator policy $\hat{\pi}$, we may define the population-level risk:

$$\mathbf{J}_{\text{DEMO},T}(\hat{\pi}; \mathbb{P}_{\text{demo}}) \triangleq \mathbb{E}_{\mathbb{P}_{\text{demo}}}\left[ \sum_{t=1}^{T} \|\hat{\pi}(\mathbf{x}_{1:t}, \mathbf{u}_{1:t-1}, t) - \mathbf{u}_t\|^2 \right] \tag{2.2}$$

We broadly term $\mathbf{J}_{\text{DEMO},T}$ the **on-expert error**, i.e. the generalization error of $\hat{\pi}$ imitating over $\mathbb{P}_{\text{demo}}$. This can be interchanged with an error scaling law from standard supervised learning, e.g. given $n$ independent training trajectories from $\mathbb{P}_{\text{demo}}$, $\mathbf{J}_{\text{DEMO},T}(\hat{\pi}; \mathbb{P}_{\text{demo}}) \lesssim n^{-\alpha}$, $\alpha \in (0, 1]$. Such

---

[1]We refer to inputs and actions interchangeably.

bounds can realized by standard learning algorithms such as empirical risk minimization (ERM), i.e. behavior cloning, but are not restricted to it; any algorithm that seeks to perfectly match the expert will drive low on-expert error $\mathbf{J}_{\text{DEMO},T}(\hat{\pi}; \mathbb{P}_{\pi^\star})$. We defer discussions of supervised learning formalisms to Appendix B. Gauging how well various algorithmic interventions (or lack thereof) mitigate compounding errors therefore translates to bounding the **trajectory error** $\mathbf{J}_{\text{TRAJ},T}$ in terms of the **on-expert error** $\mathbf{J}_{\text{DEMO},T}$.

**The Compounding Errors problem.** We now formally describe the compounding errors problem. Let alg be a (possibly randomized) mapping from a sample of $n$ trajectories $S_n \overset{\text{i.i.d}}{\sim} \mathbb{P}_{\text{demo}}$ to an imitator policy $\hat{\pi} \sim \text{alg}(S_n)$. The problem instance suffers *exponential compounding errors* if:

$$\mathbb{E}_{\hat{\pi}, S_n}[\mathbf{J}_{\text{TRAJ},T}(\hat{\pi})] \gtrsim C^T \cdot \mathbb{E}_{\hat{\pi}, S_n}[\mathbf{J}_{\text{DEMO},T}(\hat{\pi}; \mathbb{P}_{\text{demo}})], \tag{2.3}$$

for some $C > 1$. In other words, imitating via empirical risk minimization on a given demonstration distribution $\mathbb{P}_{\text{demo}}$ leads to learned policies $\hat{\pi}$ that suffer exponentially more **trajectory error** rolled out in closed-loop compared to their **on-expert** regression error. As proposed in prior work (Tu et al., 2022; Pfrommer et al., 2022), compounding error can be understood through the lens of control-theoretic *stability*, which describes the sensitivity of the dynamics to perturbations of the state or input. By the control-theoretic nature of the ensuing definitions and analysis, **we provide a concise primer to key control-theoretic concepts in Appendix C**. We consider a notion of *incremental stability* (Angeli, 2002; Tran et al., 2016).

**Definition 2.1** (EISS, Figure 5). A system $\mathbf{x}_{t+1} = f(\mathbf{x}_t, \mathbf{u}_t)$ is $(C_{\text{ISS}}, \rho)$-*exponentially incrementally input-to-state stable* (EISS) if for all pairs of initial conditions $(\mathbf{x}_1, \mathbf{x}'_1)$ and input sequences $(\{\mathbf{u}_t\}_{t \geq 1}, \{\mathbf{u}'_t\}_{t \geq 1})$, there exist constants $C_{\text{ISS}} \geq 1$, $\rho \in (0,1)$[2] such that for any $t \geq 1$:

$$\|\mathbf{x}_t - \mathbf{x}'_t\| \leq C_{\text{ISS}} \rho^{t-1} \|\mathbf{x}_1 - \mathbf{x}'_1\| + C_{\text{ISS}} \sum_{k=1}^{t-1} \rho^{t-1-k} \|\mathbf{u}_k - \mathbf{u}'_k\|, \quad t \geq 1.$$

We say a policy-dynamics pair $(\pi, f)$ is $(C_{\text{ISS}}, \rho)$-EISS if the induced closed-loop dynamics $f^\pi$ is $(C_{\text{ISS}}, \rho)$-EISS. We also denote the shorthands $C_{\text{stab}} \triangleq \frac{C_{\text{ISS}}}{1-\rho}$, $c_{\text{stab}} \triangleq C_{\text{stab}}^{-1}$.

In other words, incremental stability ensures that bounded input perturbations lead to bounded future state deviations, with their effect decaying in time.[3] Thus, incremental stability can be viewed as a continuous-control analog to notions of "recoverability" (Ross et al., 2011; Foster et al., 2024). Henceforth, we will refer to "stability" and EISS interchangeably. Particularly relevant to Section 3, we note that "open-loop stable" dynamics (i.e., satisfying EISS even in the absence of feedback policies) are salient in various robotic applications via low-level controllers.

**Incremental Stability as a Natural Abstraction for End-Effector Control.** End-effector control outputs desired motion relative to position, which are then tracked by low-level high-frequency controllers, such as proportional-derivative (PD) controllers. Hence, the presence of a **low-level tracking controller renders the closed-loop between the desired position and system state incrementally stable** (c.f. Block et al. (2024)). In other words, end-effector control renders imitation of, e.g., *position* commands as taking place in an open-loop stable dynamical system.

Our base assumption across both Section 3 and Section 4 is that the *expert-induced* closed-loop system $f^{\pi^\star}$ is incrementally stable. This formalizes a notion of expert robustness by implying the expert can eventually recover from bounded input perturbations. As strong as this may seem, EISS of the expert does not assume away the compounding errors issue (see Theorem A); for example, if the candidate policy $\hat{\pi}$ destabilizes the system, the resulting "input perturbations" to $f^{\pi^\star}$ are exponentially growing.

**Theorem A** (Motivating lower bounds, Informal vers. of Simchowitz et al. (2025, Theorems 1 & 4)). *There exists families $\mathcal{P}_{\text{stab}}$ and $\mathcal{P}_{\text{unst}}$ of policies and dynamics such that:*

*(i) For every $(\pi, g) \in \mathcal{P}_{\text{stab}}$, $g$ **is open-loop EISS and** $(\pi, g)$ **is closed-loop EISS**, and $\pi, g$ are Lipschitz and smooth. However, any learning algorithm which returns smooth, Lipschitz, Markovian policies with state-independent stochasticity must suffer exponential-in-$T$ compounding error (2.3) when learning from $n$ expert trajectories from some $(\pi, g) \in \mathcal{P}_{\text{stab}}$.*

---

[2] We note traditional definitions of nonlinear stability may track separate $\beta, \gamma$ for the transient bound $\beta(\|\mathbf{x} - \mathbf{x}'\|, t)$ and the input gain $\gamma(\|\mathbf{u} - \mathbf{u}'\|)$. It suffices for our purposes to combine these under $C_{\text{ISS}}$ for clarity.

[3] The stability definition and ensuing results can be loosened to polynomial decay or local variants with appropriate modifications, though we note the lower-bound constructions in Theorem A are EISS systems.

*(ii) For every $(\pi, g) \in \mathcal{P}_{\text{unst}}$, $(\pi, g)$ **is closed-loop EISS but** $g$ **need not be open-loop EISS**, and $\pi, g$ are Lipschitz and smooth. However, any learning algorithm, without restriction, suffers exponential-in-$T$ compounding error (2.3) when learning from $n$ expert trajectories on some $(\pi, g) \in \mathcal{P}_{\text{stab}}$.*

The bounds ensure that for at least one instance $(\pi, g)$ in $\mathcal{P}_{\text{stab}}$ (resp. $\mathcal{P}_{\text{unst}}$), the learner suffers exponential-in-$T$ compounding error if that instance $(\pi, g)$ is the ground truth, and the learner receives $n$ expert demonstrations from that instance. **Our results constitute positive converses**:

- When $g$ is open-loop EISS, the chunked policies prescribed by Practice 1 are smooth and Lipschitz but non-Markovian (by virtue of being chunked), and thus bypass Theorem A.(i)
- If $g$ is not necessarily EISS, Practice 2 *alters the distribution* over expert demonstrations to circumvent Theorem A.(ii). Note that Theorem A.(ii) precludes any purely algorithmic changes that do not change the distribution $\mathbb{P}_{\pi^\star}$ over expert demonstrations.

We note Simchowitz et al. (2025) also provide stylized policies which bypass the particular construction used in Theorem A.(i); Practice 1 serves as a natural, generally applicable solution, and Practice 2 circumvents the more challenging setting that even these stylized policies cannot address.

**Additional Notation.** **Blue** (e.g. $\pi^\star$) indicates expert-induced quantities, and **red** indicates quantities induced by a learned policy (e.g. $\hat{\pi}$). Positive semi-definite matrices are indicated by $\mathbf{Q} \succeq \mathbf{0}$, and the corresponding partial order $\mathbf{P} \succeq \mathbf{Q} \implies (\mathbf{P} - \mathbf{Q}) \succeq \mathbf{0}$. We use $\lesssim, \approx$ to omit universal constants. In the main body, we also use $O_\star(\cdot)$ to omit *polynomial* dependence on instance-dependent constants, but not algorithm-dependent constants or horizon $T$, e.g. $\frac{T C_{\text{ISS}}}{1-\rho} \sigma_{\mathbf{u}}^2 = O_\star(T \sigma_{\mathbf{u}}^2)$.

## 3 ACTION-CHUNKING SUFFICES IN OPEN-LOOP STABLE SYSTEMS

Action-chunking is a popular practice in modern sequential modeling pipelines, where a policy predicts a sequence of actions, of which some number are played *in open-loop* (Chen et al., 2021; Chi et al., 2023; Shafiullah et al., 2022). There are various intuitions of the practical benefits of action-chunking, ranging from: 1. robustness to non-Markovian / partial observability quirks in the data (Liu et al., 2025), 2. amenability to multi-modal[4] prediction, 3. improved representation learning via multi-step prediction, and 4. simulating receding-horizon control. Yet, we show that even in control settings with *unimodal, Markovian, state-feedback* experts, action-chunking serves a critical role in subverting exponential compounding errors. All proofs and extended details in this section are contained in Appendix D. We may conveniently describe chunking as follows.

**Definition 3.1** (Chunking Policy). A chunking policy is specified by a chunk-length $\ell$, and mappings $\text{chunk}_i[\pi] : \mathcal{X} \to \mathcal{U}, i \in [\ell]$ such that, for $k \in \mathbb{Z}_{\geq 0}$ and $i \in [\ell]$ and $t = k\ell + i$ for, $\pi(\mathbf{x}_{1:t}, \mathbf{u}_{1:t-1}, t) = \text{chunk}_i[\pi](\mathbf{x}_{k\ell+1}, i)$. We also write $\text{chunk}[\pi](\mathbf{x}) = (\text{chunk}_1(\mathbf{x}), \ldots, \text{chunk}_\ell(\mathbf{x}))$. For simplicity, we always assume $\ell$ divides $T - 1$.

**Practice 1** (Learning over Chunked Policies). We sample $S_n$ as denote $n$ i.i.d. trajectories drawn from the expert distribution $\mathbb{P}_{\pi^\star}$. We aim to find $\tilde{\pi}$ from a class of length-$\ell$ chunked policies, $\Pi_{\text{chunk},\ell}$, defined formally in Definition 3.2 that attains low **on-expert error**, e.g., by empirical risk minimization. We note that for chunked policies,

$$\mathbf{J}_{\text{DEMO},T}(\tilde{\pi}; \mathbb{P}_{\pi^\star}) = \mathbb{E}_{\mathbb{P}_{\pi^\star}} \left[ \sum_{k=1}^{\lceil T-1/\ell \rceil} \| \mathbf{u}_{1+(k-1)\ell:k\ell}^{\pi^\star} - \text{chunk}[\tilde{\pi}](\mathbf{x}_{(k-1)\ell}^{\pi^\star}) \|^2 \right]. \tag{3.1}$$

We now formally define the policies induced by chunking with a dynamics model.

**Definition 3.2** (Induced Chunking Policy). Let $g : \mathcal{X} \times \mathcal{U} \to \mathcal{U}$ be a dynamics map (possibly not the true dynamics $f$), and $\pi : \mathcal{X} \to \mathcal{U}$ a Markovian, deterministic policy. Given chunk length $\ell \in \mathbb{N}$, we define the induced chunked policy $\tilde{\pi} = \text{chunked}(\pi, g, \ell)$, $\tilde{\pi} : \mathcal{X} \to (\mathcal{U})^\ell$ as returning

$$\text{chunk}[\tilde{\pi}](\mathbf{x}) = \left( \pi(\mathbf{x}), \pi(g^\pi(\mathbf{x})), \pi((g^\pi)^2(\mathbf{x})), \ldots, \pi((g^\pi)^{\ell-1}(\mathbf{x})) \right), \tag{3.2}$$

where above $g^\pi(\mathbf{x}) \triangleq g(\mathbf{x}, \pi(\mathbf{x}))$, and $(g^\pi)^i$ is understood as repeated composition.

In other words, $\text{chunked}(\pi, g, \ell)$ returns a policy that, conditioning on the current state, outputs the next $\ell$ actions given by simulating $\pi$ on dynamics $g$ in closed-loop. We state the core assumptions.

---

[4]In the sense of a distribution having multiple modes.

**Figure 2:** Success rates as a function of *evaluated* action-chunk lengths on the challenging `robomimic` "tool_hang" environment with full-state observations. **Left:** Each line corresponds to a model trained for a given prediction horizon on 100 expert trajectories. Each point corresponds to the model evaluating a given chunk length ranging from receding-horizon ($\ell = 1$) to the full chunk. While prediction horizon has some (transient) effect, *evaluating* slightly longer chunks improves success drastically. **Right:** We repeat a similar set-up with 50 expert training trajectories. We see that noise-injection (Practice 2) can also synergize in this open-loop stable setting (see Appendix F), though requires modifying the data-collecting procedure rather than simply adjusting policy parameterization and evaluation as in AC.

**Assumption 3.1** (Regularity and Stability). We make the following assumptions: 1. the true dynamics $f$ are $(C_{\mathrm{ISS}}, \rho)$-EISS in open-loop, without loss of generality with $\rho \geq 1/e$, 2. all base policies $\pi \in \Pi \cup \{\pi^\star\}$ in consideration are $L_\pi$-Lipschitz: $\|\pi(\mathbf{x}) - \pi(\mathbf{x}')\| \leq L_\pi \|\mathbf{x} - \mathbf{x}'\|$.

All ensuing results stem from the following key result.

**Proposition 3.1.** *Let Assumption 3.1 hold. Let $(\hat{\pi}, \hat{f})$ be a policy-dynamics pair that is $(C_{\mathrm{ISS}}, \rho)$-EISS, and consider the corresponding chunked policy $\tilde{\pi} = \mathsf{chunked}(\hat{\pi}, \hat{f}, \ell)$. Then the closed-loop system the* chunked policy *induces on the true dynamics $(\tilde{\pi}, f)$ is $(\tilde{C}, \tilde{\rho})$-EISS, where $\tilde{C} = \log(1/\rho)^{-1} \cdot \mathrm{poly}(L_\pi, C_{\mathrm{ISS}})$ and $\tilde{\rho} = \rho^{1/2}$, as long as the chunk length is sufficiently long: $\ell > \log(1/\rho)^{-1} \cdot \log(\mathrm{poly}(L_\pi, C_{\mathrm{ISS}}))$.*

Therefore, combining Proposition 3.1 and Proposition 3.2 leads to the following compounding error guarantee on any sufficiently chunked policy. This result states that: as long as a policy "believes" it stabilizes the *simulated* dynamics at hand, then it is guaranteed to be stable on the actual dynamics **if it is chunked accordingly**. For notational simplicity, we set the prediction horizon and the executed chunk length (i.e., how many predicted actions are played before re-predicting) the same at $\ell$. However, we note that the requirement $\ell > \log(1/\rho)^{-1} \cdot \log(\mathrm{poly}(L_\pi, C_{\mathrm{ISS}}))$ is on the *executed chunk length*: playing a chunked policy $\tilde{\pi} = \mathsf{chunked}(\hat{\pi}, \hat{f}, \ell)$ in Markovian (i.e., receding-horizon) fashion does not subvert Theorem A.(i). Crucially, without action-chunking, open-loop stability of the nominal dynamics $f$ and closed-loop stability of the expert $(\pi^\star, f)$ does not imply closed-loop stability of $(\hat{\pi}, f)$ for the learned policy without action chunking $\ell = 1$. Contrast this to Proposition 3.1, which depends only on the stability properties of the true system $f$ and the closed-loop simulated system $(\hat{\pi}, \hat{f})$, and requires *no assumption on the closeness of $\hat{f}$ to $f$, or $\hat{\pi}$ to any reference policy*. Define the class of possible policy-dynamics pairs $\mathcal{P} \triangleq \{(\pi, g) : (\pi, g) \text{ is } (C_{\mathrm{ISS}}, \rho)\text{-EISS}\}$, and the induced length-$\ell$ chunked policy class: $\Pi_{\mathrm{chunk}, \ell} \triangleq \{\tilde{\pi} = \mathsf{chunked}(\pi, g, \ell) : (\pi, g) \in \mathcal{P}\}$. A key consequence of chunked policies inducing stable closed-loop dynamics is that they induce limited compounding error.

**Proposition 3.2.** *Let Assumption 3.1 hold. Let $\tilde{\pi} = \mathsf{chunked}(\hat{\pi}, \hat{f}, \ell) \in \Pi_{\mathrm{chunk}, \ell}$, and assume $(\hat{\pi}, \hat{f})$, $(\tilde{\pi}, f)$ are $(\tilde{C}, \tilde{\rho})$-EISS. Then, the following bound holds:*

$$\mathbf{J}_{\mathrm{TRAJ}, T}(\tilde{\pi}) \leq \mathrm{poly}\left(L_\pi, \tilde{C}, \frac{1}{1-\tilde{\rho}}\right) \mathbf{J}_{\mathrm{DEMO}, T}(\tilde{\pi}; \mathbb{P}_{\pi^\star}).$$

**Theorem 1.** *Let Assumption 3.1 hold. For sufficiently long chunk-length: $\ell > \log(1/\rho)^{-1} \cdot \log(\mathrm{poly}(L_\pi, C_{\mathrm{ISS}}))$, let $\tilde{\pi} = \mathsf{chunked}(\hat{\pi}, g, \ell) \in \Pi_{\mathrm{chunk}, \ell}$. We have the trajectory-error bound:*

$$\mathbf{J}_{\mathrm{TRAJ}, T}(\tilde{\pi}) \leq O_\star(1) \mathbf{J}_{\mathrm{DEMO}, T}(\tilde{\pi}; \mathbb{P}_{\pi^\star}).$$

Theorem 1 implies that when the ambient dynamics $f$ are EISS, then a sufficiently chunked imitator policy will accrue limited compounding errors—**horizon-free**—relative to the on-expert error it sees. In particular, given $\tilde{\pi}$ attaining regression generalization error $\mathbf{J}_{\mathrm{DEMO}, T}(\tilde{\pi}; \mathbb{P}_{\pi^\star}) \lesssim n^{-\alpha}$, this implies $\mathbf{J}_{\mathrm{TRAJ}, T}(\tilde{\pi}) \leq O_\star(1) n^{-\alpha}$. To summarize the key takeaways of the theoretical results:

> **Key Findings**
> - Theorem 1 implies that **executing chunks of actions in open-loop is key**; multi-step prediction alone cannot subvert Theorem A.(i) if the policy is only played in receding-horizon fashion.
> - **Requisite chunk lengths are small**, scaling logarithmically with system-dependent constants, and longer chunk lengths beyond that point provide marginal benefit (and clashes with practical concerns of prolonged open-loop execution).
> - Action-chunking is demonstrably crucial in **state-based, deterministic control**, revealing its key role in IL independent of non-Markovianity or partial observability of demonstrations.
> - However, **action chunking is not a silver bullet:** it relies crucially on stability of the *open-loop* system, which typically requires end-effector control to ensure in robot manipulation setups.

## 4 NOISE INJECTION MITIGATES COMPOUNDING ERROR UNDER SMOOTH, UNSTABLE DYNAMICS

We now consider the difficult setting where the ambient dynamics $f$ may not be open-loop stable. In this case, purely algorithmic interventions like action-chunking are generally insufficient, as erroneous actions can quickly lead to unstable behavior. In fact, we recall Theorem A states that *no* algorithm, even permitting stochastic and non-Markovian policies, can circumvent exponential compounding errors in the worst-case, provided only data from the expert-induced law $\mathbb{P}_{\pi^\star}$. This necessitates altering the demonstration distribution $\mathbb{P}_{\mathrm{demo}}$ beyond the expert's $\mathbb{P}_{\pi^\star}$, i.e., some form of additional **exploratory data collection is required**. In particular, prior approaches such as DAG-GER (Ross et al., 2011) and DART (Laskey et al., 2017) can be summarized as attempting to witness *how the expert recovers from errors*, where the former queries the expert along learned-policy rollouts, and the latter injects policy-shaped noise into the expert—similar to the approach we propose.

**Exploratory Data Collection.** However, beyond the motivating intuition, we still lack fine-grained insights into what kinds of recovery or policy errors need to be witnessed to circumvent compounding errors, if even possible. Furthermore, these works require *iterative* rounds of expert data collection based on learned policy statistics. Our point of departure is the following: if we are tracking the expert sufficiently closely, we should only need to witness how the expert policy recovers *near the expert distribution*. To this end, we consider arguably the simplest approach to inducing *local* exploration in the expert dataset: **noise injection**. In the discussion below, we fix a *noise level* $\sigma_{\mathbf{u}} > 0$, which controls the magnitude of the noise added, and a *mixture fraction* $\alpha \in [0, 1]$, that controls the proportion of trajectories collected without noise injection.

**Definition 4.1.** We define the *expert distribution under noise injection* as the distribution $\mathbb{P}_{\pi^\star, \sigma_{\mathbf{u}}}$ over trajectories $(\tilde{\mathbf{x}}_t, \tilde{\mathbf{u}}_t)_{t \geq 1}$ with $\tilde{\mathbf{x}}_1 \sim D$, and $\tilde{\mathbf{u}}_t = \pi^\star(\tilde{\mathbf{x}}_t)$, $\tilde{\mathbf{x}}_{t+1} = f(\tilde{\mathbf{x}}_t, \tilde{\mathbf{u}}_t + \sigma_{\mathbf{u}} \mathbf{z}_t)$ for $t \geq 1$, where $\mathbf{z}_t \overset{\mathrm{i.i.d}}{\sim} \mathrm{Unif}(\mathbb{B}^{d_u}(1))$ is drawn uniformly over the unit ball.[5]

In other words, noise injection collects trajectories induced when the expert's commanded actions are executed with additive noise $\sigma_{\mathbf{u}} \mathbf{z}_t$. We then consider fitting a policy $\pi$ by augmenting standard (un-noised) expert trajectories with noise-injected ones.

**Practice 2** (Exploratory Data Collection via Expert Noise Injection). For the noise-injected distribution $\mathbb{P}_{\pi^\star, \sigma_{\mathbf{u}}}$ defined above, provide a sample $S_{n, \sigma, \alpha}$ of $(\mathbf{x}_t^{(i)}, \mathbf{u}_t^{(i)})_{1 \leq t \leq T, 1 \leq i \leq n}$, where for $1 \leq i \leq \lfloor \alpha n \rfloor$ the trajectories are i.i.d. from $\mathbb{P}_{\pi^\star}$, and the remaining trajectories are drawn i.i.d. from $\mathbb{P}_{\pi^\star, \sigma_{\mathbf{u}}}$. Define the corresponding mixture distribution $\mathbb{P}_{\pi^\star, \sigma_{\mathbf{u}}, \alpha} \triangleq \alpha \mathbb{P}_{\pi^\star} + (1 - \alpha) \mathbb{P}_{\pi^\star, \sigma_{\mathbf{u}}}$. We then find $\hat{\pi}$ that attains low $\mathbf{J}_{\mathrm{DEMO}, T}(\hat{\pi}; \mathbb{P}_{\pi^\star, \sigma_{\mathbf{u}}, \alpha})$, e.g., by empirical risk minimization.

Notably, Practice 2 only collects data *once* before fitting the policy $\hat{\pi}$, and thus does not depend on learned policy rollouts. We now lay out the core regularity assumptions in this section.

**Assumption 4.1** (Regularity and Stability). Recall that a function $h : \mathbb{R}^d \to \mathbb{R}^p$ $C$-smooth if for all $\mathbf{x}, \mathbf{x}' \in \mathbb{R}^d$, $\|\nabla_{\mathbf{x}} h(\mathbf{x}) - \nabla_{\mathbf{x}} h(\mathbf{x}')\|_2 \leq C \|\mathbf{x} - \mathbf{x}'\|$. We make the following assumptions: 1. the expert policy and true dynamics $(\pi^\star, f)$ are $(C_\pi, C_{\mathrm{reg}})$-smooth, respectively, 2. all policies $\pi \in \Pi \cup \{\pi^\star\}$ are $L_\pi$-Lipschitz, 3. the closed-loop system induced by $(\pi^\star, f)$ is $(C_{\mathrm{ISS}}, \rho)$-EISS.

To understand the exploratory role of noise-injection, we gather intuition through linearizations.

---

[5]Our results hold for generic bounded noise, but it suffices to consider $\mathbf{z} \sim \mathrm{Unif}(\mathbb{B}^{d_u}(1))$ or $\mathrm{Unif}(\mathbb{S}^{d_u}(1))$.

**Analysis via Linearizations.** Our analysis of Practice 2 uses smoothness of the dynamics and policy to reason about its local linear approximation to the dynamical system along a given trajectory, called the *Jacobian linearization*.

**Definition 4.2** (Jacobian Linearization). For a fixed initial condition $\mathbf{x}_1^{\pi^\star} \sim D$, we define the *Jacobian Linearization* of the expert trajectory by setting $\mathbf{x}_{t+1}^{\pi^\star} = f^{\pi^\star}(\mathbf{x}_t^{\pi^\star})$, $\mathbf{u}_t^{\pi^\star} = \pi^\star(\mathbf{x}_t^{\pi^\star})$, and define a linear time-varying system determined by the transition matrices: $\mathbf{A}_t = \nabla_\mathbf{x} f(\mathbf{x}_t^{\pi^\star}, \mathbf{u}_t^{\pi^\star})$, $\quad \mathbf{B}_t = \nabla_\mathbf{u} f(\mathbf{x}_t^{\pi^\star}, \mathbf{u}_t^{\pi^\star})$, as well as the local linearization of the controller $\mathbf{K}_t^{\pi^\star} = \nabla_\mathbf{x} \pi^\star(\mathbf{x}_t^{\pi^\star})$.

For a smooth dynamical system, consider a perturbed trajectory $\tilde{\mathbf{x}}_t$ given by $\tilde{\mathbf{x}}_{t+1} = f(\tilde{\mathbf{x}}_t, \tilde{\mathbf{u}}_t)$, $\tilde{\mathbf{x}}_1 = \mathbf{x}_1^{\pi^\star} + \delta\mathbf{x}_1$, and $\tilde{\mathbf{u}}_t = \pi^\star(\tilde{\mathbf{x}}_t) + \delta\mathbf{u}_t$. Then, the linearization is such that the trajectory differences $\delta\mathbf{x}_t = \tilde{\mathbf{x}}_t - \mathbf{x}_t^{\pi^\star}$ satisfy up to first-order: $\delta\mathbf{x}_{t+1} \approx (\mathbf{A}_t + \mathbf{B}_t\mathbf{K}_t^{\pi^\star})\delta\mathbf{x}_t + \mathbf{B}_t\delta\mathbf{u}_t$. Therefore, for sufficiently small perturbations $\delta\mathbf{u}_t$, the evolution of the trajectory difference is primarily determined by the linear transition matrices derived from linearizations along the clean expert trajectory. We now introduce a measure of how "sensitive" the closed-loop dynamics is around the expert trajectory.

**Definition 4.3** (Linearized Controllability Gramian). The $t$-step controllability Gramian is defined as: $\mathbf{W}_{1:t}^\mathbf{u}(\mathbf{x}_1^{\pi^\star}) \triangleq \sum_{s=1}^{t-1} \mathbf{A}_{s+1:t}^{\mathrm{cl}} \mathbf{B}_s \mathbf{B}_s^\top \mathbf{A}_{s+1:t}^{\mathrm{cl}}{}^\top$, where we define *closed-loop transfer matrix* as $\mathbf{A}_{s:t}^{\mathrm{cl}} = (\mathbf{A}_{t-1} + \mathbf{B}_{t-1}\mathbf{K}_{t-1}^{\pi^\star})(\mathbf{A}_{t-2} + \mathbf{B}_{t-2}\mathbf{K}_{t-2}^{\pi^\star}) \cdots (\mathbf{A}_s + \mathbf{B}_s\mathbf{K}_s^{\pi^\star})$.

The linear controllability Gramian can be interpreted as capturing the sensitive directions of the closed-loop dynamics to perturbations (see Appendix C). In particular, $\mathbf{W}_{1:t}^\mathbf{u}(\mathbf{x}_1^{\pi^\star})$ is the (linearized) covariance matrix of the trajectory difference $\delta\mathbf{x}_t$ under uniform stochastic perturbations (e.g., $\delta\mathbf{u} \sim \mathcal{N}(\mathbf{0}, \mathbf{I})$). Therefore, directions corresponding to large eigenvalues of $\mathbf{W}_{1:t}^\mathbf{u}(\mathbf{x}_1^{\pi^\star})$ correspond to axes of $\delta\mathbf{x}_t$ that are most magnified under perturbation, and small (or zero) eigendirections correspond to those that naturally dissipate (or are unreachable). Therefore, under mean-zero, $\sigma^2$-covariance noise-injection $\delta\mathbf{u}_t$, the local excitation (i.e. *exploration*) around the expert state $\mathbf{x}_t^{\pi^\star}$ is approximated by: $\mathbb{E}[\delta\mathbf{x}_t\delta\mathbf{x}_t^\top \mid \mathbf{x}_1^{\pi^\star}] \approx \sigma^2\mathbf{W}_{1:t}^\mathbf{u}(\mathbf{x}_1^{\pi^\star})$. Though the Gramian provides a notion of local exploration, fully realizing its benefits requires certain crucial subtleties not captured in prior literature.

**Suboptimal Approaches.** We now remark on subtle but important features of Practice 2.

- Actions under $\mathbb{P}_{\pi^\star,\sigma_\mathbf{u}}$ are executed noisily, but recorded action labels are *noiseless* $\tilde{\mathbf{u}}_t = \pi^\star(\tilde{\mathbf{x}}_t)$, preventing additional regression error. This may run counter to RL theory, where noising the *policy*, e.g. $\tilde{\mathbf{u}}_t \sim \mathcal{N}(\pi^\star(\tilde{\mathbf{x}}_t), \sigma_\mathbf{u}^2\mathbf{I})$ may be desirable to induce *coverage* (Jiang & Xie, 2024).
- Only a proportion $(1 - \alpha)$ of trajectories are noise-injected; the rest are clean expert trajectories.

We relegate a detailed description of standard RL and control-theoretic perspectives (and their deficiencies) to Appendix E.1. In either case, i.e. if a noisy policy is adopted, or only noise-injected trajectories are collected, we encounter a fundamental problem. Due to the non-linearity of the dynamics, the noised actions induce a trajectory drift compared to the nominal noiseless expert. This drift means policies fitted on the noisy trajectories, even with clean action labels $\mathbb{P}_{\pi^\star,\sigma_\mathbf{u}}$, necessarily accrue an additive trajectory error scaling with $\sigma_\mathbf{u}$, **regardless of the on-expert regression error**.

**Proposition 4.1** (Drift lower bound, informal). *For any given $\sigma_\mathbf{u} > 0$ and $C_\pi > 0$, there exists a pair of two $C_\pi$-smooth policies $\pi_1, \pi_2$ such that one trajectory from the rollout distribution under each can distinguish them perfectly, but given trajectories with $\sigma_\mathbf{u}^2$-noise injection, any learning algorithm on $n$ trajectories sampled under either $\pi_1, \pi_2$ will yield a policy $\pi$ that incurs $\mathbf{J}_{\mathrm{TRAJ},T}(\pi) \geq \Omega(C_\pi^2\sigma_\mathbf{u}^4)$ trajectory error with probability $\gtrsim 1 - n\exp(-\sqrt{d_u})$.*

The formal statement and set-up of Proposition 4.1 is found in Appendix E.7. We notice that this bound scales with $C_\pi$, indicating that smoothness is a key quantity in any argument based on noising. A consequence of an additive drift is that it suggests an "optimal" choice of $\sigma_\mathbf{u} > 0$ is miniscule: $\mathbf{J}_{\mathrm{TRAJ},T}(\hat{\pi}) \lesssim \mathrm{poly}(\sigma_\mathbf{u}^{-1})\mathbf{J}_{\mathrm{DEMO},T}(\hat{\pi}; \mathbb{P}_{\mathrm{demo}}) + \mathrm{poly}(\sigma_\mathbf{u})$ implies $\sigma_\mathbf{u}^\star \approx \mathrm{poly}(\mathbf{J}_{\mathrm{DEMO},T}(\hat{\pi}; \mathbb{P}_{\mathrm{demo}}))$, which we will see is a suboptimal scaling in theory and empirically.

As for a corresponding upper bound, let us first entertain the implications of the too-strong assumption of *one-step controllability*, where $\mathbf{W}_{1:t}^\mathbf{u}(\mathbf{x}_1^{\pi^\star}) \succeq \underline{\lambda}_\mathbf{W}\mathbf{I}_{d_x}$, $\underline{\lambda}_\mathbf{W} > 0$, for all $t \geq 2$, such that under an appropriate input sequence, the (linearized) expert system $f^{\pi^\star}$ can reach any state at any time. Therefore, noise-injection will excite all modes of the linearized system, translating to persistency-of-excitation (PE) (Bai & Sastry, 1985) as traditional control theory would desire (see Appendix E.1). This yields the following (suboptimal) bound when imitating over $\mathbb{P}_{\pi^\star,\sigma_\mathbf{u}}$.

**Suboptimal Proposition 4.2.** Let Assumption 4.1 hold, and let $\mathbf{W}_{1:t}^\mathbf{u}(\mathbf{x}_1^{\pi^\star}) \succeq \underline{\lambda}_\mathbf{W}\mathbf{I}_{d_x}$, $t \geq 2$ w.p. 1 over $\mathbf{x}_1^{\pi^\star} \sim D$ for some $\underline{\lambda}_\mathbf{W} > 0$. Let $\hat{\pi}$ be a $C_\pi$-smooth candidate policy. For $\sigma_\mathbf{u}^2$ that satisfies

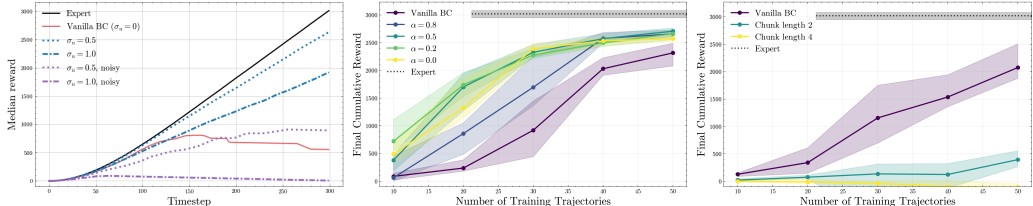

**Figure 3:** Features of Practice 2 exhibited on HalfCheetah-v5. **Left:** we compare collecting clean action labels as in Practice 2, versus the noised ones as in a coverage-based approach. We note $\sigma_{\mathbf{u}} = 1$ corresponds to sizable entry-wise input perturbations $\approx 0.4$ on an action space of $[-1, 1]^6$. Imitating with noisy labels is therefore catastrophic, yet using clean labels achieves improved performance. **Center:** Fixing $\sigma_{\mathbf{u}} = 0.5$, we vary the proportion of clean expert trajectories $\alpha \in [0, 1]$. The performance difference is marginal past a sufficient number of *noised trajectories*; see Eq. (4.1). **Right:** naively action-chunking (Practice 1) is disastrous due to open-loop instability; compare to Figure 2. Experiment details in Appendix G.

$\sigma_{\mathbf{u}}^2 \lesssim O_\star(\mathrm{poly}(1/C_\pi, 1/C_{\mathrm{reg}})) \underline{\lambda}_{\mathbf{W}}$, we have:

$$\mathbf{J}_{\mathrm{TRAJ},T}(\hat{\pi}) \lesssim O_\star(T) \underline{\lambda}_{\mathbf{W}}^{-1} \left( \frac{1}{\sigma_{\mathbf{u}}^2} \mathbf{J}_{\mathrm{DEMO},T}(\hat{\pi}; \mathbb{P}_{\pi^\star, \sigma_{\mathbf{u}}}) + C_\pi^2 C_{\mathrm{stab}}^2 \sigma_{\mathbf{u}}^2 \right).$$

The full statement and proof can be found in Appendix E.3. Though this bound avoids exponential-in-$T$ compounding trajectory error, it has several shortcomings. Besides the strictness of one-step controllability—or controllability at all (see Appendix C)—the bound suffers: 1. a drift term that scales as $\sigma_{\mathbf{u}}^2$, which is even worse than Proposition 4.1 suggests, 2. the requirement on $\sigma_{\mathbf{u}}$ and resulting bound scaling with $\underline{\lambda}_{\mathbf{W}}$, which is miniscule for Gramians with fast-decaying spectra. As such, a combination of algorithmic (e.g., $\mathbb{P}_{\pi^\star, \sigma_{\mathbf{u}}} \to \mathbb{P}_{\pi^\star, \sigma_{\mathbf{u}}, \alpha}$) and analytical innovations are required to advance the result.

## 4.1 A SHARP ANALYSIS OF EXPLORATORY DATA: EXCITING THE UNSTABLE DIRECTIONS

In light of Suboptimal Proposition 4.2, we make a few key observations. Firstly, *compounding errors are not arbitrary state perturbations*: they result from policy errors, and thus enter the state via the input channels. For smooth systems, this implies the trajectory error is primarily contained in the *controllable subspace* $\mathrm{range}(\mathbf{W}_{1:t}^{\mathbf{u}})$. However, nonlinearity in the dynamics $f$ and policies $\hat{\pi}, \pi^\star$ means error will leak outside of $\mathrm{range}(\mathbf{W}_{1:t}^{\mathbf{u}})$, which would seem to require PE, i.e. full-dimensional coverage, to detect. Our first key insight is that **as long as we enforce low error on the controllable subspace, the nonlinear error automatically regulates itself.**

**Proposition 4.3.** *Let Assumption 4.1 hold, and assume the candidate policy $\hat{\pi}$ is $C_\pi$-smooth. Fix $\mathbf{x}_1^{\hat{\pi}} = \mathbf{x}_1^{\pi^\star} = \mathbf{x}_1$, and define $\mathcal{R}_t^{\pi^\star} \triangleq \mathrm{range}(\mathbf{W}_{1:t}^{\mathbf{u}})$. Then for any given $\varepsilon \in [0, 1]$, $T \in \mathbb{N}$, as long as:*

$$\max_{1 \leq t \leq T-1} \sup_{\|\mathbf{v}\| \leq 1, \|\mathbf{w}\| \leq 1, \mathbf{w} \in \mathcal{R}_t^{\pi^\star}} \|(\hat{\pi} - \pi^\star)(\mathbf{x}_t^{\pi^\star} + \varepsilon\mathbf{w} + O_\star(1)\varepsilon^2\mathbf{v})\| \leq c_{\mathrm{stab}}\varepsilon,$$

*we are guaranteed $\max_{1 \leq t \leq T} \|\mathbf{x}_t^{\hat{\pi}} - \mathbf{x}_t^{\pi^\star}\| \leq \varepsilon$.*

This result shows that if we ensure the "generic" error $\mathbf{v}$ term scales as $\varepsilon^2$, Lipschitzness of $\hat{\pi}, \pi^\star$ automatically ensures its contribution to $\hat{\pi} - \pi^\star$ is $o(c_{\mathrm{stab}}\varepsilon)$ for small enough $\varepsilon$. For smooth systems the nonlinear error is indeed higher-order. However, it remains to control the first-order error term $\varepsilon\mathbf{w}$ lying in $\mathcal{R}_t^{\pi^\star}$, where the bound in Suboptimal Proposition 4.2 incurs dependence on the smallest (positive) eigenvalues of $\mathbf{W}_{1:t}^{\mathbf{u}}$. This is unintuitive: small eigendirections of $\mathbf{W}_{1:t}^{\mathbf{u}}$ are those that are hard to excite, such that errors should compound slowly on them. Thus, there should exist a certain $O_\star(c_{\mathrm{stab}})$ threshold below which we may ignore them. **Restricting our attention to excitable directions means we only pay the statistical cost for the level of excitation we need.**

**Proposition 4.4.** *Let Assumption 4.1 hold. For $\mathbf{x}_1^{\pi^\star} \sim D$, let $\{(\lambda_{i,t}, \mathbf{v}_{i,t})\}_{i=1}^{d_x}$ be the eigenvalues and vectors of $\mathbf{W}_{1:t}^{\mathbf{u}}$, $t \geq 2$. Define $\mathcal{R}_t^{\pi^\star}(\lambda) \triangleq \mathrm{span}\{\mathbf{v}_{i,t} : \lambda_{i,t} \geq \lambda\}$ and $\mathcal{P}_{\mathcal{R}_t^{\pi^\star}(\lambda)}$ the corresponding orthogonal projection. Recall $\mathbb{P}_{\pi^\star, \sigma_{\mathbf{u}}, \alpha}$ and set $\alpha = 0.5$. Then, for $\sigma_{\mathbf{u}} \lesssim O_\star(\lambda)$, we have:*

$$\mathbb{E}_{\mathbb{P}_{\pi^\star}} \|\mathcal{P}_{\mathcal{R}_t^{\pi^\star}(\lambda)} \nabla_{\mathbf{x}}(\hat{\pi} - \pi^\star)(\mathbf{x}_t^{\pi^\star})\|_{\mathrm{op}}^2 \lesssim \frac{d_u}{\sigma_{\mathbf{u}}^2 \lambda} \mathbb{E}_{\mathbb{P}_{\pi^\star, \sigma_{\mathbf{u}}, \alpha}} \|(\hat{\pi} - \pi^\star)(\tilde{\mathbf{x}}_t)\|^2 + \frac{d_u \sigma_{\mathbf{u}}^2}{\lambda} C_\pi^2 C_{\mathrm{stab}}^4.$$

This is precisely where our *algorithmic* prescription arises: Proposition 4.4 suggests that certifying the learned policy $\hat{\pi}$ matches $\pi^\star$ up to first-order on $\mathcal{R}_t^{\pi^\star}(\lambda)$ requires data both at $\mathbf{x}_t^{\pi^\star}$ and

around it (e.g. via noise-injection). This translates to imitating on the *mixture distribution* $\mathbb{P}_{\pi^\star, \sigma_{\mathbf{u}}, \alpha}$. Therefore, combining Proposition 4.3, which translates imitating $\pi^\star$ well *in a neighborhood* to low trajectory error, with Proposition 4.4, which guarantees imitating on $\mathbb{P}_{\pi^\star, \sigma_{\mathbf{u}}, \alpha}$ matches $\nabla_{\mathbf{x}}(\hat{\pi} - \pi^\star)$ up to a flexible excitation level, leads to our main guarantee of Practice 2.

**Theorem 2.** *Let Assumption 4.1 hold. Let $\hat{\pi}$ be a $L_\pi$-Lipschitz, $C_\pi$-smooth policy. Then, for $\sigma_{\mathbf{u}} \lesssim O_\star(\mathrm{poly}(1/C_\pi, 1/C_{\mathrm{reg}})) = O_\star(1)$, we have:*

$$\mathbf{J}_{\mathrm{TRAJ},T}(\hat{\pi}) \lesssim O_\star(T)\, \sigma_{\mathbf{u}}^{-2} \mathbf{J}_{\mathrm{DEMO},T}(\hat{\pi}; \mathbb{P}_{\pi^\star, \sigma_{\mathbf{u}}, 0.5}).$$

*In particular, setting $\sigma_{\mathbf{u}} = O_\star(1)$, we have:* $\mathbf{J}_{\mathrm{TRAJ},T}(\hat{\pi}) \lesssim O_\star(T)\, \mathbf{J}_{\mathrm{DEMO},T}(\hat{\pi}; \mathbb{P}_{\pi^\star, \sigma_{\mathbf{u}}, 0.5})$.

Notably, by regressing on the mixture distribution $\mathbb{P}_{\pi^\star, \sigma_{\mathbf{u}}, \alpha}$, we are able to set $\sigma_{\mathbf{u}}$ as large as smoothness permits, rather than trading off with the regression error $\mathbf{J}_{\mathrm{DEMO},T}$ in Suboptimal Proposition 4.2. We note that a detailed analysis in fact reveals:

$$\mathbf{J}_{\mathrm{TRAJ},T}(\hat{\pi}) \lesssim O_\star(1)\, \mathbf{J}_{\mathrm{DEMO},T}(\hat{\pi}; \mathbb{P}_{\pi^\star}) + T \sum_{t=1}^{T-1} \mathbb{P}_{\pi^\star, \sigma_{\mathbf{u}}, \alpha} \big[ \|(\hat{\pi} - \pi^\star)(\tilde{\mathbf{x}}_t)\|^2 \gtrsim O_\star(\sigma_{\mathbf{u}}^2) \big] \quad (4.1)$$

In other words, the trajectory error can be bounded as a term scaling *horizon-free* with the *un-noised* on-expert error and a sum over "error events" on the *mixture* expert distribution. [6] On the other hand, if mild moment-equivalence conditions such as hypercontractivity (Wainwright, 2019; Ziemann & Tu, 2022) hold on the estimation error, then the dependence on both $T$ and $\sigma_{\mathbf{u}}$ can be attached to higher-order factors, e.g., $\mathbf{J}_{\mathrm{TRAJ},T}(\hat{\pi}) \lesssim O_\star(1)\, \mathbf{J}_{\mathrm{DEMO},T}(\hat{\pi}; \mathbb{P}_{\pi^\star}) + O_\star(T/\sigma_{\mathbf{u}}^4)\, \mathbf{J}_{\mathrm{DEMO},T}(\hat{\pi}; \mathbb{P}_{\pi^\star, \sigma_{\mathbf{u}}, \alpha})^2$. In particular, this would imply the impact of $T$ and $\sigma_{\mathbf{u}}$ vanish when $\mathbf{J}_{\mathrm{DEMO},T}(\hat{\pi}; \mathbb{P}_{\pi^\star, \sigma_{\mathbf{u}}, \alpha})$ is sufficiently small (i.e. when $n$ is large), e.g., $\mathbf{J}_{\mathrm{DEMO},T}(\hat{\pi}; \mathbb{P}_{\pi^\star, \sigma_{\mathbf{u}}, \alpha}) \lesssim O_\star(\sigma_{\mathbf{u}}^4/T)$ implies $\mathbf{J}_{\mathrm{TRAJ},T}(\hat{\pi}) \lesssim O_\star(1)\, \mathbf{J}_{\mathrm{DEMO},T}(\hat{\pi}; \mathbb{P}_{\pi^\star, \sigma_{\mathbf{u}}, \alpha})$. As a consequence, this reveals that we may only need "sufficiently many" noise-injected trajectories to ensure stable closed-loop behavior (see e.g. Figure 3, **center**) that scales *horizon-free*. We direct detailed derivations and discussion to Appendix E.6. To summarize the key takeaways of this section:

> **Key Findings**
>
> - Proposition 4.3 and Proposition 4.4 isolate that the key role of noise-injection is **ensuring the first-order policy error on the controllable subspace $\mathcal{R}_t^{\pi^\star}$ is detectable**.
> - Proposition 4.4 further shows **we only require supervision on the *excitable* subspace $\mathcal{R}_t^{\pi^\star}(\lambda)$** therein. In particular, we bypass stringent requirements of RL-theoretic coverage or control-theoretic PE.
> - **Simple white noise suffices to explore $\mathcal{R}_t^{\pi^\star}(\lambda)$**, as the most excitable (fastest compounding error) directions receive the most supervision.
> - **Imitating a mixture of clean and noise-injected trajectories** bypasses additive error scaling with $\sigma_{\mathbf{u}}$ (see Proposition 4.1). This further larger noise-levels implies $\sigma_{\mathbf{u}} > 0$ *can be beneficial*.

## 5 DISCUSSION AND LIMITATIONS

Our action-chunking guarantees rely on a structural assumption of $(\hat{\pi}, \hat{f}) \in \mathcal{P}$ being an EISS pair. We believe either explicitly enforcing this, e.g., via regularization (Sindhwani et al., 2018; Mehta et al., 2025) or hierarchy (Matni et al., 2024), or attaining it indirectly via implicit biases (Chi et al., 2023), are interesting directions of inquiry. We assume smoothness in Section 4, which is not strictly satisfied in some applications, such as in model-predictive control (Garcia et al., 1989). We remark our lower bound Proposition 4.1 depends on smoothness in $C_\pi$, which implies it is in some sense a fundamental aspect of noise-injection. However, we believe our results should extend to piece-wise notions (Block et al., 2023), and note ongoing research exploring *smoothing* for learning in dynamical systems (Suh et al., 2022; Pang et al., 2023; Pfrommer et al., 2024). In general, we leave a sharp characterization of the role of smoothness and control-theoretic quantities in IL as an open problem. We also note though our theory suggests isotropic noise injection suffices, this may not be desirable in some practical contexts, such as highly dexterous robotics. In light of our findings elucidating the precise role of local exploration, we leave designing robust practical recipes for perturbative data collection as future inquiry. Lastly, we leave investigating the marginal benefit of *iterative* interaction (Ross et al., 2011; Laskey et al., 2017; Kelly et al., 2019; Hu et al., 2025) as future work.

---

[6]Note applying Markov's inequality to the second term recovers Theorem 2.

## ACKNOWLEDGMENTS

TZ gratefully acknowledges a gift from AWS AI to Penn Engineering's ASSET Center for Trustworthy AI. TZ and NM are supported in part by NSF Award SLES-2331880, NSF CAREER award ECCS-2045834, NSF EECS-2231349, and AFOSR Award FA9550-24-1-0102. MS acknowledges support from a Google Robotics Award and Toyota Research Institute University 2.0 Fellowship.

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

## A  SUPPLEMENTARY FIGURES

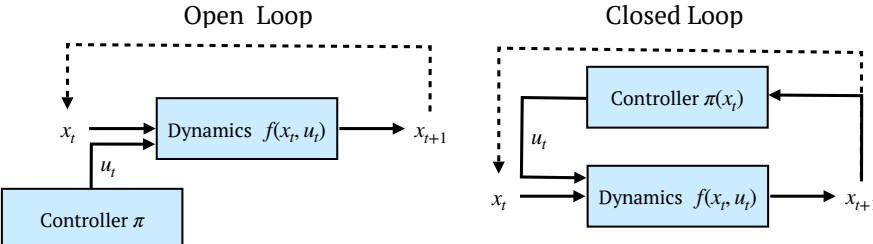

**Figure 4:** A comparison of open-loop control, where the policy generates actions without accessing the system state, and closed-loop control, where the policy's generated actions condition on the system state. While action-chunks are generated closed-loop, the actions within a chunk are executed "open-loop."

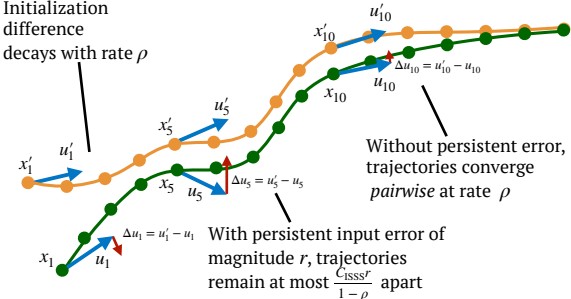

**Figure 5:** A visualization of EISS (Definition 2.1), which guarantees pairwise contraction of trajectories.

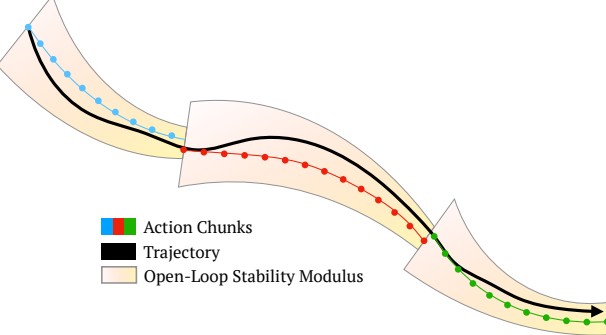

**Figure 6:** We visualize the stabilizing effect of using multiple action chunks (shown in different colors) when evaluating a "chunked" policy (with corresponding trajectory shown in black). As the open-loop dynamics on each chunk is stabilizing, this ensures closed-loop-EISS of the resulting learned policy over multiple chunks.

## B  ADDITIONAL DISCUSSION

**Extended Related Work.** Imitation learning from expert demonstrations has emerged as a dominant technique for learning performant models across applications such as: self-driving vehicles (Hussein et al., 2017; Bojarski et al., 2016; Bansal et al., 2018), visuomotor policies (Finn et al., 2017; Zhang et al., 2018), and large-scale robotic decision-making models (Zitkovich et al., 2023; Black et al., 2024). As such, the compounding error phenomenon is well-documented, dating back even to the introduction of IL (Pomerleau, 1988).

In discrete state-action settings, the seminal work in Ross & Bagnell (2010); Ross et al. (2011) propose an *iterative, interactive* procedure to collect examples of corrective data, seeing widespread adoption (Kelly et al., 2019; Sun et al., 2023). On the theoretical side, compounding errors appears more benign in discrete settings, with naive behavior cloning (BC) attaining a discrepancy between

Noise Injection on the Excitable Manifold

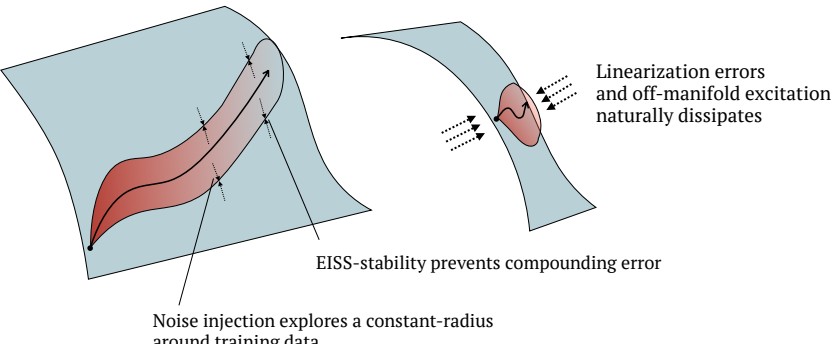

**Figure 7:** The effect of noise injection for controllable versus uncontrollable subspaces. We illustrate the key advantage of Proposition 4.3, namely, that noise injection occurs primarily in more excitable directions. By leveraging this mechanism, we are able to derive better error rates (Suboptimal Proposition 4.2 vs Proposition 4.3)

training and execution error at most quadratic in the horizon. Recent work by Foster et al. (2024) even demonstrates that modifying the loss may result in performance that has *no* adverse dependence on horizon. However, these works operate in settings ill-suited for continuous control, where the expert policy must be estimated in information-theoretic distances that are not feasible, e.g., even for deterministic policies in continuous action spaces.

Accordingly, prior work which applies IL to continuous control settings has involved more elaborate set-ups to enable stable performance. For example, recent advances in generative policies are typically paired with action-chunked execution (Practice 1), see e.g., (Chen et al., 2021; Shafiullah et al., 2022; Chi et al., 2023; Zhao & Grover, 2023; Liu et al., 2025). Other works have considered tools from robust control (Hertneck et al., 2018; Yin et al., 2021) and stability regularization (Sindhwani et al., 2018; Mehta et al., 2025) to promote stability around observed data. Lastly, various works have proposed different forms of data augmentation as a way to promote robustness to distribution shift, including iteratively shaped noise injection during expert demonstrations (Laskey et al., 2017), and noising observed states/actions (Ke et al., 2021; 2024; Block et al., 2024). Our proposed Practice 2 can be viewed as a distilled, non-iterative version of DART (Laskey et al., 2017).

A complementary line of work has attempted to understand the theoretical foundations of imitating in continuous settings. Tu et al. (2022) parameterize a scale of "incremental stability" (see Definition 2.1) and study its impact on the statistical generalization of IL. Pfrommer et al. (2022) proposes sufficient conditions for benign compounding errors in a similar setting. However, the resulting algorithms have exceedingly strong requirements, e.g., stability oracles or $\partial \text{input}/\partial \text{state}$ derivative sketching, respectively. Rounding off this line of work, Simchowitz et al. (2025) offers definitive evidence that exponential compounding errors cannot be avoided by altering the learning procedure, motivating the interventions we study.

**Supervised Learning Preliminaries.** Given a demonstration distribution over trajectories $\mathbb{P}_{\text{demo}}$, consider a sample $S_n$ of $n$ i.i.d. trajectories $(\mathbf{x}_t^{(i)}, \mathbf{u}_t^{(i)})_{1 \le t \le T, 1 \le i \le n}$ from $\mathbb{P}_{\text{demo}}$. The *empirical risk* of a candidate policy $\hat{\pi}$ over this sample is defined as:

$$\mathbf{J}_{\text{EMP},T}(\hat{\pi}; S_n) \triangleq \sum_{i=1}^{n} \sum_{t=1}^{T} \|\hat{\pi}(\mathbf{x}_{1:t}^{(i)}, \mathbf{u}_{1:t-1}^{(i)}, t) - \mathbf{u}_t^{(i)}\|^2, \tag{B.1}$$

where the form of $\hat{\pi}(\mathbf{x}_{1:t}^{(i)}, \mathbf{u}_{1:t-1}^{(i)}, t)$ depends on policy parameterization (e.g. Markovian or chunked (3.1)). Notably, $\mathbb{E}_{S_n}[\mathbf{J}_{\text{EMP},T}(\hat{\pi}; S_n)] = \mathbf{J}_{\text{DEMO},T}(\hat{\pi}; \mathbb{P}_{\text{demo}})$. Our work is independent of how $\hat{\pi}$ is derived, as we are concerned only with how the **on-expert error** (i.e. imitation generalization error) $\mathbf{J}_{\text{DEMO},T}(\hat{\pi}; \mathbb{P}_{\text{demo}})$ translates to closed-loop **trajectory error** $\mathbf{J}_{\text{TRAJ},T}(\hat{\pi})$. However, to conceptualize how $\mathbf{J}_{\text{DEMO},T}(\hat{\pi}; \mathbb{P}_{\text{demo}})$ scales in terms of $n$, we may consider the quintessential Empirical Risk Minimization (ERM) algorithm: $\hat{\pi} \in \arg\min_{\pi} \mathbf{J}_{\text{EMP},T}(\hat{\pi}; S_n)$; notably, this corresponds to the ob-

jective of vanilla **behavior cloning**. Since $\mathbf{J}_{\text{DEMO},T}(\hat{\pi}; \mathbb{P}_{\text{demo}})$ is simply the expected error over the same distribution $\mathbb{P}_{\text{demo}}$ that generated the data, one can apply standard supervised learning bounds for ERM, for example, $\mathbf{J}_{\text{DEMO},T}(\hat{\pi}; \mathbb{P}_{\text{demo}}) \lesssim n^{-1}$ corresponding to *parametric*/"fast" rates and $\mathbf{J}_{\text{DEMO},T}(\hat{\pi}; \mathbb{P}_{\text{demo}}) \lesssim n^{-\alpha}$, $\alpha \in (0, 1)$ corresponding to "non-parametric" scaling, see e.g. Bartlett & Mendelson (2002); Shalev-Shwartz & Ben-David (2014); Wainwright (2019) for standard references, and Tu et al. (2022); Pfrommer et al. (2022); Simchowitz et al. (2025) for discussion specific to imitation learning.

We further note that the above proxies for the scaling of $\mathbf{J}_{\text{DEMO},T}(\hat{\pi}; \mathbb{P}_{\text{demo}})$ ignore the axis of trajectory horizon $T$; for long trajectories that experience varying degrees of stationarity/ergodicity, $\mathbf{J}_{\text{DEMO},T}(\hat{\pi}; \mathbb{P}_{\text{demo}})$ plausibly also improves with $T$ despite the temporal dependence within a trajectory. Various learning-theoretic works in linear and nonlinear control demonstrate this formally under various dependence structures, see e.g. (Simchowitz et al., 2018; Ziemann & Tu, 2022; Ziemann et al., 2024) for discussion surrounding the related problem of system identification, and (Zhang et al., 2023) for specification to linear imitation learning. As these works only affect the theoretical scaling of $\mathbf{J}_{\text{DEMO},T}(\hat{\pi}; \mathbb{P}_{\text{demo}})$, our analysis is independent of these learning-theoretic discussions.

As a final remark, we mention that action-chunking by definition induces a different policy class $\Pi_{\text{chunk},\ell}$ than the class of "base" Markovian policies $\Pi$; for example, given an input state, the former outputs $\ell$ actions rather than one. We surmise the additional statistical burden of predicting with a chunked policy class is negligible especially considering it mitigates exponential compounding trajectory error. Firstly, Theorem 1 implies the requisite chunk length $\ell$ is small (*logarithmic* in system parameters), so even if $\Pi_{\text{chunk},\ell}$ is treated naively as an $\ell$-step predictor class $\mathcal{X} \to \mathcal{U}^{\otimes \ell}$, the difference from inflating the output dimension between $\Pi_{\text{chunk},\ell}$ and $\Pi$ is similarly small. Furthermore, $\Pi_{\text{chunk},\ell}$ is not a naive $\ell$-step predictor (though it often suffices to implement action-chunking as such in practice; see Appendix G), as it constrains the output to be rolled out through the candidate dynamics. As such, this constraint intuitively should reduce the asymptotic variance of $\Pi_{\text{chunk},\ell}$ compared to direct $\ell$-step prediction $\mathcal{X} \to \mathcal{U}^{\otimes \ell}$, though establishing this concretely remains a relatively unexplored problem; see Somalwar et al. (2025) for initial studies on the related problem of system-identification in linear systems.

## C  CONTROL-THEORY PRELIMINARIES

Before proceeding to the technical statements and proofs, we provide here a primer to some fundamental intuitions, objects, and motivations in control theory.

We start with the most basic model and task: a linear time-invariant system regulating to the origin $\mathbf{0}$. A linear system obeys the transition dynamics:

$$\mathbf{x}_{t+1} = \mathbf{A}\mathbf{x}_t + \mathbf{B}\mathbf{u}_t.$$

Here, we can already introduce some of the common terms used in control. When $\mathbf{B} = \mathbf{0}$, the system is said to evolve **autonomously**: $\mathbf{x}_{t+1} = \mathbf{A}\mathbf{x}_t$. A fundamental fact about autonomous (discrete-time) linear systems is that they are stable if and only if $\rho(\mathbf{A}) < 1$, where $\rho(\cdot)$ denotes the spectral radius. The cases $\rho(\mathbf{A}) > 1$ and $\rho(\mathbf{A}) = 1$ are referred as (exponentially) unstable and marginally unstable, respectively.

In many cases, the autonomous evolution is unstable, and requires a controller to, e.g., stabilize to the origin. **Open-loop control** generally refers to when the control input at time $t$ does not depend on the state at that time, e.g.,

$$\mathbf{x}_{t+1} = \mathbf{A}\mathbf{x}_t + \mathbf{B}\mathbf{u}_t, \quad \text{given } \mathbf{u}_1, \ldots, \mathbf{u}_t \text{ predetermined.}$$

The general understanding in controls engineering is that prolonged open-loop control is undesirable[7], as small model mismatches or unseen disturbances can shift the optimal control sequence away from the predetermined one, and may even render the system unstable. Therefore, stabilization is often performed via **closed-loop control**, which yields control inputs that condition on the observed state(s). The canonical example is a (linear) **feedback controller**:

$$\mathbf{x}_{t+1} = \mathbf{A}\mathbf{x}_t + \mathbf{B}\mathbf{u}_t, \quad \mathbf{u}_t = \pi(\mathbf{x}_t) \triangleq \mathbf{K}\mathbf{x}_t.$$

---

[7]Which is why action-chunking, by intentionally executing chunks of inputs in open-loop, may be a somewhat surprising practice to a controls engineer.

We note that the system evolution under a feedback controller can be equivalently written as a system *autonomously* evolving with dynamics $\mathbf{x}_{t+1} = (\mathbf{A} + \mathbf{BK})\mathbf{x}_t$. Just as $\rho(\mathbf{A}) < 1$ determines the exponential stability of the autonomous system to $\mathbf{0}$, we say a controller stabilizes the system, or alternatively, renders the system **closed-loop stable** if $\rho(\mathbf{A} + \mathbf{BK}) < 1$.

**Remark C.1** (Open- vs. Closed-loop Stable). We remark that the above discussion leads to the general terming of a system being **open-loop** or **closed-loop** stable. In particular, **open-loop stability** generally refers to a system satisfying a given definition of stability without the need of a feedback controller (e.g., $\rho(\mathbf{A}) < 1$), where **closed-loop stability** refers to a system achieving a given definition of stability in closed-loop with a feedback controller (e.g., $\rho(\mathbf{A} + \mathbf{BK}) < 1$).

With the definition of feedback control and (linear) stability in hand, we consider notions of the steerability of a system.

**Definition C.1** (Controllability and Stabilizability (Kailath, 1980)). A linear dynamics pair $(\mathbf{A}, \mathbf{B}) \in \mathbb{R}^{d_x \times d_x} \times \mathbb{R}^{d_x \times d_u}$ is **controllable** if and only if the matrix $\mathcal{C} \triangleq \begin{bmatrix} \mathbf{B} & \mathbf{AB} & \cdots & \mathbf{A}^{d_x-1}\mathbf{B} \end{bmatrix}$ has rank $d_x$. This is equivalent to saying, given any initial state $\mathbf{x}_1$ and goal state $\mathbf{x}_g$, there exists a sequence of inputs $\{\mathbf{u}_1, \cdots, \mathbf{u}_{d_x}\}$ such that executing them steers the state to $\mathbf{x}_{d_x+1} = \mathbf{x}_g$, i.e., *every state is eventually reachable*.

A dynamics pair $(\mathbf{A}, \mathbf{B})$ is **stabilizable** if for any eigenvalue of $\mathbf{A}$ with $|\lambda| \geq 1$, we have that $\begin{bmatrix} \mathbf{A} - \lambda\mathbf{I} & \mathbf{B} \end{bmatrix}$ is full row-rank. This is equivalent to: there exists $\mathbf{K}$ such that $\rho(\mathbf{A} + \mathbf{BK}) < 1$. In other words, a system is stabilizable if all uncontrollable modes are autonomously stable.

Stabilizability is the minimal condition under which stable closed-loop control is possible, since any uncontrollable unstable mode is impossible to stabilize. Controllability is somewhat stronger, saying that (barring unmodeled disturbances) any state can be achieved under an appropriate control sequence — one of the key technical innovations in our ensuing analysis is bypassing relying on (linearized) controllability, see Appendix E.4. Both controllability and stabilizability are binary conditions, and do not describe, e.g., which directions are "more" or "less" controllable. This motivates the controllability Gramian.

**Definition C.2** ((Time-invariant) Controllability Gramian). Given dynamics pair $(\mathbf{A}, \mathbf{B})$, the time-$t$ controllability Gramian is given by:

$$\mathbf{W}_{1:t}^{\mathbf{u}} \triangleq \sum_{s=1}^{t-1} \mathbf{A}^{s-1}\mathbf{BB}^{\top}(\mathbf{A}^{s-1})^{\top}.$$

The controllability Gramian admits a few equivalent interpretations. In particular, the controllability Gramian is the covariance matrix of the state $\mathbf{x}_t$ under zero-mean, identity-covariance inputs, which exposes the state directions that are relatively easier or harder to excite, corresponding to larger or smaller eigendirections of $\mathbf{W}_{1:t}^{\mathbf{u}}$. We also note that, particularly relevant to the IL setting, we may similarly define the controllability Gramian of a closed-loop system in feedback with a given controller $\mathbf{K}$:

$$\mathbf{W}_{1:t}^{\mathbf{u}} \triangleq \sum_{s=1}^{t-1} \mathbf{A}_{\mathrm{cl}}^{s-1}\mathbf{BB}^{\top}(\mathbf{A}_{\mathrm{cl}}^{s-1})^{\top}, \quad \mathbf{A}_{\mathrm{cl}} \triangleq \mathbf{A} + \mathbf{BK}.$$

We note that despite the storied history of linear control, our setting contends with nonlinear dynamics and policies. In particular, the dynamics is now governed by a possibly nonlinear transition

$$\mathbf{x}_{t+1} = f(\mathbf{x}_t, \mathbf{u}_t).$$

To build up to the **incremental input-to-state stability** we consider (see Definition 2.1), we start first with the same task as in the above linear case, where we want to stabilize to the origin. There, a well-known notion of nonlinear stability is **input-to-state stability**.

**Definition C.3** (Input-to-State Stability (Khalil, 2002; Sontag, 2013)). A system $\mathbf{x}_{t+1} = f(\mathbf{x}_t, \mathbf{u}_t)$ is **input-to-state stable** if there exists class-$\mathcal{KL}$ and class-$\mathcal{K}$ functions $\beta(z, t)$ and $\gamma(z)$ [8] such that for any initial state $\mathbf{x}_1$ and control sequence $\{\mathbf{u}_s\}_{s \geq 1}$, we have for all $t \geq 1$:

$$\|\mathbf{x}_t\| \leq \beta(\|\mathbf{x}_1\|, t-1) + \gamma\left(\max_{k \leq t-1} \|\mathbf{u}_k\|\right).$$

---

[8] A function $\gamma(z)$ is class $\mathcal{K}$ if it is continuous, increasing in $z$, and satisfies $\gamma(0) = 0$, and a function $\beta(z, t)$ is class $\mathcal{KL}$ if it is continuous, $\beta(\cdot, t)$ is class $\mathcal{K}$ for each fixed $t$, and $\beta(z, \cdot)$ is decreasing for each fixed $z$.

Input-to-state stability states that the dependence of a future state's magnitude on the initial state decays at some rate determined by $\beta$, and bounded inputs have bounded effect on the state across time. This notion of stability is very general, such as through choices of norm $\|\cdot\|$, and moduli $\beta$, $\gamma$, and further admits many extensions, e.g. locality. To reduce the distracting technical baggage to a minimum, we do not discuss the many extensions, and make the stability quantitative via *exponential* stability, where $\beta(z,t) \triangleq C_{\text{ISS}}\rho^t z$, and $\gamma$ is a exponential convolution of past $\|\mathbf{u}_k\|$ across time, see Definition 2.1.

However, input-to-state stability invariably considers tasks regulating the state to a fixed point. In general, this is not the case in imitation learning. Therefore, to capture that the stability we care about is not necessarily to a prescribed equilibrium point, but to a *trajectory*, we consider **incremental stability** (Angeli, 2002; Tran et al., 2016), where "incremental" refers to the fact that rather than contending with the state itself, we consider the state/trajectory difference. The definition of **incremental input-to-state stability** naturally follows.

The above concepts serve as the core background behind the control-theoretic terminology used in our presentation and analysis. However, beyond what is presented here, many technical complications arise in our nonlinear incremental setting. For example, the "incremental" part (i.e. trajectory-tracking) means many intuitions and specialized tools for the canonical task of stabilizing to a prescribed fixed point no longer hold. Further, even for the control task of regulating a *time-invariant* nonlinear system $f(\mathbf{x}, \mathbf{u})$ to $\mathbf{0}$, iteratively linearizing the trajectory yields a *time-varying* linearized system. Furthermore, even if the linearized controllability Gramian $\mathbf{W}_{1:t}^{\mathbf{u}}$ is rank-deficient, this does not mean directions lying in its null-space are unexcitable/unreachable due to the contribution of the nonlinear component of the dynamics. Contending with these complications arising from the generality of our setting—which is core to presenting a sufficiently general descriptive framework—is the subject of the ensuing theoretical analysis.

## D    PROOFS AND ADDITIONAL DETAILS FOR SECTION 3

We first introduce some additional definitions:

**Definition D.1** (Additional Error Definitions)**.**  Given $p \geq 1$, define the $p$-th power errors:

$$\mathbf{J}_{\text{TRAJ},p,T}(\hat{\pi}) \triangleq \mathbb{E}_{\hat{\pi},\pi^\star}\left[\sum_{t=1}^T \min\left\{1, \|\mathbf{x}_t^{\hat{\pi}} - \mathbf{x}_t^{\pi^\star}\|^p + \|\mathbf{u}_t^{\hat{\pi}} - \mathbf{u}_t^{\pi^\star}\|^p\right\}\right]$$

$$\mathbf{J}_{\text{DEMO},p,T}(\hat{\pi}; \mathbb{P}_{\text{demo}}) \triangleq \mathbb{E}_{\mathbb{P}_{\text{demo}}}\left[\sum_{t=1}^T \|\hat{\pi}(\mathbf{x}_{1:t}, \mathbf{u}_{1:t-1}, t) - \mathbf{u}_t^{(i)}\|^p\right].$$

Note that $\mathbf{J}_{\text{TRAJ},2,T} \equiv \mathbf{J}_{\text{TRAJ},T}$, $\mathbf{J}_{\text{DEMO},p,T} \equiv \mathbf{J}_{\text{DEMO},T}$. We further define the trajectory *state* error:

$$\mathbf{J}_{\text{TRAJ},p,T}^{\mathbf{x}}(\hat{\pi}) \triangleq \mathbb{E}_{\hat{\pi},\pi^\star}\left[\sum_{t=1}^T \min\left\{1, \|\mathbf{x}_t^{\hat{\pi}} - \mathbf{x}_t^{\pi^\star}\|^p\right\}\right].$$

We now state some elementary results.

**Lemma D.1.** *Assume $\hat{\pi}$ is a Markovian, $L_\pi$-Lipschitz policy. Then:*

$$\mathbf{J}_{\text{TRAJ},p,T}(\hat{\pi}) \leq (1 + (2L_\pi)^p)\,\mathbf{J}_{\text{TRAJ},p,T}^{\mathbf{x}}(\hat{\pi}) + 2^p\mathbf{J}_{\text{DEMO},p,T}(\hat{\pi}; \mathbb{P}_{\pi^\star}).$$

*Proof.* Following the definition of $\mathbf{J}_{\text{TRAJ},p,T}$, we may add and subtract $\|\mathbf{u}_t^{\hat{\pi}} - \hat{\pi}(\mathbf{x}_t^{\pi^\star}) + \hat{\pi}(\mathbf{x}_t^{\pi^\star}) - \mathbf{u}_t^{\pi^\star}\|^p$ and apply convexity of $\|\cdot\|^p$ to yield:

$$\mathbf{J}_{\text{TRAJ},T}(\hat{\pi}) \leq \mathbb{E}_{\hat{\pi},\pi^\star}\left[\sum_{t=1}^T \min\left\{1, \|\mathbf{x}_t^{\hat{\pi}} - \mathbf{x}_t^{\pi^\star}\|^p + 2^p\|\hat{\pi}(\mathbf{x}_t^{\hat{\pi}}) - \hat{\pi}(\mathbf{x}_t^{\pi^\star})\|^p + 2^p\|\hat{\pi}(\mathbf{x}_t^{\pi^\star}) - \pi^\star(\mathbf{x}_t^{\pi^\star})\|^p\right\}\right].$$

Applying Lipschitzness of $\hat{\pi}$ to the second term, and observing the last term is precisely the summand in $\mathbf{J}_{\text{DEMO},p,T}$ completes the proof. $\qquad\square$

**Lemma D.2** (Kantorovich-Rubinstein). *Define the norm on $\mathbb{R}_x^d \times \mathbb{R}_u^d$: $\|(\mathbf{x}, \mathbf{u})\|_{\mathbf{x},\mathbf{u}} \triangleq \|\mathbf{x}\| + \|\mathbf{u}\|$. Then, define the class of cost functions $\mathsf{c}(\mathbf{x}, \mathbf{u}) \in \mathcal{C}_{\mathrm{Lip}(1)}$ that is 1-Lipschitz in $\|\cdot\|_{\mathbf{x},\mathbf{u}}$. Then, we have the following:*

$$\mathbf{J}_{\mathrm{TRAJ},1,T}(\hat{\pi}) \leq \sup_{\mathsf{c}_{1:T} \subset \mathcal{C}_{\mathrm{Lip}(1)}} \mathbf{J}_{\mathrm{COST},T}(\hat{\pi}; \mathsf{c}_{1:T}).$$

The above is a straightforward application of Kantorovich-Rubinstein strong duality (Villani et al., 2009) by pulling out a conditional expectation over $\mathbf{x}_1^{\pi^\star}, \mathbf{x}_1^{\hat{\pi}} = \mathbf{x}_1$ on both sides. The inequality then follows due to the clipping at 1 in the definition of $\mathbf{J}_{\mathrm{TRAJ},1,T}$.

We will often use the following bound on triangular Toeplitz matrices.

**Lemma D.3.** *Given $\rho \in [0, 1)$, define the matrix:*

$$\mathbf{A}(\rho) = \begin{bmatrix} 1 & 0 & 0 & \cdots & 0 \\ \rho & 1 & 0 & \cdots & 0 \\ \rho^2 & \rho & 1 & \cdots & 0 \\ \vdots & & & & \\ \rho^{d-1} & \rho^{d-2} & \rho^{d-3} & \cdots & 1 \end{bmatrix} \in \mathbb{R}^{d \times d}.$$

*Given $1 \leq p < \infty$, the following bound holds on the induced $\ell^p \to \ell^p$ operator norm of $\mathbf{A}(\rho)$:*

$$\|\mathbf{A}(\rho)\|_{p \to p} \leq \sum_{s=0}^{d-1} \rho^s \leq \frac{1}{1-\rho}.$$

*Proof.* We may prove this straightforwardly from an application of the Riesz-Thorin interpolation theorem (Stein & Shakarchi, 2011), which states that fixing $\mathbf{A}(\rho)$, the mapping $1/p \mapsto \|\mathbf{A}(\rho)\|_{p \to p}$ is log-convex for $p \in [1, \infty]$. In particular, by taking the convex combination $1/p = 1 \cdot (1/p) + 0 \cdot (1 - 1/p)$, we find:

$$\log \|\mathbf{A}(\rho)\|_{p \to p} \leq (1/p) \log \|\mathbf{A}(\rho)\|_{1 \to 1} + (1 - 1/p) \log \|\mathbf{A}(\rho)\|_{\infty \to \infty}$$

$$\|\mathbf{A}(\rho)\|_{p \to p} \leq \|\mathbf{A}(\rho)\|_{1 \to 1}^{1/p} \|\mathbf{A}(\rho)\|_{\infty \to \infty}^{1 - 1/p}.$$

We then utilize the basic fact that $\|\mathbf{A}(\rho)\|_{1 \to 1}$ and $\|\mathbf{A}(\rho)\|_{\infty \to \infty}$ correspond to the maximum column and row sum of $\mathbf{A}(\rho)$, respectively, which completes the result. $\square$

We may now state the detailed version of Proposition 3.1.

**Proposition D.4** (Full ver. of Proposition 3.1). *Let Assumption 3.1 hold. Let $(\hat{\pi}, \hat{f})$ be a policy-dynamics pair that is $(C_{\mathrm{ISS}}, \rho)$-EISS, and consider the corresponding chunked policy $\tilde{\pi} = \mathsf{chunked}(\hat{\pi}, \hat{f}, \ell)$. Then the closed-loop system the chunked policy induces on the true dynamics $(\tilde{\pi}, f)$ is $(\tilde{C}, \rho^{1-a})$-EISS, where $a \in (0, 1)$ and $\tilde{C} = a^{-1} \log(1/\rho)^{-1} \cdot \mathrm{poly}(L_\pi, C_{\mathrm{ISS}})$, as long as the chunk length is sufficiently long: $\ell \gtrsim a^{-1} \log(1/\rho)^{-1} \cdot \log(\mathrm{poly}(L_\pi, C_{\mathrm{ISS}}, 1/a))$.*

*Proof of Proposition 3.1.* Let us define the chunk-indexing shorthand $t_k \triangleq (k-1)\ell + 1$, such that $t_1 = 1$. Toward establishing EISS of the closed-loop chunked system, we want to show for a sequence of input perturbations $\{\mathbf{u}_t\}_{t \geq 1}$ and two trajectories $\{\mathbf{x}_t^{\tilde{\pi}}\}_{t \geq 1}$, $\{\overline{\mathbf{x}}_t^{\tilde{\pi}}\}_{t \geq 1}$ evolving as:

$$\mathbf{x}_{t+1}^{\tilde{\pi}} = f^{\tilde{\pi}}(\mathbf{x}_t^{\tilde{\pi}}, \mathbf{0}), \quad \mathbf{x}_1^{\tilde{\pi}} = \mathbf{x}_1$$
$$\overline{\mathbf{x}}_{t+1}^{\tilde{\pi}} = f^{\tilde{\pi}}(\overline{\mathbf{x}}_t^{\tilde{\pi}}, \mathbf{u}_t), \quad \overline{\mathbf{x}}_1^{\tilde{\pi}} = \overline{\mathbf{x}}_1,$$

there exist some constants $C \geq 1$, $\rho \in (0, 1)$ such that:

$$\|\mathbf{x}_T^{\tilde{\pi}} - \overline{\mathbf{x}}_T^{\tilde{\pi}}\| \leq C\rho^{T-1}\|\mathbf{x}_1 - \overline{\mathbf{x}}_1\| + C \sum_{s=1}^{T-1} \rho^{T-1-s}\|\mathbf{u}_s\|.$$

To do so, we prove the following "contractivity" result going between chunks.

**Lemma D.5.** *Fix some $k \geq 1$. Recall the true dynamics $f$ is $(C_{\mathrm{ISS}}, \rho)$-EISS. Then, the following holds:*

$$\|\mathbf{x}_{t_{k+1}}^{\tilde{\pi}} - \overline{\mathbf{x}}_{t_{k+1}}^{\tilde{\pi}}\| \leq \boldsymbol{\rho}^\ell \|\mathbf{x}_{t_k}^{\tilde{\pi}} - \overline{\mathbf{x}}_{t_k}^{\tilde{\pi}}\| + C_{\mathrm{ISS}} \sum_{s=0}^{\ell-1} \rho^{\ell-1-s} \|\mathbf{u}_{t_k+s}\|,$$

*where $\boldsymbol{\rho} \triangleq \rho^{1-a}$, as long as $\ell > a^{-1}\mathrm{polylog}(1 + L_\pi, C_{\mathrm{ISS}}, \log(1/\rho), a^{-1})$. As a corollary, setting $\overline{C} = \frac{(1+L_\pi)C_{\mathrm{ISS}}^2}{3a\rho \log(1/\rho)}$, for $1 \leq h \leq \ell$, we have:*

$$\|\mathbf{x}_{t_k+h}^{\tilde{\pi}} - \overline{\mathbf{x}}_{t_k+h}^{\tilde{\pi}}\| \leq \overline{C} \boldsymbol{\rho}^h \|\mathbf{x}_{t_k}^{\tilde{\pi}} - \overline{\mathbf{x}}_{t_k}^{\tilde{\pi}}\| + C_{\mathrm{ISS}} \sum_{s=0}^{h-1} \rho^{h-1-s} \|\mathbf{u}_{t_k+s}\|.$$

*Proof of Lemma D.5.* Applying $(C_{\mathrm{ISS}}, \rho)$-EISS of the true dynamics $f$, we have

$$\|\mathbf{x}_{t_{k+1}}^{\tilde{\pi}} - \overline{\mathbf{x}}_{t_{k+1}}^{\tilde{\pi}}\| \leq C_{\mathrm{ISS}}\rho^\ell \|\mathbf{x}_{t_k}^{\tilde{\pi}} - \overline{\mathbf{x}}_{t_k}^{\tilde{\pi}}\| + C_{\mathrm{ISS}} \sum_{s=0}^{\ell-1} \rho^{\ell-1-s} \|\mathbf{u}_{t_k+s}^{\tilde{\pi}} - \overline{\mathbf{u}}_{t_k+s}^{\tilde{\pi}} - \mathbf{u}_{t_k+s}\|$$

$$\leq C_{\mathrm{ISS}}\rho^\ell \|\mathbf{x}_{t_k}^{\tilde{\pi}} - \overline{\mathbf{x}}_{t_k}^{\tilde{\pi}}\| + C_{\mathrm{ISS}} \sum_{s=0}^{\ell-1} \rho^{\ell-1-s} \|\mathbf{u}_{t_k+s}^{\tilde{\pi}} - \overline{\mathbf{u}}_{t_k+s}^{\tilde{\pi}}\| + C_{\mathrm{ISS}} \sum_{s=0}^{\ell-1} \rho^{\ell-1-s} \|\mathbf{u}_{t_k+s}\|.$$

where $\mathbf{u}_{t_k+s}^{\tilde{\pi}}$ and $\overline{\mathbf{u}}_{t_k+s}^{\tilde{\pi}}$ are the $s$-th actions outputted by the chunked policy $\tilde{\pi}$ conditioned on $\mathbf{x}_{t_k}^{\tilde{\pi}}$ and $\overline{\mathbf{x}}_{t_k}^{\tilde{\pi}}$, respectively. We consider the "simulated" dynamical system that generates $\mathbf{u}^{\tilde{\pi}}, \overline{\mathbf{u}}^{\tilde{\pi}}$:

$$\mathbf{x}_{s+1} = \hat{f}^{\hat{\pi}}(\mathbf{x}_s, \mathbf{0}) = \hat{f}(\mathbf{x}_s, \hat{\pi}(\mathbf{x}_s)), \ s = 0, \ldots, \ell - 1$$

$$\mathbf{u}_{t_k+s}^{\tilde{\pi}} \triangleq \hat{\pi}(\mathbf{x}_s), \ \mathbf{x}_0 = \mathbf{x}_{t_k}^{\tilde{\pi}},$$

$$\tilde{\mathbf{x}}_{s+1} = \hat{f}^{\hat{\pi}}(\tilde{\mathbf{x}}_s, \mathbf{0}) = \hat{f}(\tilde{\mathbf{x}}_s, \hat{\pi}(\tilde{\mathbf{x}}_s)), \ s = 0, \ldots, \ell - 1$$

$$\overline{\mathbf{u}}_{t_k+s}^{\tilde{\pi}} \triangleq \hat{\pi}(\tilde{\mathbf{x}}_s), \ \tilde{\mathbf{x}}_0 = \overline{\mathbf{x}}_{t_k}^{\tilde{\pi}}.$$

Crucially, we observe that $(\hat{\pi}, \hat{f})$ is $(C_{\mathrm{ISS}}, \rho)$-EISS, and thus:

$$\|\mathbf{x}_s - \tilde{\mathbf{x}}_s\| \leq C_{\mathrm{ISS}}\rho^s \|\mathbf{x}_0 - \tilde{\mathbf{x}}_0\| = C_{\mathrm{ISS}}\rho^s \|\mathbf{x}_{t_k}^{\tilde{\pi}} - \overline{\mathbf{x}}_{t_k}^{\tilde{\pi}}\|.$$

Therefore, by the $L_\pi$-Lipschitzness of $\hat{\pi}$, we have:

$$\|\mathbf{u}_{t_k+s}^{\tilde{\pi}} - \overline{\mathbf{u}}_{t_k+s}^{\tilde{\pi}}\| \leq L_\pi \|\mathbf{x}_s - \tilde{\mathbf{x}}_s\| \leq L_\pi C_{\mathrm{ISS}}\rho^s \|\mathbf{x}_{t_k}^{\tilde{\pi}} - \overline{\mathbf{x}}_{t_k}^{\tilde{\pi}}\|.$$

Plugging this back above, we get:

$$\|\mathbf{x}_{t_{k+1}}^{\tilde{\pi}} - \overline{\mathbf{x}}_{t_{k+1}}^{\tilde{\pi}}\| \leq C_{\mathrm{ISS}}\rho^\ell \|\mathbf{x}_{t_k}^{\tilde{\pi}} - \overline{\mathbf{x}}_{t_k}^{\tilde{\pi}}\| + L_\pi C_{\mathrm{ISS}} \cdot C_{\mathrm{ISS}} \sum_{s=0}^{\ell-1} \rho^{\ell-1-s}\rho^s \|\mathbf{x}_{t_k}^{\tilde{\pi}} - \overline{\mathbf{x}}_{t_k}^{\tilde{\pi}}\|$$

$$+ C_{\mathrm{ISS}} \sum_{s=0}^{\ell-1} \rho^{\ell-1-s} \|\mathbf{u}_{t_k+s}\|$$

$$\leq C_{\mathrm{ISS}}\rho^\ell \|\mathbf{x}_{t_k}^{\tilde{\pi}} - \overline{\mathbf{x}}_{t_k}^{\tilde{\pi}}\| + L_\pi C_{\mathrm{ISS}}^2 \ell\rho^{\ell-1} \|\mathbf{x}_{t_k}^{\tilde{\pi}} - \overline{\mathbf{x}}_{t_k}^{\tilde{\pi}}\| + C_{\mathrm{ISS}} \sum_{s=0}^{\ell-1} \rho^{\ell-1-s} \|\mathbf{u}_{t_k+s}\|$$

$$\leq (1 + L_\pi) C_{\mathrm{ISS}}^2 \ell\rho^{\ell-1} \|\mathbf{x}_{t_k}^{\tilde{\pi}} - \overline{\mathbf{x}}_{t_k}^{\tilde{\pi}}\| + C_{\mathrm{ISS}} \sum_{s=0}^{\ell-1} \rho^{\ell-1-s} \|\mathbf{u}_{t_k+s}\|.$$

We solve for the requisite chunk-length by solving: $(1 + L_\pi)C_{\mathrm{ISS}}^2 \ell\rho^{\ell-1} \leq \boldsymbol{\rho}^\ell$, where $\boldsymbol{\rho} = \rho^{1-a}$. Rearranging, this amounts to $\ell \in \mathbb{N}$ satisfying:

$$a\ell \geq 1 + \frac{\log\left((1 + L_\pi)C_{\mathrm{ISS}}^2 \ell\right)}{\log(1/\rho)}.$$

To remove the $\ell$ dependence on the right-hand side, we use the following elementary result.

**Lemma D.6** (Cf. Simchowitz et al. (2018, Lemma A.4)). *Given* $\alpha \geq 1$*, for any* $\ell \in \mathbb{N}$*,* $\ell \geq \alpha \log(\ell)$ *as soon as* $\ell \geq 2\alpha \log(4\alpha)$*.*

We observe the above result holds if we add any term that does not depend on $\ell$ on the right-side of both inequalities. Applying it to the above, since $\log(1/\rho) \geq \log(e) = 1$, setting $\alpha = (a \log(1/\rho))^{-1}$, we have that

$$\ell \geq a^{-1} + \frac{\log\left((1 + L_\pi)C_{\text{ISS}}^2\right)}{a \log(1/\rho)} + \frac{4 \log(a \log(1/\rho))}{a \log(1/\rho)},$$

implies $a\ell \geq 1 + \frac{\log\left((1+L_\pi)C_{\text{ISS}}^2 \ell\right)}{\log(1/\rho)}$, which in turn implies $(1 + L_\pi)C_{\text{ISS}}^2 \ell \rho^{\ell-1} \leq \boldsymbol{\rho}^\ell$ as required. For the corollary, we observe that the maximum value attained by $r^\star \triangleq \max_{h \in \mathbb{N}}(1 + L_\pi)C_{\text{ISS}}^2 h \rho^{h-1}/\boldsymbol{\rho}^h = (1 + L_\pi)C_{\text{ISS}}^2 h \rho^{ah-1}$ is upper bounded by $\frac{(1+L_\pi)C_{\text{ISS}}^2}{3a\rho \log(1/\rho)}$, completing the result.

$\square$

Toward bounding $\|\mathbf{x}_T^{\tilde{\pi}} - \overline{\mathbf{x}}_T^{\tilde{\pi}}\|$, we define the number of full chunks traversed $K - 1 = \lfloor(T - 1)/\ell\rfloor$, and the remaining timesteps $h = T - (K-1)\ell - 1$. Further define the shorthands $\mathbf{U}_k = C_{\text{ISS}} \sum_{s=0}^{\ell-1} \rho^{\ell-1-s}\|\mathbf{u}_{t_k+s}\|$ for $k \in [K]$, and $\mathbf{U}_{K+1} = C_{\text{ISS}} \sum_{s=0}^{h-1} \rho^{h-1-s}\|\mathbf{u}_{t_k+s}\|$. Then, for $\ell$ satisfying Lemma D.5, we use Lemma D.5 to iteratively peel:

$$\|\mathbf{x}_T^{\tilde{\pi}} - \overline{\mathbf{x}}_T^{\tilde{\pi}}\| \leq C_{\text{ISS}}\rho^h\|\mathbf{x}_{t_K}^{\tilde{\pi}} - \overline{\mathbf{x}}_{t_K}^{\tilde{\pi}}\| + C_{\text{ISS}} \sum_{s=0}^{h-1} \rho^{h-1-s}\|\mathbf{u}_{t_K+s}^{\tilde{\pi}} - \overline{\mathbf{u}}_{t_K+s}^{\tilde{\pi}} - \mathbf{u}_{t_K+s}\|$$

$$\leq \overline{C}\boldsymbol{\rho}^h\|\mathbf{x}_{t_K}^{\tilde{\pi}} - \overline{\mathbf{x}}_{t_K}^{\tilde{\pi}}\| + \mathbf{U}_{K+1}$$

$$\leq \overline{C}\boldsymbol{\rho}^{\ell+h}\|\mathbf{x}_{t_{K-1}}^{\tilde{\pi}} - \overline{\mathbf{x}}_{t_{K-1}}^{\tilde{\pi}}\| + \overline{C}\boldsymbol{\rho}^\ell \mathbf{U}_K + \mathbf{U}_{K+1}$$

$$\vdots$$

$$\leq \overline{C}\boldsymbol{\rho}^{T-1}\|\mathbf{x}_1 - \mathbf{x}_1'\| + \overline{C} \sum_{k=1}^{K+1} \boldsymbol{\rho}^{(k-1)\ell}\mathbf{U}_k$$

$$\leq \overline{C}\boldsymbol{\rho}^{T-1}\|\mathbf{x}_1 - \mathbf{x}_1'\| + \overline{C}C_{\text{ISS}} \sum_{s=1}^{T-1} \boldsymbol{\rho}^{T-1-s}\|\mathbf{u}_s\|.$$

This establishes that $(\tilde{\pi}, f)$ is $(\overline{C}C_{\text{ISS}}, \rho^{1-a})$, given the chunk length satisfies $\ell > a^{-1}\log(\text{poly}(1 + L_\pi, C_{\text{ISS}}, \log(1/\rho), a^{-1}))$, and leveraging $\rho \geq 1/e$, we complete the result.

$\square$

Having established that the chunked policy on the true dynamics $(\tilde{\pi}, f)$ is $(\tilde{C}, \hat{\rho}^{1-a})$ stable, we want to show this controls compounding errors when $\tilde{\pi}$ achieves low on-expert error to an expert policy $\pi^\star$. This is a straightforward application of EISS. In particular, by treating the expert inputs as perturbations to a closed-loop system induced by $(\tilde{\pi}, f)$, we may relate $\mathbf{J}_{\text{TRAJ},1,T}$ to $\mathbf{J}_{\text{DEMO},1,T}$.

**Proposition D.7** (Full ver. of Proposition 3.2). *Let Assumption 3.1 hold. Let* $\tilde{\pi} = \text{chunked}(\hat{\pi}, \hat{f}, \ell) \in \Pi_{\text{chunk},\ell}$*, and assume* $(\hat{\pi}, \hat{f})$*,* $(\tilde{\pi}, f)$ *are* $(\tilde{C}, \tilde{\rho})$*-EISS. Then, the following bound holds:*

$$\mathbf{J}_{\text{TRAJ},p,T}^{\mathbf{x}}(\tilde{\pi}) \leq \left(\frac{\tilde{C}}{1 - \tilde{\rho}}\right)^p \mathbf{J}_{\text{DEMO},p,T}(\tilde{\pi}; \mathbb{P}_{\pi^\star}).$$

*We have subsequently:*

$$\mathbf{J}_{\text{TRAJ},p,T}(\hat{\pi}) \leq \text{poly}^p\left(L_\pi, \tilde{C}, \frac{1}{1-\tilde{\rho}}\right) \mathbf{J}_{\text{DEMO},p,T}(\hat{\pi}; \mathbb{P}_{\pi^\star}).$$

*Proof.* Given $\mathbf{x}_1^{\pi^\star} = \mathbf{x}_1^{\tilde{\pi}} \sim D$, we define $\mathbf{x}_t^{\pi^\star}, \mathbf{u}_t^{\pi^\star}$ and $\mathbf{x}_t^{\tilde{\pi}}, \mathbf{u}_t^{\tilde{\pi}}$ as the states and inputs given by the expert policy $\pi^\star$ and chunked policy $\tilde{\pi}$ in closed-loop. We may then view $(\mathbf{x}_t^{\pi^\star}, \mathbf{u}_t^{\pi^\star})$ as the resulting trajectory generated by appropriately defined "input perturbations" to the closed-loop chunked

system $f^{\tilde{\pi}}$: $\mathbf{x}_{t+1}^{\pi^\star} = f^{\tilde{\pi}}(\mathbf{x}_t^{\pi^\star}, \Delta_{\mathbf{u}_t})$, $t \geq 1$, where we define

$$\Delta_{\mathbf{u}_t} \triangleq \mathbf{u}_t^{\pi^\star} - \mathsf{chunk}_s[\tilde{\pi}](\mathbf{x}_{t_k}^{\pi^\star}),$$

and $t_k = \lfloor \frac{t-1}{\ell} \rfloor$ and $s = t - 1 \mod \ell$. Therefore, applying the $(\tilde{C}, \tilde{\rho})$-ISS of $(\tilde{\pi}, f)$, we have:

$$\|\mathbf{x}_t^{\tilde{\pi}} - \mathbf{x}_t^{\pi^\star}\| \leq \tilde{C} \sum_{s=1}^{t-1} \tilde{\rho}^{t-1-k} \|\Delta_{\mathbf{u}_t}\|$$

$$\iff \mathbf{J}_{\mathrm{TRAJ},1,T}^{\mathbf{x}}(\tilde{\pi}) \leq \frac{\tilde{C}}{1-\tilde{\rho}} \mathbf{J}_{\mathrm{DEMO},1,T}(\tilde{\pi}; \mathbb{P}_{\pi^\star}).$$

The second line follows straightforwardly by summing both sides from $t = 1$ to $T$ and applying an expectation. To extend this bound from $\mathbf{J}_{\mathrm{TRAJ},1,T}^{\mathbf{x}}$ to general $\mathbf{J}_{\mathrm{TRAJ},p,T}^{\mathbf{x}}$, we leverage Lemma D.3. We define the vectors $\mathbf{u}, \mathbf{v} \in \mathbb{R}^{T-1}$:

$$\mathbf{u} = \begin{bmatrix} \|\mathbf{x}_2^{\tilde{\pi}} - \mathbf{x}_2^{\pi^\star}\| & \cdots & \|\mathbf{x}_T^{\tilde{\pi}} - \mathbf{x}_T^{\pi^\star}\| \end{bmatrix}^\top \in \mathbb{R}^{T-1},$$

$$\mathbf{v} = \begin{bmatrix} \|\Delta_{\mathbf{u}_1}\| & \cdots & \|\Delta_{\mathbf{u}_{T-1}}\| \end{bmatrix}^\top \in \mathbb{R}^{T-1}.$$

We observe that defining $\mathbf{A}(\tilde{\rho})$ as in Lemma D.3, we have $\mathbf{u} = \tilde{C}\mathbf{A}\mathbf{v}$. Taking the $p$-norm on both sides and applying Lemma D.3 yields: $\|\mathbf{u}\|_p \leq \frac{\tilde{C}}{1-\tilde{\rho}}\|\mathbf{v}\|_p$. Taking the $p$-th power and applying an expectation over $\mathbf{x}_1 \sim D$ on both sides yields the desired bound on $\mathbf{J}_{\mathrm{TRAJ},p,T}^{\mathbf{x}}$ in terms of $\mathbf{J}_{\mathrm{DEMO},p,T}$.

To extend this a bound on $\mathbf{J}_{\mathrm{TRAJ},p,T}$, we apply a similar bound to Lemma D.1. However, we require some alterations since $\tilde{\pi}$ is not a Markovian policy. We may add and subtract to yield:

$$\|\mathbf{u}_t^{\tilde{\pi}} - \mathbf{u}_t^{\pi^\star}\| \leq \|\mathbf{u}_t^{\tilde{\pi}} - \mathsf{chunk}_s[\tilde{\pi}](\mathbf{x}_{t_k}^{\pi^\star})\| - \|\mathbf{u}_t^{\pi^\star} - \mathsf{chunk}_s[\tilde{\pi}](\mathbf{x}_{t_k}^{\pi^\star})\|.$$

Summing up the second term over $t$ yields $\mathbf{J}_{\mathrm{DEMO},T}$. To analyze the first term, we recall that $\mathbf{u}_t^{\hat{\pi}}$ and $\mathsf{chunk}_s[\tilde{\pi}](\mathbf{x}_{t_k}^{\pi^\star})$ result from conditioning on the state every $t_k$ timesteps, then playing the next $\ell$ actions generated by the simulated closed-loop system $(\hat{\pi}, \hat{f})$. Since by assumption $\hat{f}^{\tilde{\pi}}$ is $(\tilde{C}, \rho)$-ISS, this means that for each $t_k$ and $s = 0, \ldots, \ell-1$,

$$\|\mathbf{x}_{t_k+s} - \tilde{\mathbf{x}}_{t_k+s}\| \leq \tilde{C}\tilde{\rho}^s \|\mathbf{x}_{t_k}^{\tilde{\pi}} - \mathbf{x}_{t_k}^{\pi^\star}\|,$$

where $\mathbf{x}_{t_k+s} = (\hat{f}^{\tilde{\pi}})^s(\mathbf{x}_{t_k}^{\tilde{\pi}})$, $\tilde{\mathbf{x}}_{t_k+s} = (\hat{f}^{\tilde{\pi}})^s(\mathbf{x}_{t_k}^{\pi^\star})$. Furthermore, since $\mathbf{u}_{t_k+s}^{\tilde{\pi}} \triangleq \hat{\pi}(\mathbf{x}_{t_k+s})$ and similarly $\mathsf{chunk}_s[\tilde{\pi}](\mathbf{x}_{t_k}^{\pi^\star}) \triangleq \hat{\pi}(\tilde{\mathbf{x}}_{t_k+s})$, applying $L_\pi$-Lipschitzness of $\hat{\pi}$ yields:

$$\sum_{k=1}^{T-1/\ell} \sum_{s=0}^{\ell-1} \|\mathbf{u}_{t_k+s}^{\tilde{\pi}} - \mathsf{chunk}_s[\tilde{\pi}](\mathbf{x}_{t_k+s}^{\pi^\star})\|^p \leq \sum_{k=1}^{T-1/\ell} \sum_{s=0}^{\ell-1} L_\pi^p(\tilde{C}\tilde{\rho}^s)^p \|\mathbf{x}_{t_k}^{\tilde{\pi}} - \mathbf{x}_{t_k}^{\pi^\star}\|^p$$

$$\leq \left(\frac{L_\pi \tilde{C}}{1-\tilde{\rho}}\right)^p \sum_{t=1}^{T} \|\mathbf{x}_t^{\tilde{\pi}} - \mathbf{x}_t^{\pi^\star}\|^p.$$

Putting these pieces together, we get:

$$\mathbf{J}_{\mathrm{TRAJ},p,t}(\tilde{\pi}) \leq \mathbf{J}_{\mathrm{TRAJ},p,t}^{\mathbf{x}}(\tilde{\pi}) + \sum_{t=1}^{T} \min\{1, \|\mathbf{u}_t^{\tilde{\pi}} - \mathbf{u}_t^{\pi^\star}\|^p\}$$

$$\leq \mathbf{J}_{\mathrm{TRAJ},p,t}^{\mathbf{x}}(\tilde{\pi}) + 2^p \mathbf{J}_{\mathrm{DEMO},p,T}(\tilde{\pi}; \mathbb{P}_{\pi^\star}) + \left(\frac{2L_\pi \tilde{C}}{1-\tilde{\rho}}\right)^p \sum_{t=1}^{T} \min\{1, \|\mathbf{x}_t^{\tilde{\pi}} - \mathbf{x}_t^{\pi^\star}\|^p\}$$

$$\leq \left(1 + \left(\frac{2L_\pi \tilde{C}}{1-\tilde{\rho}}\right)^p\right) \mathbf{J}_{\mathrm{TRAJ},p,t}^{\mathbf{x}}(\tilde{\pi}) + 2^p \mathbf{J}_{\mathrm{DEMO},p,T}(\tilde{\pi}; \mathbb{P}_{\pi^\star}).$$

Plugging in the upper bound on $\mathbf{J}_{\mathrm{TRAJ},p,T}^{\mathbf{x}}(\hat{\pi})$ completes the result.

$\square$

In particular, specifying this result to Proposition 3.2 follows straightforwardly by setting $p = 2$. Therefore, combining Proposition D.4, which says chunking policies induces EISS, with Proposition D.7, which says EISS chunking policies induce low compounding error, yields the final guarantee.

**Theorem 3** (Full ver. of Theorem 1). *Let Assumption 3.1 hold. Given $a \in (0, 1)$, for sufficiently long chunk-length: $\ell > a^{-1} \log(1/\rho)^{-1} \cdot \log(\text{poly}(L_\pi, C_{\text{ISS}}, 1/a))$, let $\tilde{\pi} = \text{chunked}(\hat{\pi}, g, \ell) \in \Pi_{\text{chunk},\ell}$, such that $(\tilde{\pi}, f)$ is $(\tilde{C}, \rho^{1-a})$, with $\tilde{C} = a^{-1} \log(1/\rho)^{-1} \cdot \text{poly}(L_\pi, C_{\text{ISS}})$. The following bound holds on the trajectory error induced by $\tilde{\pi}$:*

$$\mathbf{J}_{\text{TRAJ},p,T}(\tilde{\pi}) \lesssim \left(1 + \frac{L_\pi \tilde{C}}{1 - \rho^{1-a}}\right)^p \mathbf{J}_{\text{DEMO},p,T}(\tilde{\pi}; \mathbb{P}_{\pi^\star}).$$

# E   PROOFS AND ADDITIONAL DETAILS FOR SECTION 4

## E.1   RL- VERSUS CONTROL-THEORETIC PERSPECTIVES

**The RL-theoretic perspective.** RL-theoretic notions of exploration often take an information-theoretic flavor, where it is captured by notions of "coverage" (Jin et al., 2021; Zhan et al., 2022; Amortila et al., 2024; Jiang & Xie, 2024). Coverage analyses rely on density ratios and thus the existence of densities. In continuous state-action spaces, expert (deterministic) policies typically do not have densities, and thus they can be induced by incorporating (possibly shaped) noise to the actions (Haarnoja et al., 2018; Schulman et al., 2017). Crucially, this makes the policy itself noisy—compare this to Practice 2, where the expert's recorded action is uncorrupted. When the noise is Gaussian, this practice turns maximum-likelihood estimation (MLE) into square-error minimization. Hence, existing analyses of behavior cloning (e.g., Foster et al. (2024)) ensure consistent imitation.

However, this comes at the price of corrupting the demonstrations provided to the learner, which in turn, we show in Appendix E.7, leads to suboptimal rates of estimation. In particular, by reducing imitation learning to MLE over noisy data, the performance of IL is dictated by the capacity of the *stochastic* policy class, as measured by a covering number $N_{\log}(\Pi, \varepsilon)$ under, e.g., the log-loss. For $\sigma_{\mathbf{u}}$-scaled Gaussian noise, this equates to covering under the Euclidean norm at resolution $\approx \sqrt{\sigma_{\mathbf{u}}^2 \varepsilon}$. For non-parametric classes–such as the lower bound constructions leading to Theorem A, this can introduce additional polynomial factors of $\sigma_{\mathbf{u}}^{-1}$ in the estimation error. These factors of $\sigma_{\mathbf{u}}^{-1}$ must then be traded off with the error induced by imitating a noisy expert rather than the true expert labels.

**The control-theoretic perspective.** In the control-theoretic literature, *persistency of excitation* (PE) is a well-established sufficient condition for ensuring parameter recovery in system-identification and adaptive control, which in turn yields performant policy synthesis (Bai & Sastry, 1985; Narendra & Annaswamy, 1987; Willems et al., 2005; Van Waarde et al., 2020). A input-sequence is "PE" if it yields a full-rank sequence of states, which guarantees parameter recovery across all modes the system may encounter. Therefore, when an expert policy may output degenerate trajectories in closed-loop,[9] a natural approach to achieve PE is to inject excitatory noise into the inputs or directly into the system state (Annaswamy, 2023). More modern analyses of both the online linear-quadratic regulator (LQR) problem (Dean et al., 2018; Mania et al., 2019; Simchowitz & Foster, 2020) and of imitation learning Pfrommer et al. (2022); Zhang et al. (2023) have similarly turned toward PE to ensure desirable learning behavior; either relying on process noise (i.e., non-degenerate noise entering additively to the state) to excite state variables, or assuming the ability to directly perturb states during expert demonstration. By contrast, our setting assumes neither the presence of process noise, nor direct access to the system state. Lastly, we do not even assume the system is *controllable*,[10] i.e., we also cannot rely on input perturbations inducing the PE condition.

**Comparisons to the RL and control perspectives.** By combining ideas from RL and control, we arrive at conclusions that may be surprising from either perspective. Compared to the RL perspective, 1. we do not have coverage in the usual sense, 2. we avoid accumulating mean-estimation error

---

[9]See e.g., cases for linear systems under an optimal LQR controller (Polderman, 1986; Lee et al., 2023).

[10]Informally the ability of a system to be steered from any state to another by applying appropriate control inputs, cf. (Kailath, 1980).

from imitating noisy action labels, 3. using the mixture distribution $\mathbb{P}_{\pi^\star, \sigma_{\mathbf{u}}, \alpha}$ subverts the additive $\sigma_{\mathbf{u}}^4$ error in Proposition 4.1. On the control-theoretic side, 1. imitating over $\mathbb{P}_{\pi^\star, \sigma_{\mathbf{u}}, \alpha}$ removes the additive $\sigma_{\mathbf{u}}^2$ error in Suboptimal Proposition 4.2, 2. we avoid any assumption of controllability *as well as* any dependence on the small eigendirections of the controllable subspace. In fact, by removing any additive $\sigma_{\mathbf{u}}$ factor, our bound suggests that we should set the noise-scale $\sigma_{\mathbf{u}}$ as large as permissible!

## E.2 PROOF PRELIMINARIES

We first recall the definition of the linearizations around expert trajectories from Definition 4.3.

$$
\begin{aligned}
\mathbf{A}_t &= \nabla_{\mathbf{x}} f(\mathbf{x}_t^{\pi^\star}, \mathbf{u}_t^{\pi^\star}), \ \mathbf{B}_t = \nabla_{\mathbf{u}} f(\mathbf{x}_t^{\pi^\star}, \mathbf{u}_t^{\pi^\star}), \ \mathbf{K}_t^{\pi^\star} = \nabla_{\mathbf{x}} \pi^\star(\mathbf{x}_t^{\pi^\star}), \\
\mathbf{A}_{s:t}^{\mathrm{cl}} &= (\mathbf{A}_{t-1} + \mathbf{B}_{t-1} \mathbf{K}_{t-1}^{\pi^\star})(\mathbf{A}_{t-2} + \mathbf{B}_{t-2} \mathbf{K}_{t-2}^{\pi^\star}) \cdots (\mathbf{A}_s + \mathbf{B}_s \mathbf{K}_s^{\pi^\star}).
\end{aligned}
\tag{E.1}
$$

We also recall the definition of the controllability Gramian: $\mathbf{W}_{1:t}^{\mathbf{u}}(\mathbf{x}_1^{\pi^\star}) \triangleq \sum_{s=1}^{t-1} \mathbf{A}_{s+1:t}^{\mathrm{cl}} \mathbf{B}_s \mathbf{B}_s^\top \mathbf{A}_{s+1:t}^{\mathrm{cl}}{}^\top$. For a noising distribution that is zero-mean with covariance $\Sigma_{\mathbf{z}}$, $\mathbf{z}_t \overset{\text{i.i.d}}{\sim} \mathcal{D}(\mathbf{0}, \Sigma_{\mathbf{z}})$, we further define the *noise controllability Gramian*:

$$
\mathbf{W}_{1:t}^{\mathbf{z}}(\mathbf{x}_1^{\pi^\star}) \triangleq \sum_{s=1}^{t-1} \mathbf{A}_{s+1:t}^{\mathrm{cl}} \mathbf{B}_s \Sigma_{\mathbf{z}} \mathbf{B}_s^\top \mathbf{A}_{s+1:t}^{\mathrm{cl}}{}^\top.
$$

Note that for $\mathbf{z}_t$ sampled from the Euclidean unit ball, we have $\Sigma_{\mathbf{z}} = \frac{1}{(d_u+2)} \mathbf{I}_{d_u} \succeq \frac{1}{3d_u} \mathbf{I}_{d_u}$, and thus:

$$
\mathbf{W}_{1:t}^{\mathbf{z}}(\mathbf{x}_1^{\pi^\star}) \succeq \frac{1}{3d_u} \mathbf{W}_{1:t}^{\mathbf{u}}(\mathbf{x}_1^{\pi^\star}).
$$

The ensuing results are written for any noising distribution $\mathcal{D}(\mathbf{0}, \Sigma_{\mathbf{z}})$ that are 1-bounded, mean-zero, with covariance $\Sigma_{\mathbf{z}} \succ \mathbf{0}$, unless otherwise stated.

We now establish that the linear (time-varying) system induced by linearizations along expert trajectories inherits $(C_{\mathrm{ISS}}, \rho)$-EISS. We note that though the original dynamics and expert policy are time-invariant, the linearized system is in general not.

**Lemma E.1.** *Let Assumption 4.1 hold. Given a nominal trajectory generated as*

$$
\mathbf{x}_{t+1}^{\pi^\star} = f(\mathbf{x}_t^{\pi^\star}, \mathbf{u}_t^{\pi^\star}), \ \mathbf{u}_t^{\pi^\star} = \pi^\star(\mathbf{x}_t^{\pi^\star}), \ t \geq 1, \ \mathbf{x}_1^{\pi^\star} \sim \mathbb{P}_{\mathbf{x}_1^{\pi^\star}},
$$

*and recall the linearizations in Eq. (E.1). Then, the following bounds hold:*

$$
\|\mathbf{A}_{1:t}^{\mathrm{cl}}\|_{\mathrm{op}} \leq C_{\mathrm{ISS}} \rho^{t-1}, \ \|\mathbf{A}_{s:t}^{\mathrm{cl}}\|_{\mathrm{op}} \leq C_{\mathrm{ISS}} \rho^{t-s}, \ \|\mathbf{A}_{s+1:t}^{\mathrm{cl}} \mathbf{B}_s\|_{\mathrm{op}} \leq C_{\mathrm{ISS}} \rho^{t-1-s}, \ \text{for all } 1 \leq s \leq t.
$$

An equivalent way to view Lemma E.1 is: for an input perturbation sequence $\{\Delta_{\mathbf{u}_t}\}_{t \geq 1}$, the incremental trajectory $\{\Delta_{\mathbf{x}_t}\}_{t \geq 1}$, $\Delta_{\mathbf{x}_t} \triangleq \hat{\mathbf{x}}_t - \mathbf{x}_t^{\pi^\star}$ induced by linearizations around an expert trajectory $\{\mathbf{x}_t^{\pi^\star}\}_{t \geq 1}$ is $(C_{\mathrm{ISS}}, \rho)$-EISS:

$$
\Delta_{\mathbf{x}_{t+1}} = (\mathbf{A}_t + \mathbf{B}_t \mathbf{K}_t^{\pi^\star}) \Delta_{\mathbf{x}_t} + \mathbf{B}_t \Delta_{\mathbf{u}_t} = \mathbf{A}_{1:t+1}^{\mathrm{cl}} \Delta_{\mathbf{x}_1} + \sum_{s=1}^t \mathbf{A}_{s+1:t+1}^{\mathrm{cl}} \mathbf{B}_s \Delta_{\mathbf{u}_s}.
$$

*Proof of Lemma E.1.* Given the nominal trajectory $\{\mathbf{x}_t^{\pi^\star}\}_{t \geq 1}$ generated by $\mathbf{x}_{t+1}^{\pi^\star} = f(\mathbf{x}_t^{\pi^\star}, \mathbf{u}_t^{\pi^\star})$ and the corresponding linearizations $\mathbf{A}_t, \mathbf{B}_t, \mathbf{K}^{\pi^\star}$ evaluated along the trajectory, consider the trajectory

$\{\tilde{\mathbf{x}}_t\}_{t\geq 1}$ generated as $\tilde{\mathbf{x}}_{t+1} = f(\tilde{\mathbf{x}}_t, \pi^\star(\tilde{\mathbf{x}}_t) + \mathbf{u}_t, t)$. Expanding the Jacobian linearizations, we have

$$\tilde{\mathbf{x}}_{t+1} - \mathbf{x}_{t+1}^{\pi^\star} = f(\tilde{\mathbf{x}}_t, \pi^\star(\tilde{\mathbf{x}}_t) + \mathbf{u}_t, t) - f(\mathbf{x}_t^{\pi^\star}, \pi^\star(\mathbf{x}_t^{\pi^\star}), t)$$

$$= \mathbf{A}_t(\tilde{\mathbf{x}}_t - \mathbf{x}_t^{\pi^\star}) + \mathbf{B}_t(\pi^\star(\tilde{\mathbf{x}}_t) + \mathbf{u}_t - \pi^\star(\mathbf{x}_t^{\pi^\star})) + \underbrace{O\left(\left\|\begin{bmatrix}\tilde{\mathbf{x}}_t - \mathbf{x}_t^{\pi^\star} \\ \pi^\star(\tilde{\mathbf{x}}_t) - \pi^\star(\mathbf{x}_t^{\pi^\star}) + \mathbf{u}_t\end{bmatrix}\right\|^2\right)}_{\triangleq \mathbf{r}_t^{\mathbf{x}}}$$

$$= (\mathbf{A}_t + \mathbf{B}_t\mathbf{K}_t^{\pi^\star})(\tilde{\mathbf{x}}_t - \mathbf{x}_t^{\pi^\star}) + \mathbf{B}_t(\mathbf{u}_t + \underbrace{O(\|\tilde{\mathbf{x}}_t - \mathbf{x}_t^{\pi^\star}\|^2)}_{\triangleq \mathbf{r}_t^{\mathbf{u}}}) + \mathbf{r}_t^{\mathbf{x}}$$

$$= \mathbf{A}_{1:t+1}^{\mathrm{cl}}(\tilde{\mathbf{x}}_1 - \mathbf{x}_1^{\pi^\star}) + \sum_{s=1}^t \mathbf{A}_{s+1:t+1}^{\mathrm{cl}}\left(\mathbf{B}_s(\mathbf{u}_s + \mathbf{r}_s^{\mathbf{u}}) + \mathbf{r}_s^{\mathbf{x}}\right),$$

(E.2)

We perform a simple sensitivity analysis to isolate $\mathbf{A}_{1:t}^{\mathrm{cl}}$. Defining the displacements $\delta_t^{\mathbf{x}} = \tilde{\mathbf{x}}_t - \mathbf{x}_t^{\pi^\star}$, and setting $\mathbf{u}_t = \mathbf{0}$, $t \geq 1$, we see that $\frac{\partial}{\partial \delta_1^{\mathbf{x}}}\delta_t^{\mathbf{x}} = \mathbf{A}_{1:t}^{\mathrm{cl}}$, since we observe $\delta_t^{\mathbf{x}} = \mathbf{A}_{1:t}^{\mathrm{cl}}\delta_1^{\mathbf{x}} + \sum_{s=1}^{t-1}\mathbf{A}_{s+1:t}^{\mathrm{cl}}(\mathbf{B}_s\mathbf{r}_s^{\mathbf{u}} + \mathbf{r}_s^{\mathbf{x}})$ is linear in $\delta_1^{\mathbf{x}}$ and the residuals $\mathbf{r}_s^{\mathbf{u}}, \mathbf{r}_s^{\mathbf{x}}$ are higher-order by definition. On the other hand, by the $(C_{\mathrm{ISS}}, \rho)$-EISS of $(\pi^\star, f)$, we know that $\|\delta_t^{\mathbf{x}}\| \leq C_{\mathrm{ISS}}\rho^{t-1}\|\delta_1^{\mathbf{x}}\|$. By definition of the operator norm, we have $\|\mathbf{A}_{1:t}^{\mathrm{cl}}\|_{\mathrm{op}} = \sup_{\mathbf{v}} \|\mathbf{A}_{1:t}^{\mathrm{cl}}\mathbf{v}\|/\|\mathbf{v}\|$, and thus by a limiting argument $\delta_1^{\mathbf{x}} \to \mathbf{0}$, we see

$$\|\mathbf{A}_{1:t}^{\mathrm{cl}}\|_{\mathrm{op}} \leq \lim_{\delta_1^{\mathbf{x}} \to \mathbf{0}} \|\delta_t^{\mathbf{x}}\|/\|\delta_1^{\mathbf{x}}\| \leq C_{\mathrm{ISS}}\rho^{t-1}.$$

To establish a similar bound on $\mathbf{A}_{s:t}^{\mathrm{cl}}$, we observe that $\mathbf{x}_{t+1}^{\pi^\star} = f(\mathbf{x}_t^{\pi^\star}, \pi^\star(\mathbf{x}_t^{\pi^\star}))$ is by definition a *time-invariant* closed-loop system, we may apply $(C_{\mathrm{ISS}}, \rho)$-EISS starting from $\delta_s^{\mathbf{x}}$ as the initial displacement such that $\|\delta_t^{\mathbf{x}}\| \leq C_{\mathrm{ISS}}\rho^{t-s}\|\delta_s^{\mathbf{x}}\|$. Applying the same argument yields:

$$\|\mathbf{A}_{s:t}^{\mathrm{cl}}\|_{\mathrm{op}} \leq \lim_{\delta_s^{\mathbf{x}} \to \mathbf{0}} \|\delta_t^{\mathbf{x}}\|/\|\delta_s^{\mathbf{x}}\| \leq C_{\mathrm{ISS}}\rho^{t-s}.$$

Now, instead setting $\tilde{\mathbf{x}}_1 - \mathbf{x}_1^{\pi^\star} = \mathbf{0}$ and an impulse input $\{\mathbf{0}, \dots, \mathbf{0}, \mathbf{u}_k, \mathbf{0}, \dots\}$ for some $k$, we have $\delta_t^{\mathbf{x}} = \mathbf{A}_{k+1:t}^{\mathrm{cl}}\mathbf{B}_k\mathbf{u}_k + \sum_{s=k}^{t-1}\mathbf{A}_{s+1:t}^{\mathrm{cl}}(\mathbf{B}_s\mathbf{r}_s^{\mathbf{u}} + \mathbf{r}_s^{\mathbf{x}})$. By the same appeal to EISS of $(\pi^\star, f)$ and limiting argument $\mathbf{u}_k \to \mathbf{0}$, we have: $\|\mathbf{A}_{k+1:t}^{\mathrm{cl}}\mathbf{B}_k\|_{\mathrm{op}} \leq C_{\mathrm{ISS}}\rho^{t-1-k}$. Notably, this holds for any $k$ and $t \geq k$, completing the proof.

$\square$

Given an expert-induced trajectory $\mathbf{x}_{t+1} = f^{\pi^\star}(\mathbf{x}_t)$, $t \in [T-1]$, consider *noise-injected* trajectories $(\tilde{\mathbf{x}}_t, \tilde{\mathbf{u}}_t)_{t\geq 1} \sim \mathbb{P}_{\pi^\star, \sigma_{\mathbf{u}}}$ as in Definition 4.1. Our next result demonstrates that the noise-injected trajectories are well-described by the expert linearizations, up to a higher-order term quadratic in the noise-scale $\sigma_{\mathbf{u}}$.

**Proposition E.2.** *Let Assumption 4.1 hold. Consider noise-injected expert trajectories $\{\tilde{\mathbf{x}}_t, \pi^\star(\tilde{\mathbf{x}}_t)\}_{t\geq 1} \sim \mathbb{P}_{\pi^\star, \sigma_{\mathbf{u}}}$ for a given initial condition $\tilde{\mathbf{x}}_1 \sim D$: $\tilde{\mathbf{x}}_{t+1} = f(\tilde{\mathbf{x}}_t, \pi^\star(\tilde{\mathbf{x}}_t) + \sigma_{\mathbf{u}}\mathbf{z}_t)$, $\mathbf{z}_t \overset{\mathrm{i.i.d}}{\sim} \mathcal{D}(\mathbf{0}, \Sigma_{\mathbf{z}})$. Consider the linearizations along an expert trajectory given in (E.1), setting $\mathbf{x}_1^{\pi^\star} = \tilde{\mathbf{x}}_1$. Define the linear and residual components of the noised state $\tilde{\mathbf{x}}_t$:*

$$\tilde{\mathbf{x}}_t^{\mathrm{lin}} \triangleq \mathbf{x}_t^{\pi^\star} + \sum_{s=1}^{t-1}\mathbf{A}_{s+1:t}^{\mathrm{cl}}\mathbf{B}_s\mathbf{u}_s, \quad \tilde{\mathbf{x}}_t^{\mathrm{res}} \triangleq \tilde{\mathbf{x}}_t - \tilde{\mathbf{x}}_t^{\mathrm{lin}}, \quad t \geq 1. \tag{E.3}$$

*Then, as long as $\sigma_{\mathbf{u}} \leq \frac{1}{2}c_{\mathrm{stab}}\frac{\sqrt{1+4C_{\mathbf{K}}^2}}{C_\pi}$, and defining $C_{\mathbf{r}} \triangleq C_\pi + 4C_{\mathrm{reg}}(1 + 4C_{\mathbf{K}}^2)$, we have $\|\tilde{\mathbf{x}}_t^{\mathrm{res}}\| \leq C_{\mathrm{stab}}^3 C_{\mathbf{r}}\sigma_{\mathbf{u}}^2$, $t \geq 1$ almost surely over $\tilde{\mathbf{x}}_1 \sim D$ and $\{\mathbf{z}_s\} \overset{\mathrm{i.i.d}}{\sim} \mathcal{D}(\mathbf{0}, \Sigma_{\mathbf{z}})$.*

*Proof of Proposition E.2.* Given the nominal trajectory $\{\mathbf{x}_t^{\pi^\star}\}_{t\geq 1}$ generated by $\mathbf{x}_{t+1}^{\pi^\star} = f(\mathbf{x}_t^{\pi^\star}, \mathbf{u}_t^{\pi^\star})$ and the corresponding linearizations $\mathbf{A}_t, \mathbf{B}_t, \mathbf{K}^{\pi^\star}$ (E.1) evaluated along the trajectory, consider the

trajectory $\{\tilde{\mathbf{x}}_t\}_{t \geq 1}$ generated as $\tilde{\mathbf{x}}_{t+1} = f(\tilde{\mathbf{x}}_t, \pi^\star(\tilde{\mathbf{x}}_t) + \mathbf{u}_t, t)$, with $\tilde{\mathbf{x}}_1 = \mathbf{x}_1^{\pi^\star}$. Then, following (E.2), we may write:

$$
\begin{aligned}
\tilde{\mathbf{x}}_{t+1} - \mathbf{x}_{t+1}^{\pi^\star} &= f(\tilde{\mathbf{x}}_t, \pi^\star(\tilde{\mathbf{x}}_t) + \mathbf{u}_t, t) - f(\mathbf{x}_t^{\pi^\star}, \pi^\star(\mathbf{x}_t^{\pi^\star}), t) \\
&= \mathbf{A}_{1:t+1}^{\mathrm{cl}}(\tilde{\mathbf{x}}_1 - \mathbf{x}_1^{\pi^\star}) + \sum_{s=1}^{t} \mathbf{A}_{s+1:t+1}^{\mathrm{cl}}\left(\mathbf{B}_s(\mathbf{u}_s + \mathbf{r}_s^{\mathbf{u}}) + \mathbf{r}_s^{\mathbf{x}}\right) \\
&= \sum_{s=1}^{t} \mathbf{A}_{s+1:t+1}^{\mathrm{cl}}\left(\mathbf{B}_s(\mathbf{u}_s + \mathbf{r}_s^{\mathbf{u}}) + \mathbf{r}_s^{\mathbf{x}}\right).
\end{aligned}
$$

where we recall $\mathbf{r}_t^{\mathbf{x}}$ and $\mathbf{r}_t^{\mathbf{u}}$ are the second-order remainder terms of the dynamics and policy outputs, respectively. By Assumption 4.1, these are bounded by:

$$
\begin{aligned}
\|\mathbf{r}_t^{\mathbf{u}}\| &\leq C_\pi \|\tilde{\mathbf{x}}_t - \mathbf{x}_t^{\pi^\star}\|^2 \\
\|\mathbf{r}_t^{\mathbf{x}}\| &\leq C_{\mathrm{reg}}\left(\|\tilde{\mathbf{x}}_t - \mathbf{x}_t^{\pi^\star}\|^2 + \|\pi^\star(\tilde{\mathbf{x}}_t) - \pi^\star(\mathbf{x}_t^{\pi^\star}) + \mathbf{u}_t\|^2\right) \\
&\leq C_{\mathrm{reg}}\left(\|\tilde{\mathbf{x}}_t - \mathbf{x}_t^{\pi^\star}\|^2 + 2\|\pi^\star(\tilde{\mathbf{x}}_t) - \pi^\star(\mathbf{x}_t^{\pi^\star})\|^2 + 2\|\mathbf{u}_t\|^2\right) \\
&\leq C_{\mathrm{reg}}\left((1 + 4C_{\mathbf{K}}^2)\|\tilde{\mathbf{x}}_t - \mathbf{x}_t^{\pi^\star}\|^2 + 4\|\mathbf{r}_t^{\mathbf{u}}\|^2 + 2\|\mathbf{u}_t\|^2\right) \\
&\leq C_{\mathrm{reg}}\left((1 + 4C_{\mathbf{K}}^2)\|\tilde{\mathbf{x}}_t - \mathbf{x}_t^{\pi^\star}\|^2 + 4C_\pi^2\|\tilde{\mathbf{x}}_t - \mathbf{x}_t^{\pi^\star}\|^4 + 2\|\mathbf{u}_t\|^2\right).
\end{aligned}
$$

Defining $\varepsilon_{\mathbf{x}_t} = \|\tilde{\mathbf{x}}_t - \mathbf{x}_t^{\pi^\star}\|$, and $\mathbf{u}_t \overset{\mathrm{i.i.d}}{\sim} \mathcal{D}(\mathbf{0}, \Sigma_{\mathbf{u}}; \sigma_{\mathbf{u}})$ are iid zero-mean, $\Sigma_{\mathbf{u}}$ covariance, $\sigma_{\mathbf{u}}$-bounded random vectors, we want to bound the mean and covariance of $\tilde{\mathbf{x}}_t$. We note the presence of the quartic term $4C_\pi^2\varepsilon_{\mathbf{x}_t}^4$ in our remainder term; we first impose $\varepsilon_{\mathbf{x}_t}^2 \leq \frac{1+4C_{\mathbf{K}}^2}{4C_\pi^2}$ to absorb it into the quadratic term, then show this constraint is obviated for sufficiently small $\|\mathbf{u}_s\|$.

Since $(\pi^\star, f)$ is $(C_{\mathrm{ISS}}, \rho)$-EISS, we have $\varepsilon_{\mathbf{x}_t} \leq C_{\mathrm{ISS}} \sum_{s=1}^{t-1} \rho^{t-s-1}\|\mathbf{u}_s\| \leq \frac{C_{\mathrm{ISS}}}{1-\rho} \max_{s \leq t-1}\|\mathbf{u}_s\|$. Therefore, we have:

$$
\begin{aligned}
\|\mathbf{r}_t^{\mathbf{u}}\| &\leq C_\pi \varepsilon_{\mathbf{x}_t}^2 \leq C_\pi C_{\mathrm{stab}}^2 \max_{s \leq t-1}\|\mathbf{u}_s\|^2 \\
&\leq C_\pi C_{\mathrm{stab}}^2 \sigma_{\mathbf{u}}^2 \\
\|\mathbf{r}_t^{\mathbf{x}}\| &\leq C_{\mathrm{reg}}\left((1 + 4C_{\mathbf{K}}^2)\varepsilon_{\mathbf{x}_t}^2 + 4C_\pi^2 \varepsilon_{\mathbf{x}_t}^4 + 2\|\mathbf{u}_t\|^2\right) \\
&\leq C_{\mathrm{reg}}\left(2(1 + 4C_{\mathbf{K}}^2)C_{\mathrm{stab}}^2 \max_{s \leq t-1}\|\mathbf{u}_s\|^2 + 2\|\mathbf{u}_t\|^2\right) \\
&\leq 4C_{\mathrm{reg}}(1 + 4C_{\mathbf{K}}^2)C_{\mathrm{stab}}^2 \sigma_{\mathbf{u}}^2.
\end{aligned}
$$

These hold as long as $\sigma_{\mathbf{u}}$ is small enough such that $\varepsilon_{\mathbf{x}_t}^2 \leq C_{\mathrm{stab}}^2 \sigma_{\mathbf{u}}^2 \leq \frac{1+4C_{\mathbf{K}}^2}{4C_\pi^2}$, which holds for $\sigma_{\mathbf{u}} \leq \frac{1}{2} c_{\mathrm{stab}} \frac{\sqrt{1+4C_{\mathbf{K}}^2}}{C_\pi}$. With these perturbation bounds in hand, we now move onto bounding the linear and residual components of $\tilde{\mathbf{x}}_t$. We have immediately:

$$
\begin{aligned}
\tilde{\mathbf{x}}_t - \mathbf{x}_t^{\pi^\star} &= \sum_{s=1}^{t-1} \mathbf{A}_{s+1:t}^{\mathrm{cl}}\left(\mathbf{B}_s(\mathbf{u}_s + \mathbf{r}_s^{\mathbf{u}}) + \mathbf{r}_s^{\mathbf{x}}\right) \\
\implies \|\tilde{\mathbf{x}}_t^{\mathrm{res}}\| = \|\tilde{\mathbf{x}}_t^{\mathrm{lin}} - \mathbf{x}_t^{\pi^\star}\| &= \left\|\sum_{s=1}^{t-1} \mathbf{A}_{s+1:t}^{\mathrm{cl}}\left(\mathbf{B}_s \mathbf{r}_s^{\mathbf{u}} + \mathbf{r}_s^{\mathbf{x}}\right)\right\| \\
&\leq \sum_{s=1}^{t-1} \|\mathbf{A}_{s+1:t}^{\mathrm{cl}}\mathbf{B}_s\|_{\mathrm{op}}\|\mathbf{r}_s^{\mathbf{u}}\| + \|\mathbf{A}_{s+1:t}^{\mathrm{cl}}\|_{\mathrm{op}}\|\mathbf{r}_s^{\mathbf{x}}\| \\
&\leq C_{\mathrm{stab}}^3 \underbrace{\left(C_\pi + 4C_{\mathrm{reg}}(1 + 4C_{\mathbf{K}}^2)\right)}_{\triangleq C_{\mathbf{r}}} \sigma_{\mathbf{u}}^2.
\end{aligned}
$$

This completes the proof. $\qquad\square$

We now proceed with the *one-step controllable* setting, where $\mathbf{W}_{1:t}^{\mathbf{u}} \succ \underline{\lambda}_{\mathbf{W}}\mathbf{I}$ for all $t \geq 2$, leading up to Suboptimal Proposition 4.2, where we also fit $\hat{\pi}$ purely on noise-injected trajectories, in order to grasp the core ideas and the remaining key deficiencies.

### E.3 ONE-STEP CONTROLLABLE CASE: PERSISTENCY OF EXCITATION

We consider settings where the controllability Gramians induced by linearizations around an expert trajectory are always full-rank.

**Assumption E.1** (Linearized one-step controllability). Let $\mathbf{W}_{1:t}^{\mathbf{u}}(\mathbf{x}_1^{\pi^\star}) \succeq \underline{\lambda}_{\mathbf{W}}\mathbf{I}_{d_u}$, $t \geq 2$ w.p. 1 over $\mathbf{x}_1^{\pi^\star} \sim D$ for some $\underline{\lambda}_{\mathbf{W}} > 0$. Consider the noise-controllability Gramians $\mathbf{W}_{1:t}^{\mathbf{z}}$ as defined in Definition 4.3. Accordingly, there exists $\underline{\lambda}_{\mathbf{z}} > 0$ such that w.p. 1 over $\mathbf{x}_1^{\pi^\star} \sim D$, $\mathbf{W}_{1:t}^{\mathbf{z}}(\mathbf{x}_1^{\pi^\star}) \succeq \underline{\lambda}_{\mathbf{z}}\mathbf{I}_{d_x}$, $t \geq 2$.

Proposition E.2 in conjunction with Assumption E.1 implies the noise-injected expert states $\tilde{\mathbf{x}}_t$ form a *full-rank* covariance around $\mathbf{x}_t^{\pi^\star}$ for *each* timestep $t = 2, \ldots, T$. This corresponds with the well-known notion of *persistency of excitation* from the control literature (Annaswamy, 2023). As a consequence of Proposition E.2, we have the following excitation bound.

**Corollary E.1.** *Let Assumption 4.1 hold and $C_{\mathbf{r}}$ be as defined in Proposition E.2. Recall the noise-controllability Gramian $\mathbf{W}_{1:t}^{\mathbf{z}}$ as in Assumption E.1. As long as:*

$$\sigma_{\mathbf{u}} \lesssim \min\left\{\lambda_{\min}^+\left(\mathbf{W}_{1:t}^{\mathbf{z}}(\mathbf{x}_1^{\pi^\star})\right) c_{\text{stab}}^4 C_{\mathbf{r}}^{-1}, \; c_{\text{stab}}\frac{\sqrt{1+4C_{\mathbf{K}}^2}}{C_\pi}\right\},$$

*the following holds almost surely over $\tilde{\mathbf{x}}_1 = \mathbf{x}_1^{\pi^\star} \sim \mathbb{P}_{\mathbf{x}_1^{\pi^\star}}$ and $\{\mathbf{z}_s\} \overset{\text{i.i.d}}{\sim} \mathcal{D}(\mathbf{0}, \Sigma_{\mathbf{z}})$:*

$$\mathbb{E}_{\tilde{\mathbf{x}}_t|\mathbf{x}_1^{\pi^\star}}\left[\left(\tilde{\mathbf{x}}_t - \mathbf{x}_t^{\pi^\star}\right)\left(\tilde{\mathbf{x}}_t - \mathbf{x}_t^{\pi^\star}\right)^\top\right] \succeq \frac{\sigma_{\mathbf{u}}^2}{2}\mathbf{W}_{1:t}^{\mathbf{z}}(\mathbf{x}_1^{\pi^\star}). \tag{E.4}$$

*Proof of Corollary E.1.* Denoting $\mathbf{C} = \sum_{s=1}^{t-1} \mathbf{A}_{s+1:t}^{\text{cl}}\mathbf{B}_s\mathbf{u}_s$ and $\mathbf{E} = \sum_{s=1}^{t-1} \mathbf{A}_{s+1:t}^{\text{cl}}(\mathbf{B}_s\mathbf{r}_s^{\mathbf{u}} + \mathbf{r}_s^{\mathbf{x}})$, we bound the second moment of $\tilde{\mathbf{x}}_t - \mathbf{x}_t^{\pi^\star}$:

$$\mathbb{E}_{\mathbf{u}}\left[\left(\tilde{\mathbf{x}}_t - \mathbf{x}_t^{\pi^\star}\right)\left(\tilde{\mathbf{x}}_t - \mathbf{x}_t^{\pi^\star}\right)^\top\right] = \mathbb{E}_{\mathbf{u}}\left[(\mathbf{C} + \mathbf{E})(\mathbf{C} + \mathbf{E})^\top\right]$$

$$= \mathbb{E}_{\mathbf{u}}\left[\mathbf{C}\mathbf{C}^\top\right] + \mathbb{E}_{\mathbf{u}}\left[\mathbf{E}\mathbf{E}^\top\right] + \mathbb{E}_{\mathbf{u}}\left[\mathbf{E}\mathbf{C}^\top + \mathbf{C}\mathbf{E}^\top\right]$$

$$\succeq \mathbb{E}_{\mathbf{u}}\left[\mathbf{C}\mathbf{C}^\top\right] + \mathbb{E}_{\mathbf{u}}\left[\mathbf{E}\mathbf{C}^\top + \mathbf{C}\mathbf{E}^\top\right]$$

By Weyl's inequality (Horn & Johnson, 2012), we have for each $k = 1, \ldots, \text{rank}(\mathbb{E}_{\mathbf{u}}\left[\mathbf{C}\mathbf{C}^\top\right])$:

$$\left|\lambda_k(\mathbb{E}_{\mathbf{u}}[(\mathbf{C} + \mathbf{E})(\mathbf{C} + \mathbf{E})^\top]) - \lambda_k(\mathbb{E}_{\mathbf{u}}[\mathbf{C}\mathbf{C}^\top])\right| \leq 2\|\mathbf{E}\mathbf{C}^\top\|_{\text{op}}$$

$$\leq 2\left(\frac{C_{\text{ISS}}}{1-\rho}\sigma_{\mathbf{u}}\right)\left(C_{\text{stab}}^3\left(C_\pi + 4C_{\text{reg}}(1 + 4C_{\mathbf{K}}^2)\right)\sigma_{\mathbf{u}}^2\right)$$

$$= 2C_{\text{stab}}^4 C_{\mathbf{r}}\sigma_{\mathbf{u}}^3.$$

Rearranging the above yields, for each $k = 1, \ldots, \text{rank}(\mathbb{E}_{\mathbf{u}}\left[\mathbf{C}\mathbf{C}^\top\right])$:

$$\lambda_k(\mathbb{E}_{\mathbf{u}}[(\mathbf{C} + \mathbf{E})(\mathbf{C} + \mathbf{E})^\top]) \geq \lambda_k(\mathbb{E}_{\mathbf{u}}[\mathbf{C}\mathbf{C}^\top]) - 2C_{\text{stab}}^4\left(C_\pi + 4C_{\text{reg}}(1 + 4C_{\mathbf{K}}^2)\right)\sigma_{\mathbf{u}}^3.$$

Therefore, for sufficiently small $\sigma_{\mathbf{u}}$ such that:

$$\sigma_{\mathbf{u}} \leq \frac{1}{4}\frac{\lambda_{\min}^+(\mathbb{E}_{\mathbf{u}}[\mathbf{C}\mathbf{C}^\top])}{\sigma_{\mathbf{u}}^2} c_{\text{stab}}^4 C_{\mathbf{r}}^{-1},$$

where $\lambda_{\min}^+(\cdot)$ denotes the smallest positive eigenvalue, we have $\lambda_k(\mathbb{E}_{\mathbf{u}}[(\mathbf{C} + \mathbf{E})(\mathbf{C} + \mathbf{E})^\top]) \geq \frac{1}{2}\lambda_k(\mathbb{E}_{\mathbf{u}}[\mathbf{C}\mathbf{C}^\top])$, $k = 1, \ldots, \text{rank}(\mathbb{E}_{\mathbf{u}}\left[\mathbf{C}\mathbf{C}^\top\right])$ such that

$$\mathbb{E}_{\mathbf{u}}\left[\left(\tilde{\mathbf{x}}_t - \mathbf{x}_t^{\pi^\star}\right)\left(\tilde{\mathbf{x}}_t - \mathbf{x}_t^{\pi^\star}\right)^\top\right] \succeq \frac{1}{2}\mathbb{E}_{\mathbf{u}}\left[\mathbf{C}\mathbf{C}^\top\right]$$

$$= \frac{1}{2} \sum_{s=1}^{t-1} \mathbf{A}_{s+1:t}^{\mathrm{cl}} \mathbf{B}_s \Sigma_{\mathbf{u}} \mathbf{B}_s^{\top} \mathbf{A}_{s+1:t}^{\mathrm{cl}}{}^{\top}.$$

$\square$

Proposition E.2 demonstrates that noise injection yields full-rank exploration around the expert trajectory that is essentially described by the controllability Gramian induced by linearizations around the expert trajectory. In this case, we show that a policy $\hat{\pi}$ attaining low on-expert error does not suffer exponential compounding error. The first ingredient is an adapted result from Pfrommer et al. (2022) that certifies low trajectory error as long as policies are persistently close in a tube around the expert trajectory.

**Proposition E.3** (TaSIL (Pfrommer et al., 2022)). *Assume the closed-loop system induced by $(\pi^{\star}, f)$ is $(C_{\mathrm{ISS}}, \rho)$-EISS. For any (deterministic) policy $\hat{\pi}$ and initial state $\mathbf{x}_1$, let $\mathbf{x}_1^{\hat{\pi}} = \mathbf{x}_1^{\pi^{\star}} = \mathbf{x}_1$, and consider the closed-loop trajectories generated by $\hat{\pi}$ and $\pi^{\star}$:*

$$\mathbf{x}_{t+1}^{\hat{\pi}} = f(\mathbf{x}_t^{\hat{\pi}}, \hat{\pi}(\mathbf{x}_t^{\hat{\pi}})), \quad \mathbf{x}_{t+1}^{\pi^{\star}} = f(\mathbf{x}_t^{\pi^{\star}}, \pi^{\star}(\mathbf{x}_t^{\pi^{\star}})), \quad t \geq 1. \tag{E.5}$$

*Then for any given $\varepsilon > 0$, $T \in \mathbb{N}$, as long as:*

$$\max_{1 \leq t \leq T-1} \sup_{\|\mathbf{w}\| \leq 1} \|(\hat{\pi} - \pi^{\star})(\mathbf{x}_t^{\pi^{\star}} + \varepsilon \mathbf{w})\| \leq c_{\mathrm{stab}} \varepsilon,$$

*we are guaranteed $\max_{1 \leq t \leq T} \|\mathbf{x}_t^{\hat{\pi}} - \mathbf{x}_t^{\pi^{\star}}\| \leq \varepsilon$.*

An elementary proof to Proposition E.3 can be found in e.g., Simchowitz et al. (2025, Lemma I.4). Our next ingredient demonstrates that if noise injection induces full-rank state covariances, closeness in a tube with radius proportional to the noise variance is certified, up to higher-order perturbations from smoothness.

**Lemma E.4.** *Let Assumption 4.1 hold, and let Assumption E.1 hold with $\underline{\lambda}_{\mathbf{z}} > 0$. Let $\{\mathbf{x}_t^{\pi^{\star}}\}_{t=1}^T$, $\{\tilde{\mathbf{x}}_t\}_{t=1}^T$ be expert and noise-injected states initialized from a given $\mathbf{x}_1^{\pi^{\star}} = \tilde{\mathbf{x}}_1$. Let $\hat{\pi}$ be any $C_{\pi}$-smooth (deterministic) policy. For sufficiently small noise-scale $\sigma_{\mathbf{u}} \lesssim \min\left\{ c_{\mathrm{stab}}^3 C_{\mathbf{r}}^{-1} \sqrt{\underline{\lambda}_{\mathbf{z}}}, \ c_{\mathrm{stab}} \sqrt{1 + 4C_{\mathbf{K}}^2} C_{\pi}^{-1} \right\}$, the following holds for each $t = 1, \ldots, T-1$:*

$$\sup_{\|\mathbf{w}\| \leq 1} \|(\hat{\pi} - \pi^{\star})(\mathbf{x}_t^{\pi^{\star}} + \varepsilon \mathbf{w})\|^2 \leq 16 \mathbb{E}_{\tilde{\mathbf{x}}_t | \mathbf{x}_1^{\pi^{\star}}} \left[ \|\hat{\pi}(\tilde{\mathbf{x}}_t) - \pi^{\star}(\tilde{\mathbf{x}}_t)\|^2 \right] + 9 C_{\pi}^2 C_{\mathrm{stab}}^4 \sigma_{\mathbf{u}}^4,$$

*for any $\varepsilon \leq \sigma_{\mathbf{u}} \sqrt{\lambda_{\min}(\mathbf{W}_{1:t}^{\mathbf{z}}(\mathbf{x}_1^{\pi^{\star}}))/2}$.*

*Proof.* Toward upper-bounding the left-hand side of the desired inequality, we have:

$$\sup_{\|\mathbf{w}\| \leq 1} \|(\hat{\pi} - \pi^{\star})(\mathbf{x}_t^{\pi^{\star}} + \varepsilon \mathbf{w})\|^2$$

$$\leq \sup_{\|\mathbf{w}\| \leq 1} 2\|(\hat{\pi} - \pi^{\star})(\mathbf{x}_t^{\pi^{\star}}) + \varepsilon \nabla_{\mathbf{x}}(\hat{\pi} - \pi^{\star})(\mathbf{x}_t^{\pi^{\star}})\mathbf{w}\|^2 + 8 C_{\pi}^2 \varepsilon^4$$

$$\leq 4\|(\hat{\pi} - \pi^{\star})(\mathbf{x}_t^{\pi^{\star}})\|^2 + \sup_{\|\mathbf{w}\| \leq 1} 4\|\nabla_{\mathbf{x}}(\hat{\pi} - \pi^{\star})(\mathbf{x}_t^{\pi^{\star}})\mathbf{w}\|^2 \varepsilon^2 + 8 C_{\pi}^2 \varepsilon^4$$

$$\leq 4\|(\hat{\pi} - \pi^{\star})(\mathbf{x}_t^{\pi^{\star}})\|^2 + 4\|\nabla_{\mathbf{x}}(\hat{\pi} - \pi^{\star})(\mathbf{x}_t^{\pi^{\star}})\|_{\mathrm{op}}^2 \varepsilon^2 + 8 C_{\pi}^2 \varepsilon^4, \tag{E.6}$$

where use the fact that $\hat{\pi} - \pi^{\star}$ is at worst $2C_{\pi}$-smooth, and repeatedly apply $(a + b)^2 \leq 2a^2 + 2b^2$. We now lower bound $\mathbb{E}_{\tilde{\mathbf{x}}_t | \mathbf{x}_1^{\pi^{\star}}} \left[ \|\hat{\pi}(\tilde{\mathbf{x}}_t) - \pi^{\star}(\tilde{\mathbf{x}}_t)\|^2 \right]$. Recall the linear and residual decomposition of $\tilde{\mathbf{x}}_t = \tilde{\mathbf{x}}_t^{\mathrm{lin}} + \tilde{\mathbf{x}}_t^{\mathrm{res}}$ from Proposition E.2. Applying the $C_{\pi}$-smoothness of $\hat{\pi}$ and $\pi^{\star}$, we have:

$$\hat{\pi}(\tilde{\mathbf{x}}_t) - \pi^{\star}(\tilde{\mathbf{x}}_t) = (\hat{\pi} - \pi^{\star})(\mathbf{x}_t^{\pi^{\star}}) + \nabla_{\mathbf{x}}(\hat{\pi} - \pi^{\star})(\mathbf{x}_t^{\pi^{\star}})^{\top}(\tilde{\mathbf{x}}_t^{\mathrm{lin}} - \mathbf{x}_t^{\pi^{\star}} + \tilde{\mathbf{x}}_t^{\mathrm{res}}) + \mathbf{r}_t^{\pi},$$

where $\|\mathbf{r}_t^{\pi}\| \leq 2C_{\pi}\|\tilde{\mathbf{x}}_t - \mathbf{x}_t^{\pi^{\star}}\|^2 \leq 2C_{\pi} C_{\mathrm{stab}}^2 \sigma_{\mathbf{u}}^2$ by applying $C_{\pi}$-smoothness and $(C_{\mathrm{ISS}}, \rho)$-EISS (Definition 2.1) under $\sigma_{\mathbf{u}}$-bounded input perturbations. Therefore, we may lower bound:

$$\|\hat{\pi}(\tilde{\mathbf{x}}_t) - \pi^{\star}(\tilde{\mathbf{x}}_t)\|^2 \geq \frac{1}{2}\|(\hat{\pi} - \pi^{\star})(\mathbf{x}_t^{\pi^{\star}}) + \nabla_{\mathbf{x}}(\hat{\pi} - \pi^{\star})(\mathbf{x}_t^{\pi^{\star}})^{\top}(\tilde{\mathbf{x}}_t^{\mathrm{lin}} - \mathbf{x}_t^{\pi^{\star}})\|^2 - \|\nabla_{\mathbf{x}}(\hat{\pi} - \pi^{\star})(\mathbf{x}_t^{\pi^{\star}})^{\top}\tilde{\mathbf{x}}_t^{\mathrm{res}} + \mathbf{r}_t^{\pi}\|^2$$

$$\geq \frac{1}{2}\|(\hat{\pi} - \pi^\star)(\mathbf{x}_t^{\pi^\star}) + \nabla_\mathbf{x}(\hat{\pi} - \pi^\star)(\mathbf{x}_t^{\pi^\star})^\top(\tilde{\mathbf{x}}_t^{\mathrm{lin}} - \mathbf{x}_t^{\pi^\star})\|^2$$
$$- 2\|\nabla_\mathbf{x}(\hat{\pi} - \pi^\star)(\mathbf{x}_t^{\pi^\star})^\top\tilde{\mathbf{x}}_t^{\mathrm{res}}\|^2 - 8C_\pi^2 C_{\mathrm{stab}}^4\sigma_\mathbf{u}^4.$$

Taking the expectation on both sides, we have:

$$\mathbb{E}_{\tilde{\mathbf{x}}_t|\mathbf{x}_1^{\pi^\star}}\left[\|\hat{\pi}(\tilde{\mathbf{x}}_t) - \pi^\star(\tilde{\mathbf{x}}_t)\|^2\right] \geq \frac{1}{2}\mathbb{E}_{\tilde{\mathbf{x}}_t|\mathbf{x}_1^{\pi^\star}}\left[\|(\hat{\pi} - \pi^\star)(\mathbf{x}_t^{\pi^\star}) + \nabla_\mathbf{x}(\hat{\pi} - \pi^\star)(\mathbf{x}_t^{\pi^\star})^\top(\tilde{\mathbf{x}}_t^{\mathrm{lin}} - \mathbf{x}_t^{\pi^\star})\|^2\right]$$
$$- 2\mathbb{E}_{\tilde{\mathbf{x}}_t|\mathbf{x}_1^{\pi^\star}}\left[\|\nabla_\mathbf{x}(\hat{\pi} - \pi^\star)(\mathbf{x}_t^{\pi^\star})^\top\tilde{\mathbf{x}}_t^{\mathrm{res}}\|^2\right] - 8C_\pi^2 C_{\mathrm{stab}}^4\sigma_\mathbf{u}^4.$$

Notably, $\mathbb{E}_{\tilde{\mathbf{x}}_t|\mathbf{x}_1^{\pi^\star}}\left[\tilde{\mathbf{x}}_t^{\mathrm{lin}} - \mathbf{x}_t^{\pi^\star}\right] = \mathbf{0}$, and thus the first term on the right-hand side can be expanded to yield:

$$\mathbb{E}_{\tilde{\mathbf{x}}_t|\mathbf{x}_1^{\pi^\star}}\left[\|(\hat{\pi} - \pi^\star)(\mathbf{x}_t^{\pi^\star}) + \nabla_\mathbf{x}(\hat{\pi} - \pi^\star)(\mathbf{x}_t^{\pi^\star})^\top(\tilde{\mathbf{x}}_t^{\mathrm{lin}} - \mathbf{x}_t^{\pi^\star})\|^2\right]$$
$$= \mathbb{E}_{\tilde{\mathbf{x}}_t|\mathbf{x}_1^{\pi^\star}}\left[\|(\hat{\pi} - \pi^\star)(\mathbf{x}_t^{\pi^\star})\|^2\right] + \mathrm{tr}\left(\nabla_\mathbf{x}(\hat{\pi} - \pi^\star)(\mathbf{x}_t^{\pi^\star})^\top\mathbb{E}_{\tilde{\mathbf{x}}_t|\mathbf{x}_1^{\pi^\star}}\left[(\tilde{\mathbf{x}}_t^{\mathrm{lin}} - \mathbf{x}_t^{\pi^\star})(\tilde{\mathbf{x}}_t^{\mathrm{lin}} - \mathbf{x}_t^{\pi^\star})^\top\right]\nabla_\mathbf{x}(\hat{\pi} - \pi^\star)(\mathbf{x}_t^{\pi^\star})\right)$$
$$= \mathbb{E}_{\tilde{\mathbf{x}}_t|\mathbf{x}_1^{\pi^\star}}\left[\|(\hat{\pi} - \pi^\star)(\mathbf{x}_t^{\pi^\star})\|^2\right] + \mathrm{tr}\left(\nabla_\mathbf{x}(\hat{\pi} - \pi^\star)(\mathbf{x}_t^{\pi^\star})^\top\mathbf{W}_{1:t}^\mathbf{u}(\mathbf{x}_1^{\pi^\star})\nabla_\mathbf{x}(\hat{\pi} - \pi^\star)(\mathbf{x}_t^{\pi^\star})\right),$$

On the other hand, expanding the second term yields:

$$\mathbb{E}_{\tilde{\mathbf{x}}_t|\mathbf{x}_1^{\pi^\star}}\left[\|\nabla_\mathbf{x}(\hat{\pi} - \pi^\star)(\mathbf{x}_t^{\pi^\star})^\top\tilde{\mathbf{x}}_t^{\mathrm{res}}\|^2\right] = \mathrm{tr}\left(\nabla_\mathbf{x}(\hat{\pi} - \pi^\star)(\mathbf{x}_t^{\pi^\star})^\top\mathbb{E}_{\tilde{\mathbf{x}}_t|\mathbf{x}_1^{\pi^\star}}\left[\tilde{\mathbf{x}}_t^{\mathrm{res}}\tilde{\mathbf{x}}_t^{\mathrm{res}\top}\right]\nabla_\mathbf{x}(\hat{\pi} - \pi^\star)(\mathbf{x}_t^{\pi^\star})\right)$$
$$\leq \mathrm{tr}\left(\nabla_\mathbf{x}(\hat{\pi} - \pi^\star)(\mathbf{x}_t^{\pi^\star})^\top\nabla_\mathbf{x}(\hat{\pi} - \pi^\star)(\mathbf{x}_t^{\pi^\star})\right)C_{\mathrm{stab}}^6 C_\mathbf{r}^2\sigma_\mathbf{u}^4,$$

where we applied Proposition E.2 for the second line. Therefore, for sufficiently small noise level:

$$\sigma_\mathbf{u}^2 \leq \frac{1}{8}c_{\mathrm{stab}}^6 C_\mathbf{r}^{-2}\lambda_{\min}(\mathbf{W}_{1:t}^\mathbf{z}(\mathbf{x}_1^{\pi^\star})),$$

we may combine the first and second terms to yield:

$$\mathbb{E}_{\tilde{\mathbf{x}}_t|\mathbf{x}_1^{\pi^\star}}\left[\|\hat{\pi}(\tilde{\mathbf{x}}_t) - \pi^\star(\tilde{\mathbf{x}}_t)\|^2\right]$$
$$\geq \frac{1}{2}\mathbb{E}_{\tilde{\mathbf{x}}_t|\mathbf{x}_1^{\pi^\star}}\left[\|(\hat{\pi} - \pi^\star)(\mathbf{x}_t^{\pi^\star})\|^2\right] + \frac{1}{4}\mathrm{tr}\left(\nabla_\mathbf{x}(\hat{\pi} - \pi^\star)(\mathbf{x}_t^{\pi^\star})^\top\mathbf{W}_{1:t}^\mathbf{u}(\mathbf{x}_1^{\pi^\star})\nabla_\mathbf{x}(\hat{\pi} - \pi^\star)(\mathbf{x}_t^{\pi^\star})\right) - 8C_\pi^2 C_{\mathrm{stab}}^4\sigma_\mathbf{u}^4$$
$$\geq \frac{1}{2}\left(\mathbb{E}_{\tilde{\mathbf{x}}_t|\mathbf{x}_1^{\pi^\star}}\left[\|(\hat{\pi} - \pi^\star)(\mathbf{x}_t^{\pi^\star})\|^2\right] + \frac{1}{2}\lambda_{\min}(\mathbf{W}_{1:t}^\mathbf{u}(\mathbf{x}_1^{\pi^\star}))\|\nabla_\mathbf{x}(\hat{\pi} - \pi^\star)(\mathbf{x}_t^{\pi^\star})\|_{\mathrm{op}}^2\right) - 8C_\pi^2 C_{\mathrm{stab}}^4\sigma_\mathbf{u}^4,$$

where we used the elementary inequalities $\mathrm{tr}(\mathbf{PQ}) \geq \lambda_{\min}(\mathbf{P})\mathrm{tr}(\mathbf{Q}) \geq \lambda_{\min}(\mathbf{P})\lambda_{\max}(\mathbf{Q})$, for any $\mathbf{P} \succ \mathbf{0}, \mathbf{Q} \succeq \mathbf{0}$. Notably, the validity of this inequality rests on $\mathbf{P} \triangleq \mathbf{W}_{1:t}^\mathbf{u}(\mathbf{x}_1^{\pi^\star}) \succ \mathbf{0}$ granted by Assumption E.1. Rearranging (E.6) yields:

$$\|(\hat{\pi} - \pi^\star)(\mathbf{x}_t^{\pi^\star})\|^2 + \|\nabla_\mathbf{x}(\hat{\pi} - \pi^\star)(\mathbf{x}_t^{\pi^\star})\|_{\mathrm{op}}^2\varepsilon^2 \geq \frac{1}{4}\sup_{\|\mathbf{w}\|\leq 1}\|(\hat{\pi} - \pi^\star)(\mathbf{x}_t^{\pi^\star} + \varepsilon\mathbf{w})\|^2 - 2C_\pi^2\varepsilon^4.$$

For $\varepsilon^2 \leq \frac{1}{2}\lambda_{\min}(\mathbf{W}_{1:t}^\mathbf{u}(\mathbf{x}_1^{\pi^\star})) = \frac{\sigma_\mathbf{u}^2}{2}\lambda_{\min}(\mathbf{W}_{1:t}^\mathbf{z}(\mathbf{x}_1^{\pi^\star}))$, plugging this into the above sequence of inequalities yields:

$$\mathbb{E}_{\tilde{\mathbf{x}}_t|\mathbf{x}_1^{\pi^\star}}\left[\|\hat{\pi}(\tilde{\mathbf{x}}_t) - \pi^\star(\tilde{\mathbf{x}}_t)\|^2\right] \geq \frac{1}{2}\left(\frac{1}{4}\sup_{\|\mathbf{w}\|\leq 1}\|(\hat{\pi} - \pi^\star)(\mathbf{x}_t^{\pi^\star} + \varepsilon\mathbf{w})\|^2 - 2C_\pi^2\varepsilon^4\right) - 8C_\pi^2 C_{\mathrm{stab}}^4\sigma_\mathbf{u}^4$$
$$\geq \frac{1}{16}\sup_{\|\mathbf{w}\|\leq 1}\|(\hat{\pi} - \pi^\star)(\mathbf{x}_t^{\pi^\star} + \varepsilon\mathbf{w})\|^2 - \frac{1}{2}C_\pi^2\varepsilon^4 - 8C_\pi^2 C_{\mathrm{stab}}^4\sigma_\mathbf{u}^4.$$

We have trivially that $\varepsilon^2 \leq \frac{1}{2}\lambda_{\min}(\mathbf{W}_{1:t}^\mathbf{u}(\mathbf{x}_1^{\pi^\star})) \leq \frac{\sigma_\mathbf{u}^2}{2}C_{\mathrm{stab}}^2$, and thus rearranging the inequality yields the desired inequality:

$$\sup_{\|\mathbf{w}\|\leq 1}\|(\hat{\pi} - \pi^\star)(\mathbf{x}_t^{\pi^\star} + \varepsilon\mathbf{w})\|^2 \leq 16\mathbb{E}_{\tilde{\mathbf{x}}_t|\mathbf{x}_1^{\pi^\star}}\left[\|\hat{\pi}(\tilde{\mathbf{x}}_t) - \pi^\star(\tilde{\mathbf{x}}_t)\|^2\right] + 9C_\pi^2 C_{\mathrm{stab}}^4\sigma_\mathbf{u}^4.$$

$\square$

Therefore, using Lemma E.4 to certify the tube condition in Proposition E.3 yields the (suboptimal) imitation guarantee.

**Suboptimal Proposition 4.2.** Let Assumption 4.1 hold, and let $\mathbf{W}_{1:t}^{\mathbf{u}}(\mathbf{x}_1^{\pi^\star}) \succeq \underline{\lambda}_{\mathbf{W}} \mathbf{I}_{d_x}$, $t \geq 2$ w.p. 1 over $\mathbf{x}_1^{\pi^\star} \sim D$ for some $\underline{\lambda}_{\mathbf{W}} > 0$. Let $\hat{\pi}$ be a $C_\pi$-smooth candidate policy. For $\sigma_{\mathbf{u}}^2$ that satisfies $\sigma_{\mathbf{u}}^2 \lesssim O_\star(\text{poly}(1/C_\pi, 1/C_{\text{reg}})) \underline{\lambda}_{\mathbf{W}}$, we have:

$$\mathbf{J}_{\text{TRAJ},T}(\hat{\pi}) \lesssim O_\star(T) \underline{\lambda}_{\mathbf{W}}^{-1} \left( \frac{1}{\sigma_{\mathbf{u}}^2} \mathbf{J}_{\text{DEMO},T}(\hat{\pi}; \mathbb{P}_{\pi^\star, \sigma_{\mathbf{u}}}) + C_\pi^2 C_{\text{stab}}^2 \sigma_{\mathbf{u}}^2 \right).$$

*Proof of Suboptimal Proposition 4.2.* Using the identity for non-negative random variable $Z$ supported on $[0,1]$, $\mathbb{E}[Z] = \int_0^1 \mathbb{P}[Z > \varepsilon] \, \mathrm{d}\varepsilon$, we have:

$$\mathbb{E}_{\mathbf{x}_1 \sim \mathbb{P}_{\mathbf{x}_1^{\pi^\star}}} \left[ \max_{1 \leq t \leq T} \|\mathbf{x}_t^{\hat{\pi}} - \mathbf{x}_t^{\pi^\star}\|^2 \wedge 1 \right] = \int_0^1 \mathbb{P} \left[ \max_{1 \leq t \leq T} \|\mathbf{x}_t^{\hat{\pi}} - \mathbf{x}_t^{\pi^\star}\|^2 > \varepsilon \right] \, \mathrm{d}\varepsilon$$

$$= \int_0^\tau \mathbb{P} \left[ \max_{1 \leq t \leq T} \|\mathbf{x}_t^{\hat{\pi}} - \mathbf{x}_t^{\pi^\star}\|^2 > \varepsilon \right] \, \mathrm{d}\varepsilon + \int_\tau^1 \mathbb{P} \left[ \max_{1 \leq t \leq T} \|\mathbf{x}_t^{\hat{\pi}} - \mathbf{x}_t^{\pi^\star}\|^2 > \varepsilon \right] \, \mathrm{d}\varepsilon$$

$$\leq \int_0^\tau \mathbb{P} \left[ \max_{1 \leq t \leq T} \|\mathbf{x}_t^{\hat{\pi}} - \mathbf{x}_t^{\pi^\star}\|^2 > \varepsilon \right] \, \mathrm{d}\varepsilon + \mathbb{P} \left[ \max_{1 \leq t \leq T} \|\mathbf{x}_t^{\hat{\pi}} - \mathbf{x}_t^{\pi^\star}\|^2 > \tau \right]$$

where we choose a splitting point $\tau \in [0,1]$ to be determined later. Now, applying Proposition E.3 yields:

$$\int_0^\tau \mathbb{P} \left[ \max_{1 \leq t \leq T} \|\mathbf{x}_t^{\hat{\pi}} - \mathbf{x}_t^{\pi^\star}\|^2 > \varepsilon \right] \, \mathrm{d}\varepsilon + \mathbb{P} \left[ \max_{1 \leq t \leq T} \|\mathbf{x}_t^{\hat{\pi}} - \mathbf{x}_t^{\pi^\star}\|^2 > \tau \right]$$

$$\leq \int_0^\tau \mathbb{P} \left[ \max_{1 \leq t \leq T} \sup_{\|\mathbf{w}\| \leq 1} \|(\hat{\pi} - \pi^\star)(\mathbf{x}_t^{\pi^\star} + \sqrt{\varepsilon}\mathbf{w})\|^2 > c_{\text{stab}}^2 \varepsilon \right] \, \mathrm{d}\varepsilon$$

$$+ \mathbb{P} \left[ \max_{1 \leq t \leq T} \sup_{\|\mathbf{w}\| \leq 1} \|(\hat{\pi} - \pi^\star)(\mathbf{x}_t^{\pi^\star} + \sqrt{\tau}\mathbf{w})\|^2 > c_{\text{stab}}^2 \tau \right].$$

For the first term, we have:

$$\int_0^\tau \mathbb{P} \left[ \max_{1 \leq t \leq T} \sup_{\|\mathbf{w}\| \leq 1} \|(\hat{\pi} - \pi^\star)(\mathbf{x}_t^{\pi^\star} + \sqrt{\varepsilon}\mathbf{w})\|^2 > c_{\text{stab}}^2 \varepsilon \right] \, \mathrm{d}\varepsilon$$

$$\leq \int_0^\tau \mathbb{P} \left[ \max_{1 \leq t \leq T} \sup_{\|\mathbf{w}\| \leq 1} \|(\hat{\pi} - \pi^\star)(\mathbf{x}_t^{\pi^\star} + \sqrt{\tau}\mathbf{w})\|^2 > c_{\text{stab}}^2 \varepsilon \right] \, \mathrm{d}\varepsilon$$

$$\leq \min \left\{ \tau, C_{\text{stab}}^2 \mathbb{E}_{\mathbf{x}_1} \left[ \max_{1 \leq t \leq T} \sup_{\|\mathbf{w}\| \leq 1} \|(\hat{\pi} - \pi^\star)(\mathbf{x}_t^{\pi^\star} + \tau\mathbf{w})\|^2 \right] \right\},$$

where the last line arises from combining the trivial bound $\int_0^\tau \mathbb{P}[Z > \varepsilon] \, \mathrm{d}\varepsilon \leq \tau$ and by performing the variable substitution $\varepsilon' = c_{\text{stab}}^2 \varepsilon$, then applying the identity $\mathbb{E}[Z] = \int_0^1 \mathbb{P}[Z > \varepsilon'] \, \mathrm{d}\varepsilon'$. Therefore, setting $\tau \leq \tilde{\sigma}_{\mathbf{u}}^2$, we apply Lemma E.4 to get:

$$\int_0^\tau \mathbb{P} \left[ \max_{1 \leq t \leq T} \sup_{\|\mathbf{w}\| \leq 1} \|(\hat{\pi} - \pi^\star)(\mathbf{x}_t^{\pi^\star} + \varepsilon\mathbf{w})\|^2 > \frac{1 - \rho}{C_{\text{ISS}}} \varepsilon \right] \, \mathrm{d}\varepsilon$$

$$\leq \min \left\{ \tau, C_{\text{stab}}^2 \mathbb{E}_{\mathbf{x}_1} \left[ \max_{1 \leq t \leq T} 16 \mathbb{E}_{\tilde{\mathbf{x}}_t | \mathbf{x}_1^{\pi^\star}} \left[ \|\hat{\pi}(\tilde{\mathbf{x}}_t) - \pi^\star(\tilde{\mathbf{x}}_t)\|^2 \right] + \overline{C} C_\pi^2 \sigma_{\mathbf{u}}^4 \right] \right\}$$

$$\leq \min \left\{ \tau, C_{\text{stab}}^2 \overline{C} C_\pi^2 \sigma_{\mathbf{u}}^4 + 16 C_{\text{stab}}^2 \sum_{t=1}^T \mathbb{E}_{\mathbf{x}_t} \left[ \|\hat{\pi}(\tilde{\mathbf{x}}_t) - \pi^\star(\tilde{\mathbf{x}}_t)\|^2 \right] \right\}.$$

For the second term, we apply Markov's inequality and similarly bound:

$$\mathbb{P} \left[ \max_{1 \leq t \leq T} \sup_{\|\mathbf{w}\| \leq 1} \|(\hat{\pi} - \pi^\star)(\mathbf{x}_t^{\pi^\star} + \sqrt{\tau}\mathbf{w})\|^2 > c_{\text{stab}}^2 \tau \right]$$

$$\leq C_{\text{stab}}^2 \tau^{-1} \mathbb{E}_{\mathbf{x}_1} \left[ \max_{1 \leq t \leq T} 16 \mathbb{E}_{\tilde{\mathbf{x}}_t | \mathbf{x}_1^{\pi^\star}} \left[ \| \hat{\pi}(\tilde{\mathbf{x}}_t) - \pi^\star(\tilde{\mathbf{x}}_t) \|^2 \right] + 9 C_\pi^2 C_{\text{stab}}^4 \sigma_{\mathbf{u}}^4 \right]$$

$$\leq C_{\text{stab}}^2 \tau^{-1} \left( 9 C_\pi^2 C_{\text{stab}}^4 \sigma_{\mathbf{u}}^4 + 16 \sum_{t=1}^{T} \mathbb{E}_{\mathbf{x}_t} \left[ \| \hat{\pi}(\tilde{\mathbf{x}}_t) - \pi^\star(\tilde{\mathbf{x}}_t) \|^2 \right] \right).$$

Combining the two bounds and setting $\tau = \tilde{\sigma}_{\mathbf{u}}^2$ yields a bound on $\mathbb{E}_{\mathbf{x}_1 \sim \mathbb{P}_{\mathbf{x}_1^{\pi^\star}}} \left[ \max_{1 \leq t \leq T} \| \mathbf{x}_t^{\hat{\pi}} - \mathbf{x}_t^{\pi^\star} \|^2 \wedge 1 \right]$ in terms of $\mathbf{J}_{\text{DEMO},2,T}(\hat{\pi}; \mathbb{P}_{\pi^\star, \sigma_{\mathbf{u}}})$ and an additive drift term. By summing over each $1 \leq t \leq T$, we get a bound on $\mathbf{J}_{\text{TRAJ},2,T}^{\mathbf{x}}(\hat{\pi})$, accruing a $T$ factor. Now, by Lemma D.1, we have:

$$\mathbf{J}_{\text{TRAJ},2,T}(\hat{\pi}) \leq \left( 1 + 4 L_\pi^2 \right) \mathbf{J}_{\text{TRAJ},2,T}^{\mathbf{x}}(\hat{\pi}) + 4 \mathbf{J}_{\text{DEMO},2,T}(\hat{\pi}; \mathbb{P}_{\pi^\star}).$$

It remains to relate $\mathbf{J}_{\text{DEMO},2,T}(\hat{\pi}; \mathbb{P}_{\pi^\star})$ to $\mathbf{J}_{\text{DEMO},2,T}(\hat{\pi}; \mathbb{P}_{\pi^\star, \sigma_{\mathbf{u}}})$. Since the injected noise is by definition $\sigma_{\mathbf{u}}$-bounded, applying $(C_{\text{ISS}}, \rho)$-EISS of $(\pi^\star, f)$ yields w.p. 1 over any $\mathbf{x}_1^{\pi^\star}$ and $\{\mathbf{z}\} \overset{\text{i.i.d}}{\sim} \mathcal{D}(\mathbf{0}, \Sigma_{\mathbf{z}})$:

$$\| \tilde{\mathbf{x}}_t - \mathbf{x}_t^{\pi^\star} \| \leq C_{\text{ISS}} \sum_{s=1}^{t-1} \rho^{t-1-s} \| \sigma_{\mathbf{u}} \mathbf{z}_s \|$$

$$\leq C_{\text{stab}} \sigma_{\mathbf{u}}.$$

In other words, for a given $\mathbf{z}_t \sim \mathcal{D}(\mathbf{0}, \Sigma_{\mathbf{z}})$ we always have:

$$\| (\hat{\pi} - \pi^\star)(\mathbf{x}_t^{\pi^\star}) \| \leq \| (\hat{\pi} - \pi^\star)(\mathbf{x}_t^{\pi^\star} + \sigma_{\mathbf{u}} \mathbf{z}_t) \| + 2 L_\pi C_{\text{stab}} \sigma_{\mathbf{u}}.$$

Squaring both sides and taking an expectation yields the following bound on $\mathbf{J}_{\text{DEMO},2,T}(\hat{\pi}; \mathbb{P}_{\pi^\star})$:

$$\mathbf{J}_{\text{DEMO},2,T}(\hat{\pi}; \mathbb{P}_{\pi^\star}) \lesssim \mathbf{J}_{\text{DEMO},2,T}(\hat{\pi}; \mathbb{P}_{\pi^\star, \sigma_{\mathbf{u}}}) + T L_\pi^2 C_{\text{stab}}^2 \sigma_{\mathbf{u}}^2.$$

Putting the pieces together, we have:

$$\mathbf{J}_{\text{TRAJ},2,T}(\hat{\pi}) \leq \left( 1 + 4 L_\pi^2 \right) \mathbf{J}_{\text{TRAJ},2,T}^{\mathbf{x}}(\hat{\pi}) + 4 \mathbf{J}_{\text{DEMO},2,T}(\hat{\pi}; \mathbb{P}_{\pi^\star})$$

$$\lesssim \left( 1 + 4 L_\pi^2 \right) C_{\text{stab}}^2 \underline{\lambda}_{\mathbf{z}}^{-1} T \left( \frac{1}{\sigma_{\mathbf{u}}^2} \mathbf{J}_{\text{DEMO},2,T}(\hat{\pi}; \mathbb{P}_{\pi^\star, \sigma_{\mathbf{u}}}) + C_\pi^2 C_{\text{stab}}^4 \sigma_{\mathbf{u}}^2 \right)$$

$$+ \mathbf{J}_{\text{DEMO},2,T}(\hat{\pi}; \mathbb{P}_{\pi^\star, \sigma_{\mathbf{u}}}) + T L_\pi^2 C_{\text{stab}}^2 \sigma_{\mathbf{u}}^2.$$

When $\mathcal{D}(\mathbf{0}, \Sigma_{\mathbf{z}})$ is the uniform distribution over the ball, we have $\underline{\lambda}_{\mathbf{z}} \approx \underline{\lambda}_{\mathbf{W}}/d_u$. Lumping terms together, this completes the proof of Suboptimal Proposition 4.2.

$\square$

This result says that if noise injection fully excites the state space, then the trajectory error is bounded by the on-expert error evaluated on the noise-injected law $\mathbb{P}_{\pi^\star, \sigma_{\mathbf{u}}}$ plus a higher-order error term from smoothness. Note that simply regressing on the expert trajectories without noise injection, even the smooth one-step controllable case considered here, can suffer from exponential compounding error (see Simchowitz et al. (2025, Theorem 4)). Though this is a marked improvement upon vanilla behavior cloning, this set-up leaves open a couple deficiencies. Firstly, performing behavior cloning on $\mathbb{P}_{\pi^\star, \sigma_{\mathbf{u}}}$ yields a drift term $\approx \sigma_{\mathbf{u}}^2$ that persists even when $\mathbf{J}_{\text{DEMO},T}(\hat{\pi}; \mathbb{P}_{\pi^\star, \sigma_{\mathbf{u}}})$ is small; this introduces a trade-off on the noise-scale, where larger $\sigma_{\mathbf{u}}$ benefits the excitation, but exacerbates the drift. We demonstrate in Appendix E.7 that this additive factor is fundamental. Secondly, one-step controllability–and in a similar vein persistency of excitation–is a strong condition (e.g. requires $d_u = d_x$); typically we do not expect inputs to be able to excite every mode in a system, let alone instantaneously.

## E.4 DEPARTING FROM CONTROLLABILITY AND PERSISTENCY OF EXCITATION

We now consider the case where we lack controllability, one-step or otherwise. In other words, the linear controllability Gramians need not be full-rank: $\text{rank}(\mathbf{W}_{1:t}^{\mathbf{u}}(\mathbf{x}_1^{\pi^\star})) < d_x$. Furthermore, as promised in the body, we hope to lift the inverse dependence on the smallest positive eigenvalue of controllability Gramian, including when it is rank-deficient. On the technical front, a few barriers

are present. Firstly, the state-covariance bound in Corollary E.1 imposes a constraint on $\sigma_{\mathbf{u}}$ scaling with the smallest positive eigenvalue of $\mathbf{W}_{1:t}^{\mathbf{z}}$—this can be exponentially small in $d_x$ in various cases. Secondly, Proposition E.3 requires certifying that $\hat{\pi}$ and $\pi^\star$ match on a (full-dimensional) ball around the expert trajectory, and subsequently the "expectation-to-uniform" bound in Lemma E.4 requires a full-rank covariance.

Given these technical difficulties, we introduce the notion of the "reachable subspace" under the *linearized* system under the expert.

**Definition E.1.** Fix any $\mathbf{x}_1^{\pi^\star} \sim D$. Recall the expert linearizations from Eq. (E.1). Define the *reachable subspace* of the expert closed-loop system at time $t$:

$$\mathcal{R}_t^{\pi^\star} \triangleq \left\{ \sum_{s=1}^{t-1} \mathbf{A}_{s+1:t}^{\mathrm{cl}} \mathbf{B}_{ts} \mathbf{u}_s \;\Big|\; \{\mathbf{u}_s\}_{s=1}^{t-1} \subset \mathbb{R}^{d_u} \right\}.$$

The following facts hold:

- $\mathcal{R}_t^{\pi^\star}$ is a linear subspace of $\mathbb{R}^{d_x}$.

- Given any positive-definite $\Sigma \succ 0$, the associated controllability Gramian satisfies $\mathrm{rank}\left( \sum_{s=1}^{t-1} \mathbf{A}_{s+1:t}^{\mathrm{cl}} \mathbf{B}_s \Sigma \mathbf{B}_s^\top \mathbf{A}_{s+1:t}^{\mathrm{cl}}{}^\top \right) = \dim(\mathcal{R}_t^{\pi^\star})$ for each $t \geq 1$.

Let $\{(\lambda_{i,t}, \mathbf{v}_{i,t})\}_{i=1}^{d_x}$ be the eigenvalues and vectors of $\mathbf{W}_{1:t}^{\mathbf{u}}$, $t \geq 2$.[11] Let us further define the reachable subspace *truncated at* $\lambda$:

$$\mathcal{R}_t^{\pi^\star}(\lambda) \triangleq \mathrm{span}\{\mathbf{v}_{i,t} : \lambda_{i,t} \geq \lambda\},$$

as well as the corresponding orthogonal projection matrix $\mathcal{P}_{\mathcal{R}_t^{\pi^\star}(\lambda)}$. We also abuse notation and denote $\mathcal{R}_t^{\pi^\star}(\lambda)^\perp$ as the subspace component of $\mathcal{R}_t^{\pi^\star}$ orthogonal to $\mathcal{R}_t^{\pi^\star}(\lambda)$.

In line with the body, we will consider $\mathcal{D}(\mathbf{0}, \Sigma_{\mathbf{z}}) = \mathrm{Unif}(\mathbb{B}^{d_u}(1))$, such that $\mathbf{W}_{1:t}^{\mathbf{z}} \succeq \frac{1}{3d_u} \mathbf{W}_{1:t}^{\mathbf{u}}$. As previewed in the body, the main guiding intuition moving forward is as follows: 1. by smoothness of the dynamics, most of the error should be contained in the (linearized) reachable subspace, 2. the small eigendirections of the controllability Gramian are precisely those that are hard-to-excite, and thus should accumulate compounding errors slowly enough to "ignore" them. We start by proving a restricted "Jacobian sketching" result (cf. Proposition 4.4). We note that though we present Proposition 4.3 first in the body, we will in fact use an extended version of it that relies on the subsequent result.

**Proposition E.5** (Full ver. of Proposition 4.4). *Let Assumption 4.1 hold. For $\mathbf{x}_1^{\pi^\star} \sim D$, define $\mathcal{R}_t^{\pi^\star}(\lambda) \triangleq \mathrm{span}\{\mathbf{v}_{i,t} : \lambda_{i,t} \geq \lambda\}$ and $\mathcal{P}_{\mathcal{R}_t^{\pi^\star}(\lambda)}$ as in Definition E.1, for some $\lambda \geq \lambda_{\min}^+(\mathbf{W}_{1:t}^{\mathbf{u}}(\mathbf{x}_1^{\pi^\star}))$. Then, for $\sigma_{\mathbf{u}}$ satisfying:*

$$\sigma_{\mathbf{u}} \lesssim \min\left\{ \lambda d_u^{-1} c_{\mathrm{stab}}^4 C_{\mathbf{r}}^{-1}, \; c_{\mathrm{stab}} \frac{\sqrt{1 + 4C_{\mathbf{K}}^2}}{C_\pi} \right\} = O_\star(\lambda).$$

*we have the following bound for each $t \geq 2$:*

$$\|\mathcal{P}_{\mathcal{R}_t^{\pi^\star}(\lambda)} \nabla_{\mathbf{x}}(\hat{\pi} - \pi^\star)(\mathbf{x}_t^{\pi^\star})\|_{\mathrm{op}}^2 \lesssim \frac{d_u}{\sigma_{\mathbf{u}}^2 \lambda} \left( \|(\hat{\pi} - \pi^\star)(\mathbf{x}_t^{\pi^\star})\|^2 + \mathbb{E}_{\tilde{\mathbf{x}}_t | \mathbf{x}_1^{\pi^\star}} \|(\hat{\pi} - \pi^\star)(\tilde{\mathbf{x}}_t)\|^2 \right) + \frac{d_u \sigma_{\mathbf{u}}^2}{\lambda} C_\pi^2 C_{\mathrm{stab}}^4.$$

We note that Proposition 4.4 is recovered by applying an expectation over $\mathbf{x}_1^{\pi^\star}$ on both sides of the inequality.

*Proof of Proposition E.5.* First, we consider the following adaptation of Corollary E.1

**Corollary E.2.** *Let Assumption 4.1 hold and $C_{\mathbf{r}}$ be as defined in Proposition E.2. Fix any $t \geq 2$. For $\lambda \geq \lambda_{\min}^+(\mathbf{W}_{1:t}^{\mathbf{u}}(\mathbf{x}_1^{\pi^\star}))$, set $\mathbf{P} = \mathcal{P}_{\mathcal{R}_t^{\pi^\star}(\lambda)}$ as in Definition E.1. As long as:*

$$\sigma_{\mathbf{u}} \lesssim \min\left\{ \lambda d_u^{-1} c_{\mathrm{stab}}^4 C_{\mathbf{r}}^{-1}, \; c_{\mathrm{stab}} \frac{\sqrt{1 + 4C_{\mathbf{K}}^2}}{C_\pi} \right\},$$

---

[11]Though we omit it for clarity, recall all these quantities implicitly condition on $\mathbf{x}_1^{\pi^\star}$.

*the following holds almost surely over* $\tilde{\mathbf{x}}_1 = \mathbf{x}_1^{\pi^\star} \sim D$ *and* $\{\mathbf{z}_s\} \stackrel{\text{i.i.d}}{\sim} \mathcal{D}(\mathbf{0}, \Sigma_{\mathbf{z}})$:

$$\mathbb{E}_{\tilde{\mathbf{x}}_t | \mathbf{x}_1^{\pi^\star}} \left[ \mathbf{P} \left( \tilde{\mathbf{x}}_t - \mathbf{x}_t^{\pi^\star} \right) \left( \tilde{\mathbf{x}}_t - \mathbf{x}_t^{\pi^\star} \right)^\top \mathbf{P} \right] \succeq \frac{\sigma_{\mathbf{u}}^2}{2} \mathbf{P} \mathbf{W}_{1:t}^{\mathbf{z}} (\mathbf{x}_1^{\pi^\star}) \mathbf{P}. \tag{E.7}$$

The proof of Corollary E.2 follows from a one-line modification in the proof of Corollary E.1, where instead of requiring Weyl's inequality to hold over all positive eigenvalues $k = 1, \ldots, \text{rank}(\mathbf{W}_{1:t}^{\mathbf{u}})$, we need only to consider up to $k = 1, \ldots, p$, $p = \dim(\mathcal{R}_t^{\pi^\star}(\lambda))$, for which $\lambda_p(\mathbf{W}_{1:t}^{\mathbf{z}}) \gtrsim d_u^{-1} \lambda_p(\mathbf{W}_{1:t}^{\mathbf{u}}) \geq d_u^{-1} \lambda$.

We proceed by applying the $C_\pi$-smoothness of $\hat{\pi}$ and $\pi^\star$, we have:

$$\hat{\pi}(\tilde{\mathbf{x}}_t) - \pi^\star(\tilde{\mathbf{x}}_t) = (\hat{\pi} - \pi^\star)(\mathbf{x}_t^{\pi^\star}) + \nabla_{\mathbf{x}} (\hat{\pi} - \pi^\star)(\mathbf{x}_t^{\pi^\star})^\top (\tilde{\mathbf{x}}_t^{\text{lin}} - \mathbf{x}_t^{\pi^\star} + \tilde{\mathbf{x}}_t^{\text{res}}) + \mathbf{r}_t^\pi,$$

where $\|\mathbf{r}_t^\pi\| \leq 2C_\pi \|\tilde{\mathbf{x}}_t - \mathbf{x}_t^{\pi^\star}\|^2 \leq 2C_\pi C_{\text{stab}}^2 \sigma_{\mathbf{u}}^2$ by applying $C_\pi$-smoothness and $(C_{\text{ISS}}, \rho)$-EISS (Definition 2.1) under $\sigma_{\mathbf{u}}$-bounded input perturbations. Therefore, we may lower bound:

$$\|\hat{\pi}(\tilde{\mathbf{x}}_t) - \pi^\star(\tilde{\mathbf{x}}_t)\| \geq \|\nabla_{\mathbf{x}}(\hat{\pi} - \pi^\star)(\mathbf{x}_t^{\pi^\star})^\top (\tilde{\mathbf{x}}_t - \mathbf{x}_t^{\pi^\star})\| - \|(\hat{\pi} - \pi^\star)(\mathbf{x}_t^{\pi^\star}) + \mathbf{r}_t^\pi\|$$
$$\geq \|\nabla_{\mathbf{x}}(\hat{\pi} - \pi^\star)(\mathbf{x}_t^{\pi^\star})^\top (\tilde{\mathbf{x}}_t - \mathbf{x}_t^{\pi^\star})\| - \|(\hat{\pi} - \pi^\star)(\mathbf{x}_t^{\pi^\star})\| - 2C_\pi C_{\text{stab}}^2 \sigma_{\mathbf{u}}^2.$$

Rearranging the above inequality, squaring both sides, and applying the inequality $(a + b + c)^2 \leq 3(a^2 + b^2 + c^2)$ we have:

$$\|\nabla_{\mathbf{x}}(\hat{\pi} - \pi^\star)(\mathbf{x}_t^{\pi^\star})^\top (\tilde{\mathbf{x}}_t - \mathbf{x}_t^{\pi^\star})\|^2 \leq 3 \left( \|(\hat{\pi} - \pi^\star)(\mathbf{x}_t^{\pi^\star})\|^2 + \|\hat{\pi}(\tilde{\mathbf{x}}_t) - \pi^\star(\tilde{\mathbf{x}}_t)\|^2 + 2C_\pi^2 C_{\text{stab}}^4 \sigma_{\mathbf{u}}^4 \right).$$

Taking an expectation over the noise injection on both sides, we may apply Corollary E.2 on the left-hand side: for $\sigma_{\mathbf{u}}$ satisfying the requirements therein, we have:

$$\mathbb{E}_{\tilde{\mathbf{x}}_t | \mathbf{x}_1^{\pi^\star}} \left[ \|\nabla_{\mathbf{x}}(\hat{\pi} - \pi^\star)(\mathbf{x}_t^{\pi^\star})^\top (\tilde{\mathbf{x}}_t - \mathbf{x}_t^{\pi^\star})\|^2 \right]$$
$$= \text{tr} \left( \nabla_{\mathbf{x}}(\hat{\pi} - \pi^\star)(\mathbf{x}_t^{\pi^\star})^\top \mathbb{E}_{\tilde{\mathbf{x}}_t | \mathbf{x}_1^{\pi^\star}} \left[ (\tilde{\mathbf{x}}_t - \mathbf{x}_t^{\pi^\star})(\tilde{\mathbf{x}}_t - \mathbf{x}_t^{\pi^\star})^\top \right] \nabla_{\mathbf{x}}(\hat{\pi} - \pi^\star)(\mathbf{x}_t^{\pi^\star}) \right)$$
$$\geq \text{tr} \left( \nabla_{\mathbf{x}}(\hat{\pi} - \pi^\star)(\mathbf{x}_t^{\pi^\star})^\top \mathcal{P}_{\mathcal{R}_t^{\pi^\star}(\lambda)} \mathbb{E}_{\tilde{\mathbf{x}}_t | \mathbf{x}_1^{\pi^\star}} \left[ (\tilde{\mathbf{x}}_t - \mathbf{x}_t^{\pi^\star})(\tilde{\mathbf{x}}_t - \mathbf{x}_t^{\pi^\star})^\top \right] \mathcal{P}_{\mathcal{R}_t^{\pi^\star}(\lambda)} \nabla_{\mathbf{x}}(\hat{\pi} - \pi^\star)(\mathbf{x}_t^{\pi^\star}) \right)$$
$$\geq \frac{\sigma_{\mathbf{u}}^2}{2} \text{tr} \left( \nabla_{\mathbf{x}}(\hat{\pi} - \pi^\star)(\mathbf{x}_t^{\pi^\star})^\top \mathcal{P}_{\mathcal{R}_t^{\pi^\star}(\lambda)} \mathbf{W}_{1:t}^{\mathbf{z}}(\mathbf{x}_1^{\pi^\star}) \mathcal{P}_{\mathcal{R}_t^{\pi^\star}(\lambda)} \nabla_{\mathbf{x}}(\hat{\pi} - \pi^\star)(\mathbf{x}_t^{\pi^\star}) \right)$$
$$\gtrsim \sigma_{\mathbf{u}}^2 \|\mathcal{P}_{\mathcal{R}_t^{\pi^\star}(\lambda)} \nabla_{\mathbf{x}}(\hat{\pi} - \pi^\star)(\mathbf{x}_t^{\pi^\star})\|_{\text{op}}^2 d_u^{-1} \lambda,$$

where we applied Corollary E.2 on the second-to-last line, and for the last line we used by definition $\lambda_{\min}^+(\mathcal{P}_{\mathcal{R}_t^{\pi^\star}(\lambda)} \mathbf{W}_{1:t}^{\mathbf{z}}(\mathbf{x}_1^{\pi^\star}) \mathcal{P}_{\mathcal{R}_t^{\pi^\star}(\lambda)}) \gtrsim d_u^{-1} \lambda_{\min}^+(\mathcal{P}_{\mathcal{R}_t^{\pi^\star}(\lambda)} \mathbf{W}_{1:t}^{\mathbf{u}}(\mathbf{x}_1^{\pi^\star}) \mathcal{P}_{\mathcal{R}_t^{\pi^\star}(\lambda)}) \gtrsim d_u^{-1} \lambda$. Thus, re-arranging the inequalities, we have

$$\|\mathcal{P}_{\mathcal{R}_t^{\pi^\star}(\lambda)} \nabla_{\mathbf{x}}(\hat{\pi} - \pi^\star)(\mathbf{x}_t^{\pi^\star})\|_{\text{op}}^2$$
$$\lesssim \frac{d_u}{\sigma_{\mathbf{u}}^2 \lambda} \left( \|(\hat{\pi} - \pi^\star)(\mathbf{x}_t^{\pi^\star})\|^2 + \mathbb{E}_{\tilde{\mathbf{x}}_t | \mathbf{x}_1^{\pi^\star}} \|\hat{\pi}(\tilde{\mathbf{x}}_t) - \pi^\star(\tilde{\mathbf{x}}_t)\|^2 \right) + \frac{d_u \sigma_{\mathbf{u}}^2}{\lambda} C_\pi^2 C_{\text{stab}}^4,$$

which completes the result.

$\square$

In light of Proposition E.5, we have demonstrated that small estimation error along both un-noised *and* noise-injected states implies a first-order closeness of $\hat{\pi}$ and $\pi^\star$ along a subspace of our choosing. However, by choosing the excitation threshold $\lambda$ that we guarantee closeness above, we do not track: 1. error in the reachable subspace below the $\lambda$ threshold, 2. error for non-linearity. As stated, Proposition E.3 requires uniform closeness on a $\varepsilon$-scaled unit ball, which Proposition E.5 does not grant. Our next step is to prove the full version of Proposition 4.3.

**Proposition E.6.** *Let Assumption 4.1 hold. For any initial state* $\mathbf{x}_1$, *let* $\mathbf{x}_1^{\hat{\pi}} = \mathbf{x}_1^{\pi^\star} = \mathbf{x}_1$, *and consider the closed-loop trajectories generated by* $\hat{\pi}$ *and* $\pi^\star$. *Define the constant* $C_{\text{rem}} \triangleq 2C_{\text{reg}}(3 +$

$2C_{\mathbf{K}}^2 + 2C_\pi^2$). *Fix any sequence* $\{\lambda_t\}_{t=1}^{T-1}$, *where each* $\lambda_t \in [\lambda_{\min}^+(\mathbf{W}_{1:t}^{\mathbf{u}}(\mathbf{x}_1^{\pi^\star})), \lambda_{\max}(\mathbf{W}_{1:t}^{\mathbf{u}}(\mathbf{x}_1^{\pi^\star}))]$. *Then for any given* $\varepsilon \in [0, 1]$, $T \in \mathbb{N}$, *as long as:*

$$\max_{1 \le t \le T-1} \sup_{\substack{\|\mathbf{w}\| \le 1, \mathbf{w} \in \mathcal{R}_t^{\pi^\star}(\lambda_t) \\ \|\mathbf{r}\| \le 1, \mathbf{r} \in \mathcal{R}_t^{\pi^\star}(\lambda_t)^\perp \\ \|\mathbf{v}\| \le 1}} \left\| (\hat{\pi} - \pi^\star)(\mathbf{x}_t^{\pi^\star} + \varepsilon \mathbf{w} + \frac{c_{\text{stab}}}{\log(1/\rho)} \sqrt{\lambda_t} \varepsilon \mathbf{r} + C_{\text{rem}} \varepsilon^2 \mathbf{v}) \right\| \le c_{\text{stab}} \varepsilon,$$

*we are guaranteed* $\max_{1 \le t \le T} \|\mathbf{x}_t^{\hat{\pi}} - \mathbf{x}_t^{\pi^\star}\| \le \varepsilon$.

*Proof of Proposition E.6.* We prove this result by induction. Fix any $\varepsilon \in [0, 1]$. Define the quantity $C_\perp \triangleq \frac{1-\rho}{C_{\text{ISS}} \log(1/\rho)}$. Further define the shorthands $\mathcal{P}_t = \mathcal{P}_{\mathcal{R}_t^{\pi^\star}}(\lambda_t)$, and the *relative* orthogonal component $\mathcal{P}_t^\perp = \mathcal{P}_{\mathcal{R}_t^{\pi^\star}}(\lambda_t)^\perp$. Let us for each timestep $t$ define the set

$$\mathcal{V}_t \triangleq \left\{ \varepsilon \mathbf{w} + \sqrt{\lambda_t} C_\perp \varepsilon \mathbf{r} + C_{\text{rem}} \varepsilon^2 \mathbf{v} : \|\mathbf{w}\|, \|\mathbf{r}\|, \|\mathbf{v}\| \le 1, \ \mathbf{w} \in \mathcal{R}_t^{\pi^\star}(\lambda_t), \ \mathbf{r} \in \mathcal{R}_t^{\pi^\star}(\lambda_t)^\perp \right\}.$$

In addition to the statement of Proposition E.6, we claim that $\mathbf{x}_t^{\hat{\pi}} - \mathbf{x}_t^{\pi^\star} \in \mathcal{V}_t$ for each $t = 1, \dots, T$. Considering the base-case $T = 2$: since $\mathbf{x}_1^{\pi^\star} = \mathbf{x}_1^{\hat{\pi}}$ by construction, and thus $\hat{\pi}(\mathbf{x}_1^{\hat{\pi}}) = \hat{\pi}(\mathbf{x}_1^{\pi^\star})$, by assumption this satisfies $\|\hat{\pi}(\mathbf{x}_1^{\pi^\star}) - \pi^\star(\mathbf{x}_1^{\pi^\star})\| \le \frac{1-\rho}{C_{\text{ISS}}} \varepsilon$. By applying $(C_{\text{ISS}}, \rho)$-EISS, we have

$$\|\mathbf{x}_2^{\hat{\pi}} - \mathbf{x}_2^{\pi^\star}\| \le C_{\text{ISS}} \|\hat{\pi}(\mathbf{x}_1^{\pi^\star}) - \pi^\star(\mathbf{x}_1^{\pi^\star})\| \le C_{\text{ISS}} \frac{1-\rho}{C_{\text{ISS}}} \varepsilon \le \varepsilon.$$

Furthermore, recalling the definitions in Lemma E.1, we apply the $C_{\text{reg}}$-smoothness of the dynamics $f$ and take a second-order Taylor expansion around $(\mathbf{x}, \mathbf{u}) = (\mathbf{x}_1^{\pi^\star}, \pi^\star(\mathbf{x}_1^{\pi^\star}))$ to yield:

$$\mathbf{x}_2^{\hat{\pi}} - \mathbf{x}_2^{\pi^\star} = \mathbf{B}_1(\hat{\pi} - \pi^\star)(\mathbf{x}_1^{\hat{\pi}}) + \mathbf{r}_1^{\mathbf{x}}.$$

We observe this implies $\mathbf{B}_1(\hat{\pi} - \pi^\star)(\mathbf{x}_1^{\hat{\pi}}) \in \mathcal{R}_1^{\pi^\star}$, and Lemma E.1 implies $\|\mathbf{B}_1\| \le C_{\text{ISS}}$. On the other hand, since $\mathbf{W}_{1:2}^{\mathbf{u}} = \mathbf{B}_1 \mathbf{B}_1^\top$, we know $\|\mathcal{P}_1^\perp \mathbf{B}_1\| \le \sqrt{\lambda_1}$. Since $\mathbf{x}_1^{\hat{\pi}} = \mathbf{x}_1^{\pi^\star}$, we have:

$$\|\mathcal{P}_1 \mathbf{B}_1(\hat{\pi} - \pi^\star)(\mathbf{x}_1^{\hat{\pi}})\| \le C_{\text{ISS}} \|(\hat{\pi} - \pi^\star)(\mathbf{x}_1^{\hat{\pi}})\| \le C_{\text{ISS}} \frac{1-\rho}{C_{\text{ISS}}} \varepsilon \le \varepsilon$$

$$\|\mathcal{P}_1^\perp \mathbf{B}_1(\hat{\pi} - \pi^\star)(\mathbf{x}_1^{\hat{\pi}})\| \le \sqrt{\lambda_1} \|(\hat{\pi} - \pi^\star)(\mathbf{x}_1^{\hat{\pi}})\| \le \sqrt{\lambda_1} \frac{1-\rho}{C_{\text{ISS}}} \varepsilon \le \sqrt{\lambda_1} C_\perp \varepsilon$$

$$\|\mathbf{r}_1^{\mathbf{x}}\| \le C_{\text{reg}} \left( \|\mathbf{x}_1^{\hat{\pi}} - \mathbf{x}_1^{\pi^\star}\|^2 + \|\hat{\pi}(\mathbf{x}_1^{\hat{\pi}}) - \pi^\star(\mathbf{x}_1^{\pi^\star})\|^2 \right) \le C_{\text{reg}} \left( \frac{1-\rho}{C_{\text{ISS}}} \varepsilon \right)^2 \le C_{\text{rem}} \varepsilon^2,$$

which implies $\mathbf{x}_2^{\hat{\pi}} - \mathbf{x}_2^{\pi^\star} \in \mathcal{V}_2$. This completes the base-case.

Now for $T > 2$, we assume the statement holds for $T-1$; in particular, we have $\max_{1 \le t \le T-1} \|\mathbf{x}_t^{\hat{\pi}} - \mathbf{x}_t^{\pi^\star}\| \le \varepsilon$ and $\mathbf{x}_t^{\hat{\pi}} - \mathbf{x}_t^{\pi^\star} \in \mathcal{V}_t$ for $t \in [T-1]$. Then, by $(C_{\text{ISS}}, \rho)$-EISS we have:

$$\|\mathbf{x}_T^{\hat{\pi}} - \mathbf{x}_T^{\pi^\star}\| \le C_{\text{ISS}} \sum_{t=1}^{T-1} \rho^{T-1-t} \|\hat{\pi}(\mathbf{x}_t^{\hat{\pi}}) - \pi^\star(\mathbf{x}_t^{\hat{\pi}})\|$$

$$\le C_{\text{ISS}} \sum_{t=1}^{T-1} \rho^{T-1-t} \|(\hat{\pi} - \pi^\star)(\mathbf{x}_t^{\pi^\star} + \Delta_{\mathbf{x}_t})\|$$

$$\le C_{\text{ISS}} \sum_{t=1}^{T-1} \rho^{T-1-t} \left( \frac{1-\rho}{C_{\text{ISS}}} \varepsilon \right), \qquad \text{(Inductive hypothesis)}$$

where $\Delta_{\mathbf{x}_t} \triangleq \mathbf{x}_t^{\hat{\pi}} - \mathbf{x}_t^{\pi^\star}$ and the last line uses the induction hypothesis that each $\Delta_{\mathbf{x}_t} \in \mathcal{V}_t$, $t \in [T-1]$. This completes the first part of the induction step. It remains to show $\mathbf{x}_T^{\hat{\pi}} - \mathbf{x}_T^{\pi^\star} \in \mathcal{V}_T$. From the definition of the linearizations Eq. (E.2), we may write:

$$\mathbf{x}_T^{\hat{\pi}} - \mathbf{x}_T^{\pi^\star} = \sum_{s=1}^{T-1} \mathbf{A}_{s+1:T}^{\text{cl}} \mathbf{B}_s (\hat{\pi} - \pi^\star)(\mathbf{x}_s^{\hat{\pi}}) + \mathbf{A}_{s+1:T}^{\text{cl}} \mathbf{r}_s^{\mathbf{x}} \qquad (\text{E.8})$$

$$= \mathcal{P}_T \sum_{s=1}^{T-1} \mathbf{A}_{s+1:T}^{\mathrm{cl}} \mathbf{B}_s (\hat{\pi} - \pi^\star)(\mathbf{x}_s^{\hat{\pi}}) + \mathcal{P}_T^\perp \sum_{s=1}^{T-1} \mathbf{A}_{s+1:T}^{\mathrm{cl}} \mathbf{B}_s (\hat{\pi} - \pi^\star)(\mathbf{x}_s^{\hat{\pi}}) + \mathbf{A}_{s+1:T}^{\mathrm{cl}} \mathbf{r}_s^{\mathbf{x}}$$

where $\mathbf{r}_s^{\mathbf{x}}$ are the second-order remainder terms from linearizing the dynamics around $(\mathbf{x}_s^{\pi^\star}, \pi^\star(\mathbf{x}_s^{\pi^\star}))$ for $s \in [T-1]$. We first observe by definition $\sum_{s=1}^{T-1} \mathbf{A}_{s+1:T}^{\mathrm{cl}} \mathbf{B}_s (\hat{\pi} - \pi^\star)(\mathbf{x}_s^{\hat{\pi}}) \in \mathcal{R}_T^{\pi^\star}$, i.e. the first term on the first line lies in the reachable subspace. Focusing on the first term of the second line, we may trivially bound:

$$\left\| \mathcal{P}_T \sum_{s=1}^{T-1} \mathbf{A}_{s+1:T}^{\mathrm{cl}} \mathbf{B}_s (\hat{\pi} - \pi^\star)(\mathbf{x}_s^{\hat{\pi}}) \right\| \leq \sum_{s=1}^{T-1} \|\mathbf{A}_{s+1:T}^{\mathrm{cl}} \mathbf{B}_s\| \|(\hat{\pi} - \pi^\star)(\mathbf{x}_s^{\pi^\star} + \Delta_{\mathbf{x}_s})\|$$

$$\leq \sum_{s=1}^{T-1} C_{\mathrm{ISS}} \rho^{T-1-s} \left( \frac{1-\rho}{C_{\mathrm{ISS}}} \varepsilon \right) \leq \varepsilon,$$

where we used Lemma E.1 and the induction hypothesis for the last line. For the second term, we first observe that since $\mathbf{W}_{1:t}^{\mathbf{u}}(\mathbf{x}_1^{\pi^\star}) \triangleq \sum_{s=1}^{t-1} \mathbf{A}_{s+1:t}^{\mathrm{cl}} \mathbf{B}_s \mathbf{B}_s^\top \mathbf{A}_{s+1:t}^{\mathrm{cl}}{}^\top$, we have $\|\mathcal{P}_T^\perp \mathbf{A}_{s+1:T}^{\mathrm{cl}} \mathbf{B}_s\| \leq \|\mathcal{P}_T^\perp \mathbf{W}_{1:T}^{\mathbf{u}}(\mathbf{x}_1^{\pi^\star}) \mathcal{P}_T^\perp\|^{1/2} \leq \sqrt{\lambda_T}$. Alternatively, we always have by Lemma E.1 $\|\mathcal{P}_T^\perp \mathbf{A}_{s+1:T}^{\mathrm{cl}} \mathbf{B}_s\| \leq C_{\mathrm{ISS}} \rho^{T-1-s}$. Therefore, picking any $k \in [T-1]$, we have:

$$\left\| \mathcal{P}_T^\perp \sum_{s=1}^{T-1} \mathbf{A}_{s+1:T}^{\mathrm{cl}} \mathbf{B}_s (\hat{\pi} - \pi^\star)(\mathbf{x}_s^{\hat{\pi}}) \right\| \leq \sum_{s=1}^{T-1} \|\mathcal{P}_T^\perp \mathbf{A}_{s+1:T}^{\mathrm{cl}} \mathbf{B}_s\| \|(\hat{\pi} - \pi^\star)(\mathbf{x}_s^{\pi^\star} + \Delta_{\mathbf{x}_s})\|$$

$$\leq \left( k\sqrt{\lambda_T} + \sum_{s=1}^{T-1-k} C_{\mathrm{ISS}} \rho^{T-1-s} \right) \left( \frac{1-\rho}{C_{\mathrm{ISS}}} \varepsilon \right)$$

$$\leq \left( k\sqrt{\lambda_T} + \rho^{-k} \frac{C_{\mathrm{ISS}}}{1-\rho} \right) \left( \frac{1-\rho}{C_{\mathrm{ISS}}} \varepsilon \right).$$

Now, by solving for the optimal truncation point: $\min_{k \geq 1} k\sqrt{\lambda_T} + \rho^{-k} \frac{C_{\mathrm{ISS}}}{1-\rho}$, we may upper bound the resulting value by:

$$\min_{k \geq 1} k\sqrt{\lambda_T} + \rho^{-k} \frac{C_{\mathrm{ISS}}}{1-\rho} \leq \frac{\sqrt{\lambda_T}}{\log(1/\rho)} \left( 1 + \log \left( \frac{\sqrt{\lambda_T}(1-\rho)}{C_{\mathrm{ISS}} \log(1/\rho)} \right) \right)$$

$$\leq \frac{\sqrt{\lambda_T}}{\log(1/\rho)},$$

where for the last line we observe that $\sqrt{\lambda_T} \leq \sqrt{\lambda_{\max}(\mathbf{W}_{1:T}^{\mathbf{u}})} \leq \frac{C_{\mathrm{ISS}}}{1-\rho}$ by Lemma E.1, and thus $\log\left( \frac{\sqrt{\lambda_T}(1-\rho)}{C_{\mathrm{ISS}} \log(1/\rho)} \right) \leq 0$. Therefore, we may plug this back in to yield:

$$\left\| \mathcal{P}_T^\perp \sum_{s=1}^{T-1} \mathbf{A}_{s+1:T}^{\mathrm{cl}} \mathbf{B}_s (\hat{\pi} - \pi^\star)(\mathbf{x}_s^{\hat{\pi}}) \right\| \leq \frac{\sqrt{\lambda_T}}{\log(1/\rho)} \left( \frac{1-\rho}{C_{\mathrm{ISS}}} \varepsilon \right) = \sqrt{\lambda_T} C_\perp \varepsilon.$$

As for the last remainder term, we have:

$$\|\mathbf{A}_{s+1:T}^{\mathrm{cl}}\| \leq C_{\mathrm{ISS}} \rho^{T-1-s} \qquad \text{(Lemma E.1)}$$

$$\|\mathbf{r}_s^{\mathbf{x}}\| \leq C_{\mathrm{reg}} \left( \|\mathbf{x}_s^{\pi^\star} - \mathbf{x}_s^{\hat{\pi}}\|^2 + \|\hat{\pi}(\mathbf{x}_s^{\hat{\pi}}) - \pi^\star(\mathbf{x}_s^{\pi^\star})\|^2 \right) \qquad \text{(Assumption 4.1)}$$

$$\leq C_{\mathrm{reg}} \left( \varepsilon^2 + 2\|(\hat{\pi} - \pi^\star)(\mathbf{x}_s^{\pi^\star} + \Delta_{\mathbf{x}_s})\|^2 + 2\|\pi^\star(\mathbf{x}_s^{\pi^\star} + \Delta_{\mathbf{x}_s}) - \pi^\star(\mathbf{x}_s^{\pi^\star})\|^2 \right)$$

$$\leq C_{\mathrm{reg}} \left( 1 + 2c_{\mathrm{stab}}^2 + 2\|\nabla_{\mathbf{x}} \pi^\star(\mathbf{x}_s^{\pi^\star})^\top \Delta_{\mathbf{x}_s} + \mathbf{r}_s^\Delta\|^2 \right) \varepsilon^2$$

$$\leq C_{\mathrm{reg}} \left( 1 + 2c_{\mathrm{stab}}^2 + 2C_{\mathbf{K}} + 2C_\pi^2 \varepsilon^2 \right) \varepsilon^2$$

$$\leq 2C_{\mathrm{reg}} (3 + 2C_{\mathbf{K}}^2 + 2C_\pi^2) \varepsilon^2$$

$$\text{(E.9)}$$

$$\left\| \sum_{s=1}^{T-1} \mathbf{A}_{s+1:T}^{\mathrm{cl}} \mathbf{r}_s^{\mathbf{x}} \right\| \leq \sum_{s=1}^{T-1} C_{\mathrm{ISS}} \rho^{T-1-s} \|\mathbf{r}_s^{\mathbf{x}}\| \leq 2C_{\mathrm{reg}}(3 + 2C_{\mathbf{K}}^2 + 2C_\pi^2) \frac{C_{\mathrm{ISS}}}{1-\rho} \varepsilon^2 \triangleq C_{\mathrm{rem}} \varepsilon^2.$$

Therefore, putting all the pieces back into Eq. (E.8), we have:

$$\mathbf{x}_T^{\hat{\pi}} - \mathbf{x}_T^{\pi^\star} = \mathcal{P}_T \sum_{s=1}^{T-1} \mathbf{A}_{s+1:T}^{\mathrm{cl}} \mathbf{B}_s(\hat{\pi} - \pi^\star)(\mathbf{x}_s^{\hat{\pi}}) + \mathcal{P}_T^\perp \sum_{s=1}^{T-1} \mathbf{A}_{s+1:T}^{\mathrm{cl}} \mathbf{B}_s(\hat{\pi} - \pi^\star)(\mathbf{x}_s^{\hat{\pi}}) + \mathbf{A}_{s+1:T}^{\mathrm{cl}} \mathbf{r}_s^{\mathbf{x}}$$

$$\leq \varepsilon \mathbf{w} + \sqrt{\lambda_T} C_\perp \mathbf{r} + C_{\mathrm{rem}} \varepsilon^2 \mathbf{v}, \quad \|\mathbf{w}\|, \|\mathbf{r}\|, \|\mathbf{v}\| \leq 1, \quad \mathbf{w} \in \mathcal{R}_T^{\pi^\star}(\lambda_T), \quad \mathbf{r} \in \mathcal{R}_T^{\pi^\star}(\lambda_T)^\perp$$

$$\in \mathcal{V}_T.$$

we have demonstrated $\mathbf{x}_T^{\hat{\pi}} - \mathbf{x}_T^{\pi^\star} \in \mathcal{V}_T$, completing the induction step and thus the proof.

$\square$

To review, we have established two key tools in Proposition E.5 and Proposition E.6, corresponding to Proposition 4.4 and Proposition 4.3 in the body, respectively. The first states that, fixing our attention to the component of the reachable subspace that is excitable above a threshold $\lambda$ (to be determined in hindsight), we may bound the first-order, i.e. Jacobian error between $\hat{\pi}$ and $\pi^\star$ in terms of their error on the *mixture* distribution $\mathbb{P}_{\pi^\star, \sigma_{\mathbf{u}}, \alpha}$. The second states that, fixing an excitation level, as long as we ensure $\hat{\pi}$ matches $\pi^\star$ sufficiently well on the set $\mathcal{V}_t$ for each $t$, which decomposes into the "excitable", (linearly) reachable component in $\mathcal{R}^{\pi^\star}(\lambda_t)$, the low-excitation (linearly) reachable component in $\mathcal{R}^{\pi^\star}(\lambda_t)^\perp$, and a generic second-order term, the resulting closed-loop trajectories will remain close.

We are now ready to prove our main guarantee for noise injection.

### E.5 GUARANTEES WITHOUT CONTROLLABILITY: PROOF OF THEOREM 2

We dedicate most of the effort into establishing the following result.

**Proposition E.7.** *Let Assumption 4.1 hold. Let $C_{\mathbf{r}}$ be defined as in Corollary E.1 and $C_{\mathrm{rem}}$ as in Proposition E.6. Let the noise-scale $\sigma_{\mathbf{u}} > 0$ satisfy*

$$\sigma_{\mathbf{u}} \lesssim \min \left\{ \sqrt{\frac{\log(1/\rho)}{L_\pi d_u}} \frac{c_{\mathrm{stab}}^3}{C_\pi}, \; \frac{\log(1/\rho)^2}{L_\pi^2 d_u} \frac{c_{\mathrm{stab}}^4}{C_{\mathbf{r}}}, \; c_{\mathrm{stab}} \frac{\sqrt{1 + 4C_{\mathbf{K}}^2}}{C_\pi} \right\}. \tag{E.10}$$

*Consider a candidate policy $\hat{\pi}$. Defining $C_{\mathrm{traj}} \triangleq 2C_{\mathrm{rem}} L_\pi + 6C_\pi \left(1 + \frac{1}{4} \frac{c_{\mathrm{stab}}}{L_\pi} + C_{\mathrm{rem}}^2\right)$, we have the following bound on the expected (clipped) trajectory error:*

$$\mathbb{E}_{\mathbf{x}_1 \sim \mathbb{P}_{\mathbf{x}_1^{\pi^\star}}} \left[ \max_{1 \leq t \leq T} \|\mathbf{x}_t^{\hat{\pi}} - \mathbf{x}_t^{\pi^\star}\|^2 \wedge 1 \right]$$

$$\lesssim C_{\mathrm{stab}}^2 \left(1 + C_{\mathrm{traj}}^2 C_{\mathrm{stab}}^2 + \frac{d_u L_\pi^2}{\log(1/\rho)^2 \sigma_{\mathbf{u}}^2}\right) \mathbb{E}_{\mathbf{x}_1^{\pi^\star}} \left[ \max_{t \leq T-1} \|\hat{\pi}(\mathbf{x}_t^{\pi^\star}) - \pi^\star(\mathbf{x}_t^{\pi^\star})\|^2 + \mathbb{E}_{\tilde{\mathbf{x}}_t | \mathbf{x}_1^{\pi^\star}} \|\hat{\pi}(\tilde{\mathbf{x}}_t) - \pi^\star(\tilde{\mathbf{x}}_t)\|^2 \right]$$

$$\leq C_{\mathrm{stab}}^2 \left(1 + C_{\mathrm{traj}}^2 C_{\mathrm{stab}}^2 + \frac{d_u L_\pi^2}{\log(1/\rho)^2 \sigma_{\mathbf{u}}^2}\right) \mathbf{J}_{\mathrm{DEMO}, T}(\hat{\pi}; \mathbb{P}_{\pi^\star, \sigma_{\mathbf{u}}, \alpha}).$$

*Proof of Proposition E.7.* Let us define the shorthands for the per-timestep trajectory and estimation errors:

$$r_t^{\mathrm{traj}}(\hat{\pi}, \pi^\star) \triangleq \|\mathbf{x}_t^{\hat{\pi}} - \mathbf{x}_t^{\pi^\star}\|^2 \wedge 1$$

$$r_t^{\mathrm{est}}(\hat{\pi}; \pi^\star) \triangleq \|\hat{\pi}(\mathbf{x}_t^{\pi^\star}) - \pi^\star(\mathbf{x}_t^{\pi^\star})\|^2$$

$$r_t^{\mathrm{est}}(\hat{\pi}; \tilde{\pi}^\star) \triangleq \mathbb{E}_{\tilde{\mathbf{x}}_t | \mathbf{x}_1^{\pi^\star}} \left[ \|\hat{\pi}(\tilde{\mathbf{x}}_t) - \pi^\star(\tilde{\mathbf{x}}_t)\|^2 \right],$$

As in Proposition E.6, let us define a sequence $\{\lambda_t\}_{t=1}^{T-1}$, where each $\lambda_t \in [\lambda_{\min}^+(\mathbf{W}_{1:t}^{\mathbf{u}}(\mathbf{x}_1^{\pi^\star})), \lambda_{\max}(\mathbf{W}_{1:t}^{\mathbf{u}}(\mathbf{x}_1^{\pi^\star}))]$, as well as the truncated subspaces and projection matrices: $\mathcal{R}_t^{\pi^\star}(\lambda_t)$, $\mathcal{P}_t = \mathcal{P}_{\mathcal{R}_t^{\pi^\star}}(\lambda_t)$. By Proposition E.5, noise injection certifies a norm bound on $\nabla_{\mathbf{x}}(\hat{\pi} - \pi^\star)(\mathbf{x}_1^{\pi^\star})$ restricted to $\mathcal{R}_t^{\pi^\star}(\lambda_t)$, for each $t \geq 2$. Accordingly, we define the event:

$$\mathcal{E}_{\nabla \pi}(c) \triangleq \left\{ \max_{t \leq T-1} \|\mathcal{P}_t \nabla_{\mathbf{x}}(\hat{\pi} - \pi^\star)(\mathbf{x}_1^{\pi^\star})\|_{\mathrm{op}} \lesssim c \right\}.$$

We may decompose the desired quantity into:

$$\mathbb{E}_{\mathbf{x}_1^{\pi^\star}}\left[\max_{t\leq T-1} r_t^{\mathrm{traj}}(\hat{\pi},\pi^\star)\right] = \underbrace{\mathbb{E}_{\mathbf{x}_1^{\pi^\star}}\left[\max_{t\leq T-1} r_t^{\mathrm{traj}}(\hat{\pi},\pi^\star)\mathbf{1}\{\mathcal{E}_{\nabla\pi}(c_{\mathrm{stab}})\}\right]}_{T_1} + \underbrace{\mathbb{E}_{\mathbf{x}_1^{\pi^\star}}\left[\max_{t\leq T-1} r_t^{\mathrm{traj}}(\hat{\pi},\pi^\star)\mathbf{1}\{\mathcal{E}_{\nabla\pi}(c_{\mathrm{stab}})^c\}\right]}_{T_2}.$$

In addition to the requirements on $\sigma_{\mathbf{u}}$ in Proposition E.5 for $\lambda = \lambda_t$, assume that $\sigma_{\mathbf{u}}$ satisfies across $t \geq 2$: $\sigma_{\mathbf{u}} \lesssim \sqrt{\frac{\lambda_t}{d_u}\frac{c_{\mathrm{stab}}^3}{C_\pi}}$, such that $\frac{d_u\sigma_{\mathbf{u}}^2}{\lambda}C_\pi^2 C_{\mathrm{stab}}^4 \lesssim c_{\mathrm{stab}}^2$. Since $r_t^{\mathrm{traj}}(\hat{\pi},\pi^\star) \leq 1$, we may then bound $T_2$ by:

$$\begin{aligned}
T_2 = \mathbb{E}_{\mathbf{x}_1^{\pi^\star}}\left[\max_{t\leq T-1} r_t^{\mathrm{traj}}(\hat{\pi},\pi^\star)\mathbf{1}\{\mathcal{E}_{\nabla\pi}(c_{\mathrm{stab}})^c\}\right] &\leq \mathbb{P}\left[\max_{t\leq T-1}\|\mathcal{P}_t\nabla_{\mathbf{x}}(\hat{\pi}-\pi^\star)(\mathbf{x}_1^{\pi^\star})\|_{\mathrm{op}} \gtrsim c_{\mathrm{stab}}\right] \\
&\leq \mathbb{P}\left[\max_{t\leq T-1}\frac{d_u}{\lambda_t\sigma_{\mathbf{u}}^2}\left(r_t^{\mathrm{est}}(\hat{\pi};\pi^\star) + r_t^{\mathrm{est}}(\hat{\pi};\tilde{\pi}^\star)\right) \gtrsim c_{\mathrm{stab}}^2\right] \\
&\leq C_{\mathrm{stab}}^2 \frac{d_u\max_t \lambda_t^{-1}}{\sigma_{\mathbf{u}}^2}\mathbb{E}_{\mathbf{x}_1^{\pi^\star}}\left[\max_{t\leq T-1} r_t^{\mathrm{est}}(\hat{\pi};\pi^\star) + r_t^{\mathrm{est}}(\hat{\pi};\tilde{\pi}^\star)\right],
\end{aligned}$$

where the second line arises from applying Proposition E.5 and the noise-scale condition $\sigma_{\mathbf{u}} \lesssim \sqrt{\frac{\lambda_t}{d_u}\frac{c_{\mathrm{stab}}^3}{C_\pi}}$, and the last line comes from Markov's inequality. As for $T_1$, we set the decomposition for a given $\tau \in (0,1)$ to be determined later:

$$T_1 \leq \underbrace{\int_0^\tau \mathbb{P}\left[\max_{t\leq T-1} r_t^{\mathrm{traj}}(\hat{\pi},\pi^\star)\mathbf{1}\{\mathcal{E}_{\nabla\pi}(c_{\mathrm{stab}})\} > \varepsilon\right] d\varepsilon}_{T_1^a} + \underbrace{\mathbb{P}\left[\max_{t\leq T-1} r_t^{\mathrm{traj}}(\hat{\pi},\pi^\star)\mathbf{1}\{\mathcal{E}_{\nabla\pi}(c_{\mathrm{stab}})\} > \tau\right]}_{T_1^b}.$$

First, writing out the requirement of Proposition E.6, casting $\varepsilon \mapsto \sqrt{\varepsilon}$, we have:

$$\begin{aligned}
&\sup_{\substack{\|\mathbf{w}\|\leq 1, \mathbf{w}\in\mathcal{R}_t^{\pi^\star}(\lambda_t) \\ \|\mathbf{r}\|\leq 1, \mathbf{r}\in\mathcal{R}_t^{\pi^\star}(\lambda_t)^\perp \\ \|\mathbf{v}\|\leq 1}} \|(\hat{\pi}-\pi^\star)(\mathbf{x}_t^{\pi^\star} + \sqrt{\varepsilon}\mathbf{w} + \frac{c_{\mathrm{stab}}}{\log(1/\rho)}\sqrt{\lambda_t}\sqrt{\varepsilon}\mathbf{r} + C_{\mathrm{rem}}\varepsilon\mathbf{v})\| \\
&\leq \|(\hat{\pi}-\pi^\star)(\mathbf{x}_t^{\pi^\star})\| + \sup_{\substack{\|\mathbf{w}\|\leq 1, \mathbf{w}\in\mathcal{R}_t^{\pi^\star}(\lambda_t) \\ \|\mathbf{r}\|\leq 1, \mathbf{r}\in\mathcal{R}_t^{\pi^\star}(\lambda_t)^\perp \\ \|\mathbf{v}\|\leq 1}} \|\nabla_{\mathbf{x}}(\hat{\pi}-\pi^\star)(\mathbf{x}_t^{\pi^\star})^\top(\sqrt{\varepsilon}\mathbf{w} + \frac{c_{\mathrm{stab}}}{\log(1/\rho)}\sqrt{\lambda_t}\sqrt{\varepsilon}\mathbf{r} + C_{\mathrm{rem}}\varepsilon\mathbf{v})\| \\
&\quad + 2C_\pi\|\sqrt{\varepsilon}\mathbf{w} + \frac{c_{\mathrm{stab}}}{\log(1/\rho)}\sqrt{\lambda_t}\sqrt{\varepsilon}\mathbf{r} + C_{\mathrm{rem}}\varepsilon\mathbf{v}\|^2 \\
&\leq \|(\hat{\pi}-\pi^\star)(\mathbf{x}_t^{\pi^\star})\| + \sup_{\|\mathbf{w}\|\leq 1, \mathbf{w}\in\mathcal{R}_t^{\pi^\star}(\lambda_t)} \|\nabla_{\mathbf{x}}(\hat{\pi}-\pi^\star)(\mathbf{x}_t^{\pi^\star})^\top\mathbf{w}\|\sqrt{\varepsilon} + 2L_\pi\frac{c_{\mathrm{stab}}}{\log(1/\rho)}\sqrt{\lambda_t}\sqrt{\varepsilon} + 2C_{\mathrm{rem}}L_\pi\varepsilon \\
&\quad + 6C_\pi\left(1 + \frac{c_{\mathrm{stab}}}{\log(1/\rho)}\sqrt{\lambda_t} + C_{\mathrm{rem}}^2\right)\varepsilon \\
&\leq \|(\hat{\pi}-\pi^\star)(\mathbf{x}_t^{\pi^\star})\| + \|\mathcal{P}_t\nabla_{\mathbf{x}}(\hat{\pi}-\pi^\star)(\mathbf{x}_t^{\pi^\star})\|_{\mathrm{op}}\sqrt{\varepsilon} \\
&\quad + 2L_\pi\frac{c_{\mathrm{stab}}}{\log(1/\rho)}\sqrt{\lambda_t}\sqrt{\varepsilon} + \left(2C_{\mathrm{rem}}L_\pi + 6C_\pi\left(1 + \frac{c_{\mathrm{stab}}}{\log(1/\rho)}\sqrt{\lambda_t} + C_{\mathrm{rem}}^2\right)\right)\varepsilon.
\end{aligned}$$

Let us interpret what this yields. On the last line, the first term is the on-expert error term $r_t^{\mathrm{est}}(\hat{\pi};\pi^\star)$, the second term is controlled by Proposition E.5, and the rest of the terms are the errors for which we do not guarantee control. To leverage Proposition E.6, it suffices to have the last line bounded by $c_{\mathrm{stab}}\sqrt{\varepsilon}$. Intuitively, the higher order error term scaling as $\varepsilon$ automatically satisfies this for sufficiently small $\varepsilon$, which leaves the error term scaling as $\sqrt{\lambda_t\varepsilon}$. This is where we set the excitation levels $\{\lambda_t\}$ in hindsight. Observing the above, it suffices to set:

$$\lambda_t = \frac{1}{16}L_\pi^{-2}\log(1/\rho)^2, \quad t \geq 2.$$

In other words, for components of the controllability Gramian below this $\lambda_t$, the excitability is low enough such that we do not need to guarantee $\hat{\pi}, \pi^\star$ match on them. For convenience, let us now define the quantity:

$$C_{\mathsf{traj}} \triangleq 2C_{\mathsf{rem}}L_\pi + 6C_\pi \left(1 + \frac{1}{4}\frac{c_{\mathsf{stab}}}{L_\pi} + C_{\mathsf{rem}}^2\right).$$

Therefore, setting $\tau \approx C_{\mathsf{traj}}^{-2}c_{\mathsf{stab}}^2$, we may bound $T_1^a$ by applying Proposition E.6:

$$T_1^a = \int_0^\tau \mathbb{P}\left[\max_{t\le T-1} r_t^{\mathsf{traj}}(\hat{\pi}, \pi^\star)\mathbf{1}\{\mathcal{E}_{\nabla\pi}(c_{\mathsf{stab}})\} > \varepsilon\right]\,\mathrm{d}\varepsilon$$

$$\le \int_0^\tau \mathbb{P}\left[\max_{t\le T-1}\sup_{\substack{\|\mathbf{w}\|\le 1, \mathbf{w}\in\mathcal{R}_t^{\pi^\star}(\lambda_t)\\ \|\mathbf{r}\|\le 1, \mathbf{r}\in\mathcal{R}_t^{\pi^\star}(\lambda_t)^\perp\\ \|\mathbf{v}\|\le 1}} \|(\hat{\pi}-\pi^\star)(\mathbf{x}_t^{\pi^\star} + \sqrt{\varepsilon}\mathbf{w} + \frac{1}{4}\frac{c_{\mathsf{stab}}}{L_\pi}\sqrt{\varepsilon}\mathbf{r} + C_{\mathsf{rem}}\varepsilon\mathbf{v})\|^2\right.$$

$$\left.\cdot\,\mathbf{1}\{\mathcal{E}_{\nabla\pi}(c_{\mathsf{stab}})\} > c_{\mathsf{stab}}^2\varepsilon\right]\,\mathrm{d}\varepsilon$$

$$\le \int_0^\tau \mathbb{P}\left[\max_{t\le T-1}\left(r_t^{\mathsf{est}}(\hat{\pi};\pi^\star) + \|\mathcal{P}_t\nabla_{\mathbf{x}}(\hat{\pi}-\pi^\star)(\mathbf{x}_t^{\pi^\star})\|_{\mathsf{op}}^2\varepsilon + C_{\mathsf{traj}}^2\varepsilon^2\right)\mathbf{1}\{\mathcal{E}_{\nabla\pi}(c_{\mathsf{stab}})\} \gtrsim c_{\mathsf{stab}}^2\varepsilon\right]\,\mathrm{d}\varepsilon$$

$$= \int_0^\tau \mathbb{P}\left[\max_{t\le T-1} r_t^{\mathsf{est}}(\hat{\pi};\pi^\star) \gtrsim c_{\mathsf{stab}}^2\varepsilon\right]\,\mathrm{d}\varepsilon \qquad (\text{Def. of } \mathcal{E}_{\nabla\pi}(c_{\mathsf{stab}}); C_{\mathsf{traj}}^2\varepsilon \lesssim c_{\mathsf{stab}}^2 \text{ for } \varepsilon\le\tau)$$

$$\approx C_{\mathsf{stab}}^2\mathbb{E}_{\mathbf{x}_1^{\pi^\star}}\left[\max_{t\le T-1} r_t^{\mathsf{est}}(\hat{\pi};\pi^\star)\right].$$

The bound on $T_1^b$ follows similarly:

$$T_1^b \le \mathbb{P}\left[\max_{t\le T-1} r_t^{\mathsf{est}}(\hat{\pi};\pi^\star) \gtrsim c_{\mathsf{stab}}^2\tau\right]$$

$$\le C_{\mathsf{traj}}^2 C_{\mathsf{stab}}^4 \mathbb{E}_{\mathbf{x}_1^{\pi^\star}}\left[\max_{t\le T-1} r_t^{\mathsf{est}}(\hat{\pi};\pi^\star)\right]. \qquad\qquad (\text{Markov's})$$

Putting everything together, we get the final bound:

$$\mathbb{E}_{\mathbf{x}_1^{\pi^\star}}\left[\max_{t\le T-1} r_t^{\mathsf{traj}}(\hat{\pi},\pi^\star)\right] \le T_1^a + T_1^b + T_2$$

$$\le C_{\mathsf{stab}}^2\left((1 + C_{\mathsf{traj}}^2 C_{\mathsf{stab}}^2)\mathbb{E}_{\mathbf{x}_1^{\pi^\star}}\left[\max_{t\le T-1} r_t^{\mathsf{est}}(\hat{\pi};\pi^\star)\right] + \frac{d_u\max_t\lambda_t^{-1}}{\sigma_{\mathbf{u}}^2}\mathbb{E}_{\mathbf{x}_1^{\pi^\star}}\left[\max_{t\le T-1} r_t^{\mathsf{est}}(\hat{\pi};\pi^\star) + r_t^{\mathsf{est}}(\hat{\pi};\tilde{\pi}^\star)\right]\right)$$

$$\approx C_{\mathsf{stab}}^2\left((1 + C_{\mathsf{traj}}^2 C_{\mathsf{stab}}^2)\mathbb{E}_{\mathbf{x}_1^{\pi^\star}}\left[\max_{t\le T-1} r_t^{\mathsf{est}}(\hat{\pi};\pi^\star)\right] + \frac{d_u L_\pi^2}{\log(1/\rho)^2\sigma_{\mathbf{u}}^2}\mathbb{E}_{\mathbf{x}_1^{\pi^\star}}\left[\max_{t\le T-1} r_t^{\mathsf{est}}(\hat{\pi};\pi^\star) + r_t^{\mathsf{est}}(\hat{\pi};\tilde{\pi}^\star)\right]\right)$$

$$\le C_{\mathsf{stab}}^2\left(1 + C_{\mathsf{traj}}^2 C_{\mathsf{stab}}^2 + \frac{d_u L_\pi^2}{\log(1/\rho)^2\sigma_{\mathbf{u}}^2}\right)\mathbb{E}_{\mathbf{x}_1^{\pi^\star}}\left[\max_{t\le T-1} r_t^{\mathsf{est}}(\hat{\pi};\pi^\star) + r_t^{\mathsf{est}}(\hat{\pi};\tilde{\pi}^\star)\right],$$

which gives the desired result.

$$\square$$

Therefore, by using the trivial bound $\mathbf{J}_{\mathrm{TRAJ},2,T}^{\mathbf{x}}(\hat{\pi}) \le T\mathbb{E}_{\mathbf{x}_1\sim\mathbb{P}_{\mathbf{x}_1^{\pi^\star}}}\left[\max_{1\le t\le T}\|\mathbf{x}_t^{\hat{\pi}} - \mathbf{x}_t^{\pi^\star}\|^2 \wedge 1\right]$ and applying Lemma D.1 to translate to $\mathbf{J}_{\mathrm{TRAJ},2,T}(\hat{\pi})$, we get the final result.

**Theorem 4** (Trajectory error bound; full ver. of Theorem 2). *Let Assumption 4.1 hold. Let $C_{\mathbf{r}}$ be defined as in Corollary E.1 and $C_{\mathsf{rem}}$ as in Proposition E.6. Let the noise-scale $\sigma_{\mathbf{u}} > 0$ satisfy*

$$\sigma_{\mathbf{u}} \lesssim \min\left\{\sqrt{\frac{\log(1/\rho)}{L_\pi d_u}}\frac{c_{\mathsf{stab}}^3}{C_\pi}, \frac{\log(1/\rho)^2}{L_\pi^2 d_u}\frac{c_{\mathsf{stab}}^4}{C_{\mathbf{r}}}, c_{\mathsf{stab}}\frac{\sqrt{1 + 4C_{\mathbf{K}}^2}}{C_\pi}\right\}. \qquad (\text{E.11})$$

*Consider a candidate policy $\hat{\pi}$. Defining $C_{\text{traj}} \triangleq 2C_{\text{rem}}L_\pi + 6C_\pi\left(1 + \frac{1}{4}\frac{c_{\text{stab}}}{L_\pi} + C_{\text{rem}}^2\right)$, we may bound the* trajectory error *by the* on-expert error *on the mixture distribution $\mathbb{P}_{\pi^\star, \sigma_{\mathbf{u}}, 0.5}$ as:*

$$\mathbf{J}_{\text{TRAJ},2,T}(\hat{\pi}) \lesssim T(1 + L_\pi^2)C_{\text{stab}}^2\left(1 + C_{\text{traj}}^2 C_{\text{stab}}^2 + \frac{d_u L_\pi^2}{\log(1/\rho)^2 \sigma_{\mathbf{u}}^2}\right)\mathbf{J}_{\text{DEMO},T}(\hat{\pi}; \mathbb{P}_{\pi^\star, \sigma_{\mathbf{u}}, \alpha})$$

$$= O_\star(T)\,\sigma_{\mathbf{u}}^{-2}\mathbf{J}_{\text{DEMO},T}(\hat{\pi}; \mathbb{P}_{\pi^\star, \sigma_{\mathbf{u}}, 0.5}).$$

We conclude this section with a few technical remarks.

**Remark E.1** (Horizon $T$ dependence). We note the linear-in-horizon $T$ dependence arises from a naive conversion between $\max_{1 \leq t \leq T}$ and $\sum_{t=1}^T$. We note that Proposition E.7 can actually be interpreted as bounding $\mathbf{J}_{\text{TRAJ},\infty,T}(\hat{\pi}) \leq O_\star(1)\,\mathbf{J}_{\text{DEMO},\infty,T}(\hat{\pi}; \mathbb{P}_{\pi^\star, \sigma_{\mathbf{u}}, 0.5})$, for appropriately defined $\infty$/"max"-norm, which does not exhibit any horizon dependence. We expect a more fine-grained analysis, e.g. leveraging Lemma D.3, to similarly remove the $T$ dependence from $\mathbf{J}_{\text{TRAJ},p,T}$ and $\mathbf{J}_{\text{DEMO},p,T}$, with the main technical barrier in extending Proposition 4.3 (Proposition E.6).

**Remark E.2** (Noise-scale $\sigma_{\mathbf{u}}$ dependence). We note that the final bound in Theorem 4 has a $\sigma_{\mathbf{u}}^{-2}$ dependence. Firstly, we note that, by removing additive factors of $\sigma_{\mathbf{u}}$ (as in Suboptimal Proposition 4.2 or Proposition 4.1), we do not need to trade-off $\sigma_{\mathbf{u}}$ with the on-expert error $\mathbf{J}_{\text{DEMO},2,T}(\hat{\pi}; \mathbb{P}_{\pi^\star, \sigma_{\mathbf{u}}, 0.5})$, and can in fact set $\sigma_{\mathbf{u}}$ as large as permissible up to the smoothness constraints, turning the dependence $O_\star(1)$. However, observing where $\sigma_{\mathbf{u}}$ arises in the proof of Proposition E.7, it comes solely from applying Markov's inequality on the event $\mathbb{P}\left[\max_{t \leq T-1}\frac{d_u}{\lambda_t \sigma_{\mathbf{u}}^2}(r_t^{\text{est}}(\hat{\pi}; \pi^\star) + r_t^{\text{est}}(\hat{\pi}; \tilde{\pi}^\star)) \gtrsim c_{\text{stab}}^2\right]$. We can envision instead applying a Chebyshev inequality. For example, if we square both sides, we raise the estimation error to quartic in $\|(\hat{\pi} - \pi^\star)(\mathbf{x})\|$. If the estimation error satisfies moment-equivalence conditions, such as (4-2) hypercontractivity conditions that have appeared in prior learning-for-control literature (Kakade et al., 2020; Ziemann & Tu, 2022), this pushes the $\sigma_{\mathbf{u}}$ dependence to an additive higher-order term. This crystallizes the intuition that the noise-level $\sigma_{\mathbf{u}}$ actually enters the trajectory error as a higher-order term (or equivalently, in the burn-in), explaining why huge differences in $\sigma_{\mathbf{u}}$ scale have similar effects on the final performance (see Figure 1). We avoid introducing these technical conditions in the body for clarity. Similarly, we note the proof of Proposition E.7 also reveals that the on-expert error on the *mixture* distribution $\mathbb{P}_{\pi^\star, \sigma_{\mathbf{u}}, \alpha}$ enters only via the term depending on $\sigma_{\mathbf{u}}$, and thus similarly the number of noised trajectories need not scale proportionally to $n$. This explains why the final performance of an imitator policy is often not sensitive to the exact proportion of noised trajectories $\alpha$ in the training data, as long as *some* trajectories are noised and some are clean; see Figure 3.

### E.6 GUARANTEES FOR ANY $\mathbf{J}_{\text{TRAJ},p,T}$, $p \in [1, \infty)$

As stated above, by nature of Proposition E.7, setting $p \neq \infty$ our trajectory error guarantee $\mathbf{J}_{\text{TRAJ},p,T}$ in Theorem 4 naively accumulates a linear-in-horizon $T$ dependence. However, this horizon-dependence may seem qualitatively conservative; since the expert-induced system is EISS, one might hope that past "mistakes" are forgotten exponentially. Determining this rigorously requires some additional effort, as we cannot rely on our linchpin result in Proposition E.6, which translates to *per-timestep* control of on-expert error $\max_{t \geq 1}\|\hat{\pi}(\mathbf{x}_t) - \pi^\star(\mathbf{x}_t)\|$. We first establish the following key recursion.

**Lemma E.8** (Key Recursion). *Consider non-negative sequences $\{\Delta_t\}_{t \geq 1}$, $\{\Delta_t^\perp\}_{t \geq 1}$ that satisfy $\Delta_1 = \Delta_1^\perp = 0$ and for all $t \geq 2$:*

$$\Delta_t \leq \sum_{s=1}^{t-1} C_1 \rho^{t-1-s}\varepsilon_s + C_2 \rho^{t-1-s}\tau_s \Delta_s + C_3 \min\{\gamma, \rho^{t-1-s}\}\Delta_s + C_4 \rho^{t-1-s}\Delta_s^2 + C_\perp \rho^{t-1-s}\Delta_s^\perp$$

$$\Delta_t^\perp \leq \sum_{s=1}^{t-1} C_1 \rho^{t-1-s}\varepsilon_s + C_3 \min\{\gamma, \rho^{t-1-s}\}\Delta_s + C_4 \rho^{t-1-s}\Delta_s^2,$$

*for constants $C_1 \geq 1$, $C_2, C_3, C_4, C_\perp > 0$, $\rho \in [0, 1)$, $\gamma \in [0, 1)$, and non-negative sequences $\{\tau_s\}_{s \geq 1}$, $\{\varepsilon_s\}_{s \geq 1}$. Then, as long as the following conditions hold:*

$$\varepsilon_s \leq \varepsilon_{\max} \lesssim \frac{(1-\rho)^2(1 + C_\perp)}{C_4\left(1 + \frac{5C_\perp}{1-\rho}\right)}, \quad \tau_s \leq \tau_{\max} \lesssim \frac{C_\perp}{C_2\left(1 + \frac{5C_\perp}{1-\rho}\right)}, \quad \forall s \geq 1$$

$$\gamma \lesssim \left( \frac{(1-\rho)^2(1+C_\perp)}{C_3\left(1+\frac{5C_\perp}{1-\rho}\right)} \right)^4,$$

*we have that $\Delta_t$ satisfies $\Delta_t \lesssim \overline{C} \sum_{s=1}^{t-1} \bar{\rho}^{t-1-s} \varepsilon_s$, $t \geq 1$, where $\overline{C} = C_1\left(1 + \frac{C_\perp}{1-\rho}\right)$, $\bar{\rho} = \frac{1+\rho}{2}$.*

*Proof of Lemma E.8.* Toward establishing the result, we posit the existence of a sequence $\{\overline{\Delta}_t\}_{t\geq 1}$ that admits form $\overline{\Delta}_t \triangleq \overline{C} \sum_{s=1}^{t-1} \bar{\rho}^{t-1-s} \varepsilon_s$, $t \geq 2$, where $\overline{\Delta}_1 \triangleq 0$, $\overline{\Delta}_t \geq \Delta_t$ for all $t \geq 1$. We also posit a corresponding sequence $\{\overline{\Delta}_t^\perp\}_{t\geq 1}$, where $\overline{\Delta}_1^\perp \triangleq 0$, $\overline{\Delta}_t^\perp \triangleq \overline{C}_\perp \sum_{s=1}^{t-1} \bar{\rho}^{t-1-s} \varepsilon_s$, satisfying $\overline{\Delta}_t^\perp \geq \Delta_t^\perp$ for all $t \geq 1$. We will determine $\bar{\rho} \in (\rho, 1)$, $\overline{C}, \overline{C}_\perp \geq C_1$ in hindsight. As in the statement, we further impose the constraints $\varepsilon_s \leq \varepsilon_{\max}$, $\tau_s \leq \tau_{\max}$, $s \geq 1$, where $\varepsilon_{\max}, \tau_{\max}, \gamma$ will be set in hindsight. It remains to determine that $\overline{\Delta}_t \geq \Delta_t$ for all $t \geq 2$. For the base-case $t = 2$, since $\overline{\Delta}_1 = \Delta_1 = 0$, we have trivially $\Delta_2 \leq C_1 \varepsilon_1 \leq \overline{C} \varepsilon_1 = \overline{\Delta}_2$, and $\Delta_2^\perp \leq C_1 \varepsilon_1 = \overline{\Delta}_2^\perp$. Now, given $\Delta_s \leq \overline{\Delta}_s$ for all $s = 1, \ldots, t-1$, we seek to establish the induction steps $\Delta_t \leq \overline{\Delta}_t$, $\Delta_t^\perp \leq \overline{\Delta}_t^\perp$. Starting with $\Delta_t$, we may plug in $\Delta_s \leq \overline{\Delta}_s$, $s \leq t-1$ into the bound on $\Delta_t$ to yield:

$$\Delta_t \leq \sum_{s=1}^{t-1} C_1 \rho^{t-1-s} \varepsilon_s + C_2 \rho^{t-1-s} \tau_s \Delta_s + C_3 \min\{\gamma, \rho^{t-1-s}\} \Delta_s + C_4 \rho^{t-1-s} \Delta_s^2 + C_\perp \rho^{t-1-s} \Delta_s^\perp$$

$$\leq \sum_{s=1}^{t-1} C_1 \rho^{t-1-s} \varepsilon_s + C_2 \rho^{t-1-s} \tau_s \overline{\Delta}_s + C_3 \min\{\gamma, \rho^{t-1-s}\} \overline{\Delta}_s + C_4 \rho^{t-1-s} \overline{\Delta}_s^2 + C_\perp \rho^{t-1-s} \overline{\Delta}_s^\perp.$$

We now treat each summand corresponding to $C_1, C_2, C_3, C_4, C_\perp$ separately. The first term in $C_1$ straightforwardly satisfies $\lesssim \overline{\Delta}_t$ since $\rho \leq \bar{\rho}$ and $C_1 \leq \overline{C}$. Toward bounding the second term, we expand:

$$\sum_{s=1}^{t-1} C_2 \rho^{t-1-s} \tau_s \overline{\Delta}_s = \sum_{s=1}^{t-1} C_2 \rho^{t-1-s} \tau_s \cdot \overline{C} \sum_{k=1}^{s-1} \bar{\rho}^{s-1-k} \varepsilon_k$$

$$\leq C_2 \overline{C} \tau_{\max} \sum_{s=1}^{t-1} \sum_{k=1}^{s-1} \rho^{t-1-s} \bar{\rho}^{s-1-k} \varepsilon_k$$

$$= C_2 \overline{C} \tau_{\max} \sum_{k=1}^{t-2} \varepsilon_k \sum_{s=k+1}^{t-1} \rho^{t-1-s} \bar{\rho}^{s-1-k}$$

$$= C_2 \overline{C} \tau_{\max} \sum_{k=1}^{t-2} \varepsilon_k \rho^{t-k-2} \sum_{j=0}^{t-k-2} (\bar{\rho}/\rho)^j$$

$$= \frac{C_2 \overline{C} \tau_{\max}}{\bar{\rho} - \rho} \sum_{k=1}^{t-2} (\bar{\rho}^{t-k-1} - \rho^{t-k-1}) \varepsilon_k$$

$$\leq \frac{C_2 \overline{C} \tau_{\max}}{\bar{\rho} - \rho} \sum_{s=1}^{t-1} \bar{\rho}^{t-s-1} \varepsilon_s.$$

Therefore, setting $\tau_{\max}$ sufficiently small $\tau_{\max} \lesssim \frac{\bar{\rho} - \rho}{C_2}$ ensures the second summand satisfies $\lesssim \overline{C} \sum_{s=1}^{t-1} \bar{\rho}^{t-s-1} \varepsilon_s = \overline{\Delta}_t$. We may treat the second-order term corresponding to $C_4$ similarly: since by assumption $\varepsilon_s \leq \varepsilon_{\max}$, $s \geq 1$, we have $\overline{\Delta}_s \leq \frac{\overline{C}}{1-\bar{\rho}} \varepsilon_{\max}$ for all $s \geq 1$. Thus, we follow similar steps to bound:

$$\sum_{s=1}^{t-1} C_4 \rho^{t-1-s} \overline{\Delta}_s^2 \leq \frac{C_4 \overline{C} \varepsilon_{\max}}{1-\bar{\rho}} \sum_{s=1}^{t-1} \rho^{t-1-s} \overline{\Delta}_s$$

$$\leq \frac{C_4 \overline{C} \varepsilon_{\max}}{(1-\bar{\rho})(\bar{\rho}-\rho)} \sum_{s=1}^{t-1} \bar{\rho}^{t-s-1} \varepsilon_s.$$

Therefore, setting $\varepsilon_{\max}$ sufficient small $\varepsilon_{\max} \lesssim (1-\bar{\rho})(\bar{\rho}-\rho)/C_4$ ensures the last summand satisfies $\lesssim \overline{C} \sum_{s=1}^{t-1} \bar{\rho}^{t-s-1} \varepsilon_s = \overline{\Delta}_t$. It remains to bound the third term. We first observe the following elementary inequality: given $a, b \in [0,1]$, $\min\{a,b\} \le a^c b^{(1-c)}$ for any $c \in [0,1]$. Applying this to $\min\{\gamma, \rho^{t-1-s}\}$, setting $c = 1 - \log\left(\frac{\bar{\rho}+\rho}{2}\right)/\log(\rho) \in (0,1)$, we have:

$$\sum_{s=1}^{t-1} C_3 \min\{\gamma, \rho^{t-1-s}\} \overline{\Delta}_s \le C_3 \gamma^c \sum_{s=1}^{t-1} (\rho^{1-c})^{t-1-s} \overline{\Delta}_s$$

$$\le \frac{C_3 \overline{C} \gamma^c}{\bar{\rho} - \bar{\rho}+\rho/2} \sum_{s=1}^{t-1} \bar{\rho}^{t-s-1} \varepsilon_s$$

$$= \frac{2 C_3 \overline{C} \gamma^c}{\bar{\rho} - \rho} \sum_{s=1}^{t-1} \bar{\rho}^{t-s-1} \varepsilon_s.$$

In particular, this suggests that as long as $\gamma \lesssim ((\bar{\rho} - \rho)/2C_3)^{1/c}$, the third term satisfies $\lesssim \overline{C} \sum_{s=1}^{t-1} \bar{\rho}^{t-s-1} \varepsilon_s = \overline{\Delta}_t$. Lastly, given the inductive hypothesis on $\{\overline{\Delta}_s^\perp\}$ for $s = 1, \ldots, t-1$, we may bound the $C_\perp$ term:

$$\sum_{s=1}^{t-1} C_\perp \rho^{t-1-s} \overline{\Delta}_s^\perp \le C_\perp \overline{C}_\perp \sum_{s=1}^{t-1} \rho^{t-1-s} \sum_{k=1}^{s-1} \bar{\rho}^{s-1-k}$$

$$\le \frac{C_\perp \overline{C}_\perp}{\bar{\rho} - \rho} \sum_{s=1}^{t-1} \bar{\rho}^{t-1-s} \varepsilon_s.$$

Now, to complete the induction step on $\Delta_t$ and $\Delta_t^\perp$, we determine values of $\overline{C}$ and $\overline{C}_\perp$ in hindsight. We first bound $\Delta_t^\perp$. Leveraging the bounds on the $C_1, C_3$, and $C_4$ terms above, we have:

$$\Delta_t^\perp \le \sum_{s=1}^{t-1} C_1 \rho^{t-1-s} \varepsilon_s + \frac{2 C_3 \overline{C} \gamma^c}{\bar{\rho} - \rho} \bar{\rho}^{t-1-s} \varepsilon_s + \frac{C_4 \overline{C} \varepsilon_{\max}}{(1-\bar{\rho})(\bar{\rho}-\rho)} \bar{\rho}^{t-1-s} \varepsilon_s$$

$$\le \sum_{s=1}^{t-1} \left( C_1 + \frac{2 C_3 \overline{C} \gamma^c}{\bar{\rho} - \rho} + \frac{C_4 \overline{C} \varepsilon_{\max}}{(1-\bar{\rho})(\bar{\rho}-\rho)} \right) \bar{\rho}^{t-s-1} \varepsilon_s$$

We now set $\overline{C}_\perp = 2C_1$ and $\bar{\rho} = \frac{1+\rho}{2}$. Recalling that $c = 1 - \log\left(\frac{\bar{\rho}+\rho}{2}\right)/\log(\rho) = 1 - \log\left(\frac{1+3\rho}{4}\right)/\log(\rho)$, we may verify by calculus or software that $c$ is a monotonically decreasing function of $\rho$, attaining a limit from above of $\lim_{\rho \to 1_-} c = 1/4$, such that $\gamma^c \le \gamma^{1/4}$ for all $\rho \in (0,1), \gamma \le 1$. Therefore, setting:

$$\gamma \le \left( \frac{(\bar{\rho}-\rho)C_1}{2 C_3 \overline{C}} \right)^4 = \left( \frac{(1-\rho)C_1}{8 C_3 \overline{C}} \right)^4 \le 1$$

$$\varepsilon_{\max} \le \frac{(1-\rho)^2 C_1}{8 C_4 \overline{C}},$$

we have $\Delta_t^\perp \le \sum_{s=1}^{t-1} 2C_1 \bar{\rho}^{t-s-1} \varepsilon_s = C_\perp \sum_{s=1}^{t-1} \bar{\rho}^{t-s-1} \varepsilon_s \triangleq \overline{\Delta}_t^\perp$, completing the induction step $\Delta_t^\perp \le \overline{\Delta}_t^\perp$. Given $\overline{C}_\perp = 2C_1$ and $\bar{\rho} = \frac{1+\rho}{2}$, we return to the bound on $\Delta_t$, where we may collect all the bounds on the $C_1, \cdots, C_4, C_\perp$ terms to get:

$$\Delta_t \le \left( C_1 + \frac{C_2 \overline{C} \tau_{\max}}{\bar{\rho} - \rho} + \frac{2 C_3 \overline{C} \gamma^c}{\bar{\rho} - \rho} + \frac{C_4 \overline{C} \varepsilon_{\max}}{(1-\bar{\rho})(\bar{\rho}-\rho)} + \frac{C_\perp \overline{C}_\perp}{\bar{\rho} - \rho} \right) \sum_{s=1}^{t-1} \bar{\rho}^{t-1-s} \varepsilon_s$$

$$= \left( C_1 + \frac{2}{1-\rho} \left( C_2 \overline{C} \tau_{\max} + 2 C_3 \overline{C} \gamma^{1/4} + \frac{2 C_4 \overline{C} \varepsilon_{\max}}{1-\rho} + 2 C_1 C_\perp \right) \right) \sum_{s=1}^{t-1} \bar{\rho}^{t-1-s} \varepsilon_s. \tag{E.12}$$

It remains to set bounds on $\varepsilon_{\max}, \tau_{\max}, \gamma$ and set $\overline{C}$ such that the RHS satisfies $\le \overline{C} \sum_{s=1}^{t-1} \bar{\rho}^{t-1-s} \varepsilon_s$. Intuitively, we may tune $\varepsilon_{\max}, \tau_{\max}, \gamma$ such that the $C_2, C_3, C_4$ terms are as small as needed; however, the $C_\perp$ term cannot be further shrunk. Thus, setting $\overline{C} = C_1\left(1 + \frac{5C_\perp}{1-\rho}\right)$, we may set the

constraints in hindsight:

$$\varepsilon_{\max} \lesssim \frac{(1-\rho)C_1 C_\perp}{\overline{C}C_4} = \frac{(1-\rho)C_\perp}{C_4\left(1 + \frac{5C_\perp}{1-\rho}\right)}$$

$$\tau_{\max} \lesssim \frac{C_1 C_\perp}{C_2 \overline{C}} = \frac{C_\perp}{C_2\left(1 + \frac{5C_\perp}{1-\rho}\right)}$$

$$\gamma \lesssim \left(\frac{C_1 C_\perp}{C_3 \overline{C}}\right)^4 = \left(\frac{C_\perp}{C_3\left(1 + \frac{5C_\perp}{1-\rho}\right)}\right)^4.$$

Collating these constraints with (E.12), we have that under the constraints:

$$\varepsilon_{\max} \lesssim \frac{(1-\rho)^2(1+C_\perp)}{C_4\left(1 + \frac{5C_\perp}{1-\rho}\right)}, \quad \tau_{\max} \lesssim \frac{C_\perp}{C_2\left(1 + \frac{5C_\perp}{1-\rho}\right)}, \quad \gamma \lesssim \left(\frac{(1-\rho)^2(1+C_\perp)}{C_3\left(1 + \frac{5C_\perp}{1-\rho}\right)}\right)^4,$$

we have the desired bound:

$$\Delta_t \le \overline{C} \sum_{s=1}^{t-1} \bar{\rho}^{t-1-s} \varepsilon_s \lesssim C_1 \left(1 + \frac{C_\perp}{1-\rho}\right) \sum_{s=1}^{t-1} \left(\frac{1+\rho}{2}\right)^{t-1-s} \varepsilon_s,$$

completing the induction step $\Delta_t \le \overline{\Delta}_t$ and the full proof.

$$\square$$

To instantiate Lemma E.8, we recall the decomposition of $\mathbf{x}_t^{\hat{\pi}} - \mathbf{x}_t^{\pi^\star}$ into the linear reachable and non-linear components (E.8), and the first-order Taylor expansion of $\hat{\pi}(\mathbf{x}_t^{\hat{\pi}}) - \pi^\star(\mathbf{x}_s^{\hat{\pi}})$ around $\mathbf{x}_s^{\pi^\star}$:

$$\mathbf{x}_t^{\hat{\pi}} - \mathbf{x}_t^{\pi^\star} = \sum_{s=1}^{t-1} \mathbf{A}_{s+1:t}^{\mathrm{cl}} \mathbf{B}_s (\hat{\pi} - \pi^\star)(\mathbf{x}_s^{\hat{\pi}}) + \mathbf{A}_{s+1:t}^{\mathrm{cl}} \mathbf{r}_s^{\mathbf{x}},$$

$$(\hat{\pi} - \pi^\star)(\mathbf{x}_s^{\hat{\pi}}) = (\hat{\pi} - \pi^\star)(\mathbf{x}_s^{\pi^\star}) + \nabla_{\mathbf{x}}(\hat{\pi} - \pi^\star)(\mathbf{x}_s^{\pi^\star})^\top (\mathbf{x}_s^{\hat{\pi}} - \mathbf{x}_s^{\pi^\star}) + \mathbf{r}_s^{\mathbf{u}},$$

where $\mathbf{r}_s^{\mathbf{x}}, \mathbf{r}_s^{\mathbf{u}}$ are the higher-order remainder terms. Further recalling the projection matrices $\mathcal{P}_t \triangleq \mathcal{P}_{\mathcal{R}_t^{\pi^\star}}(\lambda)$ onto the top $\lambda_i \ge \lambda$ eigenspaces of $\mathbf{W}_{1:t}^{\mathbf{u}}$ and the orthogonal complement $\mathcal{P}_t^\perp$ (relative to the reachable subspace $\mathcal{R}_t^{\pi^\star}$), we may write:

$$\mathbf{x}_t^{\hat{\pi}} - \mathbf{x}_t^{\pi^\star}$$

$$= \sum_{s=1}^{t-1} \mathbf{A}_{s+1:t}^{\mathrm{cl}} \mathbf{B}_s (\hat{\pi} - \pi^\star)(\mathbf{x}_s^{\pi^\star}) + (\mathcal{P}_t + \mathcal{P}_t^\perp) \mathbf{A}_{s+1:t}^{\mathrm{cl}} \mathbf{B}_s \nabla_{\mathbf{x}}(\hat{\pi} - \pi^\star)(\mathbf{x}_s^{\pi^\star})^\top (\mathbf{x}_s^{\hat{\pi}} - \mathbf{x}_s^{\pi^\star}) + \left(\mathbf{A}_{s+1:t}^{\mathrm{cl}} \mathbf{B}_s \mathbf{r}_s^{\mathbf{u}} + \mathbf{A}_{s+1:t}^{\mathrm{cl}} \mathbf{r}_s^{\mathbf{x}}\right)$$

$$= \sum_{s=1}^{t-1} \mathbf{A}_{s+1:t}^{\mathrm{cl}} \mathbf{B}_s (\hat{\pi} - \pi^\star)(\mathbf{x}_s^{\pi^\star}) + \mathcal{P}_t^\perp \mathbf{A}_{s+1:t}^{\mathrm{cl}} \mathbf{B}_s \nabla_{\mathbf{x}}(\hat{\pi} - \pi^\star)(\mathbf{x}_s^{\pi^\star})^\top (\mathbf{x}_s^{\hat{\pi}} - \mathbf{x}_s^{\pi^\star})$$

$$\quad + \mathcal{P}_t \mathbf{A}_{s+1:t}^{\mathrm{cl}} \mathbf{B}_s \nabla_{\mathbf{x}}(\hat{\pi} - \pi^\star)(\mathbf{x}_s^{\pi^\star})^\top \mathcal{P}_s (\mathbf{x}_s^{\hat{\pi}} - \mathbf{x}_s^{\pi^\star}) + \mathcal{P}_t \mathbf{A}_{s+1:t}^{\mathrm{cl}} \mathbf{B}_s \nabla_{\mathbf{x}}(\hat{\pi} - \pi^\star)(\mathbf{x}_s^{\pi^\star})^\top \mathcal{P}_s^\perp (\mathbf{x}_s^{\hat{\pi}} - \mathbf{x}_s^{\pi^\star})$$

$$\quad + \left(\mathbf{A}_{s+1:t}^{\mathrm{cl}} \mathbf{B}_s \mathbf{r}_s^{\mathbf{u}} + \mathbf{A}_{s+1:t}^{\mathrm{cl}} \mathbf{r}_s^{\mathbf{x}}\right). \tag{E.13}$$

$$\mathcal{P}_s^\perp (\mathbf{x}_s^{\hat{\pi}} - \mathbf{x}_s^{\pi^\star})$$

$$= \sum_{k=1}^{s-1} \mathcal{P}_s^\perp \mathbf{A}_{k+1:s}^{\mathrm{cl}} \mathbf{B}_k (\hat{\pi} - \pi^\star)(\mathbf{x}_k^{\pi^\star}) + \mathcal{P}_s^\perp \mathbf{A}_{k+1:s}^{\mathrm{cl}} \mathbf{B}_k \nabla_{\mathbf{x}}(\hat{\pi} - \pi^\star)(\mathbf{x}_k^{\pi^\star})^\top (\mathbf{x}_k^{\hat{\pi}} - \mathbf{x}_k^{\pi^\star}) + \mathcal{P}_s^\perp \left(\mathbf{A}_{k+1:s}^{\mathrm{cl}} \mathbf{B}_k \mathbf{r}_k^{\mathbf{u}} + \mathbf{A}_{k+1:s}^{\mathrm{cl}} \mathbf{r}_k^{\mathbf{x}}\right). \tag{E.14}$$

We parse the expressions in (E.8) term by term.

1. First term: $\mathbf{A}_{s+1:t}^{\mathrm{cl}} \mathbf{B}_s (\hat{\pi} - \pi^\star)(\mathbf{x}_s^{\pi^\star})$ corresponds to the contribution of the *on-expert* regression error.

2. Second term: $\mathcal{P}_t^\perp \mathbf{A}_{s+1:t}^{\mathrm{cl}} \mathbf{B}_s \nabla_{\mathbf{x}}(\hat{\pi} - \pi^\star)(\mathbf{x}_s^{\pi^\star})^\top (\mathbf{x}_s^{\hat{\pi}} - \mathbf{x}_s^{\pi^\star})$ corresponds to the first-order policy error in the *low-excitation subspace* (i.e. orthogonal complement of $\mathcal{R}_t^{\pi^\star}(\lambda)$ for some $\lambda$ determined later).

3. Third and fourth terms: $\mathcal{P}_t \mathbf{A}_{s+1:t}^{\mathrm{cl}} \mathbf{B}_s \nabla_{\mathbf{x}}(\hat{\pi} - \pi^\star)(\mathbf{x}_s^{\pi^\star})^\top \mathcal{P}_s(\mathbf{x}_s^{\hat{\pi}} - \mathbf{x}_s^{\pi^\star}) + \mathcal{P}_t \mathbf{A}_{s+1:t}^{\mathrm{cl}} \mathbf{B}_s \nabla_{\mathbf{x}}(\hat{\pi} - \pi^\star)(\mathbf{x}_s^{\pi^\star})^\top \mathcal{P}_s^\perp (\mathbf{x}_s^{\hat{\pi}} - \mathbf{x}_s^{\pi^\star})$ correspond to the time-$t$ reachable component, decomposed further into the time-$s$ reachable and low-excitation components. Intuitively, Proposition E.5 ensures $\mathcal{P}_s \nabla_{\mathbf{x}}(\hat{\pi} - \pi^\star)(\mathbf{x}_s^{\pi^\star})$ is small, while the $\mathcal{P}_s^\perp$ component is automatically small by virtue of lying in the low-excitation subspace, whose evolution is tracked in (E.14).

4. Fifth term: $(\mathbf{A}_{s+1:t}^{\mathrm{cl}} \mathbf{B}_s \mathbf{r}_s^{\mathbf{u}} + \mathbf{A}_{s+1:t}^{\mathrm{cl}} \mathbf{r}_s^{\mathbf{x}})$ corresponds to the second-order residual error controlled by smoothness (Assumption 4.1).

We now work to match (E.13) to the terms in Lemma E.8 Firstly, we recall by definition of $\mathcal{P}_t$ above that $\|\mathcal{P}_t \mathbf{A}_{s+1:t}^{\mathrm{cl}} \mathbf{B}_s\|_{\mathrm{op}} \le \|\mathbf{A}_{s+1:t}^{\mathrm{cl}} \mathbf{B}_s\| \le C_{\mathrm{ISS}} \rho^{t-1-s}$, $\|\mathbf{A}_{s+1:t}^{\mathrm{cl}}\|_{\mathrm{op}} \le C_{\mathrm{ISS}} \rho^{t-1-s}$, and $\|\mathcal{P}_t^\perp \mathbf{A}_{s+1:t}^{\mathrm{cl}} \mathbf{B}_s\|_{\mathrm{op}} \le \min\{\sqrt{\lambda}, C_{\mathrm{ISS}} \rho^{t-1-s}\}$ (cf. Lemma E.1). We then denote $\Delta_t \triangleq \|\mathbf{x}_t^{\hat{\pi}} - \mathbf{x}_t^{\pi^\star}\|$, $\Delta_t^\perp \triangleq \|\mathcal{P}_t^\perp(\mathbf{x}_t^{\hat{\pi}} - \mathbf{x}_t^{\pi^\star})\|$, $\varepsilon_t \triangleq \|(\hat{\pi} - \pi^\star)(\mathbf{x}_t^{\pi^\star})\|$, $\tau_t \triangleq \|\mathcal{P}_t \nabla_{\mathbf{x}}(\hat{\pi} - \pi^\star)(\mathbf{x}_t^{\pi^\star})\|_{\mathrm{op}}$, and $\gamma \triangleq \sqrt{\lambda}/C_{\mathrm{ISS}}$. By Lipschitzness and smoothness (Assumption 4.1), we have $\|\mathcal{P}_t^\perp \nabla_{\mathbf{x}}(\hat{\pi} - \pi^\star)(\mathbf{x}_t^{\pi^\star})\|_{\mathrm{op}} \le 2L_\pi$, $\|\mathbf{r}_s^{\mathbf{x}}\| \le 2C_{\mathrm{reg}}(3 + 2C_{\mathbf{K}}^2 + 2C_\pi^2 \Delta_s^2)\Delta_s^2$ (E.9), $\|\mathbf{r}_s^{\mathbf{u}}\| \le 2C_\pi \Delta_s^2$. Plugging these definitions and bounds into (E.13) and (E.14), we have:

$$\Delta_t \le \sum_{s=1}^{t-1} C_{\mathrm{ISS}} \rho^{t-1-s} \varepsilon_s + 2C_{\mathrm{ISS}} L_\pi \min\{\sqrt{\lambda}/C_{\mathrm{ISS}}, \rho^{t-1-s}\}\Delta_s + 2C_{\mathrm{ISS}} L_\pi \rho^{t-1-s} \tau_s \Delta_s$$
$$+ 2C_{\mathrm{ISS}} L_\pi \rho^{t-1-s} \Delta_s^\perp + \left(2C_{\mathrm{reg}}(3 + 2C_{\mathbf{K}}^2 + 2C_\pi^2 \Delta_s^2) + 2C_\pi\right)\Delta_s^2$$
$$\Delta_t^\perp \le \sum_{s=1}^{t-1} C_{\mathrm{ISS}} \rho^{t-1-s} \varepsilon_s + 2C_{\mathrm{ISS}} L_\pi \min\{\sqrt{\lambda}/C_{\mathrm{ISS}}, \rho^{t-1-s}\}\Delta_s + \left(2C_{\mathrm{reg}}(3 + 2C_{\mathbf{K}}^2 + 2C_\pi^2 \Delta_s^2) + 2C_\pi\right)\Delta_s^2.$$

Under the conditions of Lemma E.8, we have $\Delta_t \le 1$ for $t \ge 1$. Instantiating the constants in Lemma E.8, we set $C_1 = C_{\mathrm{ISS}}, C_2 = 2C_{\mathrm{ISS}} L_\pi, C_3 = 2C_{\mathrm{ISS}} L_\pi, C_4 = 2C_{\mathrm{reg}}(3 + 2C_{\mathbf{K}}^2 + 2C_\pi^2) + 2C_\pi$, $C_\perp = 2C_{\mathrm{ISS}} L_\pi$, which gives the following bound.

**Lemma E.9.** *Let Assumption 4.1 hold. For any initial state $\mathbf{x}_1$, let $\mathbf{x}_1^{\hat{\pi}} = \mathbf{x}_1^{\pi^\star} = \mathbf{x}_1$, and consider the closed-loop trajectories generated by $\hat{\pi}$ and $\pi^\star$. Defining the projections onto the reachable subspace $\mathcal{P}_t \triangleq \mathcal{P}_{\mathcal{R}_t^{\pi^\star}(\lambda)}$ and the corresponding orthogonal complement $\mathcal{P}_t^\perp$ relative to $\mathcal{R}_t^{\pi^\star}$ (Definition E.1). As long as the* on-expert *quantities and excitation-level satisfy:*

$$\|(\hat{\pi} - \pi^\star)(\mathbf{x}_t^{\pi^\star})\| \lesssim \frac{(1-\rho)^3}{C_{\mathrm{reg}}(1 + L_\pi^2 + C_\pi^2) + C_\pi}, \quad \|\mathcal{P}_t \nabla_{\mathbf{x}}(\hat{\pi} - \pi^\star)(\mathbf{x}_t^{\pi^\star})\|_{\mathrm{op}} \lesssim \frac{1}{C_{\mathrm{ISS}} L_\pi}, \forall t \ge 1, \quad \lambda \lesssim \frac{(1-\rho)^9}{C_{\mathrm{ISS}}^2 L_\pi^4},$$

*then we have the following bound on the trajectory error:*

$$\|\mathbf{x}_t^{\hat{\pi}} - \mathbf{x}_t^{\pi^\star}\| \lesssim \overline{C} \sum_{s=1}^{t-1} \bar{\rho}^{t-1-s} \|(\hat{\pi} - \pi^\star)(\mathbf{x}_s^{\pi^\star})\|, \quad \forall t \ge 1, \ \overline{C} \triangleq \frac{C_{\mathrm{ISS}}(1 + C_{\mathrm{ISS}} L_\pi)}{1 - \rho}, \ \bar{\rho} \triangleq \frac{1+\rho}{2}.$$

*Notably, by applying Lemma D.1 and Lemma D.3, we get for any $p \ge 1$:*

$$\left(\sum_{s=1}^t \|\mathbf{x}_s^{\hat{\pi}} - \mathbf{x}_s^{\pi^\star}\|^p\right)^{1/p} \lesssim \frac{C_{\mathrm{ISS}}(1 + C_{\mathrm{ISS}} L_\pi)^2}{(1-\rho)^2} \left(\sum_{s=1}^{t-1} \|(\hat{\pi} - \pi^\star)(\mathbf{x}_s^{\pi^\star})\|^p\right)^{1/p}.$$

We note that Lemma E.9 bounds the trajectory error in terms of the on-expert regression error over the *un-noised* expert distribution. In particular, the only reliance on the noise-injected expert distribution enters through ensuring $\|\mathcal{P}_t \nabla_{\mathbf{x}}(\hat{\pi} - \pi^\star)(\mathbf{x}_t^{\pi^\star})\|_{\mathrm{op}}$ is sufficiently small via Proposition E.5. Intuitively, to convert Lemma E.9 to a bound in terms of $\mathbf{J}_{\mathrm{TRAJ},T}$ and $\mathbf{J}_{\mathrm{DEMO},T}$, we convert the requirements on $\|(\hat{\pi} - \pi^\star)(\mathbf{x}_t^{\pi^\star})\|$ and $\|\mathcal{P}_t \nabla_{\mathbf{x}}(\hat{\pi} - \pi^\star)(\mathbf{x}_t^{\pi^\star})\|_{\mathrm{op}}$ into additive error bounds.

**Proposition E.10.** *Let Assumption 4.1 hold. Let $C_{\mathbf{r}}$ be defined as in Corollary E.1 and $C_{\mathrm{rem}}$ as in Proposition E.6. Let $\mathcal{R}_t^{\pi^\star}(\lambda), t \ge 2$ be the truncated reachable subspaces (Definition E.1), setting*

$\lambda \approx \frac{(1-\rho)^9}{C_{\mathrm{ISS}}^2 L_\pi^4}$. *Recalling* $C_{\mathbf{r}} \triangleq C_\pi + 4C_{\mathrm{reg}}(1 + 4C_{\mathbf{K}}^2)$, *let the noise-scale* $\sigma_{\mathbf{u}} > 0$ *satisfy*

$$\sigma_{\mathbf{u}} \lesssim \min\left\{\frac{\sqrt{\lambda d_u^{-1}}}{C_\pi C_{\mathrm{ISS}}^3 L_\pi}, \lambda d_u^{-1} c_{\mathrm{stab}}^4 C_{\mathbf{r}}^{-1}, \ c_{\mathrm{stab}}\frac{\sqrt{1+4C_{\mathbf{K}}^2}}{C_\pi}\right\} = O_\star(\lambda).$$

*Consider a candidate policy* $\hat{\pi}$. *Define the probabilities:*

$$P_t^{(1)} \triangleq \mathbb{P}\left[r_t^{\mathrm{est}}(\hat{\pi}; \pi^\star) \gtrsim \frac{(1-\rho)^3}{C_{\mathrm{reg}}(1 + L_\pi^2 + C_\pi^2) + C_\pi}\right]$$

$$P_t^{(2)} \triangleq \mathbb{P}\left[\sqrt{\frac{d_u}{\lambda\sigma_{\mathbf{u}}^2}}\left(r_t^{\mathrm{est}}(\hat{\pi}; \pi^\star) + r_t^{\mathrm{est}}(\hat{\pi}; \tilde{\pi}^\star)\right) \gtrsim \frac{1}{C_{\mathrm{ISS}}L_\pi}\right].$$

*Then, for any* $p \geq 1$, *the order-*$p$ *trajectory error may be bounded as:*

$$\mathbf{J}_{\mathrm{TRAJ},p,T}(\hat{\pi})^{1/p} \lesssim \frac{\overline{C}}{1-\bar{\rho}}\mathbf{J}_{\mathrm{DEMO},p,T}(\hat{\pi}; \mathbb{P}_{\pi^\star})^{1/p} + \left(\sum_{t=1}^{T-1}(T-t)(P_t^{(1)} + P_t^{(2)})\right)^{1/p}.$$

*Proof of Proposition E.10.* Define the shorthands for the per-timestep trajectory and estimation errors:

$$r_t^{\mathrm{traj}}(\hat{\pi}, \pi^\star) \triangleq \|\mathbf{x}_t^{\hat{\pi}} - \mathbf{x}_t^{\pi^\star}\| \wedge 1$$

$$r_t^{\mathrm{est}}(\hat{\pi}; \pi^\star) \triangleq \|\hat{\pi}(\mathbf{x}_t^{\pi^\star}) - \pi^\star(\mathbf{x}_t^{\pi^\star})\|$$

$$r_t^{\mathrm{est}}(\hat{\pi}; \tilde{\pi}^\star) \triangleq \mathbb{E}_{\tilde{\mathbf{x}}_t | \mathbf{x}_1^{\pi^\star}}\left[\|\hat{\pi}(\tilde{\mathbf{x}}_t) - \pi^\star(\tilde{\mathbf{x}}_t)\|\right],$$

For a given timestep $t \geq 2$, define the event:

$$\mathcal{E}_t \triangleq \left\{\|(\hat{\pi} - \pi^\star)(\mathbf{x}_s^{\pi^\star})\| \lesssim \frac{(1-\rho)^3}{C_{\mathrm{reg}}(1 + L_\pi^2 + C_\pi^2) + C_\pi}, \ \|\mathcal{P}_s\nabla_{\mathbf{x}}(\hat{\pi} - \pi^\star)(\mathbf{x}_s^{\pi^\star})\|_{\mathrm{op}} \lesssim \frac{1}{C_{\mathrm{ISS}}L_\pi}, \ s \in [t-1]\right\},$$

in other words the burn-in conditions described in Lemma E.9, up to time $t-1$. Then, we may write:

$$\mathbb{E}_D\left[r_t^{\mathrm{traj}}(\hat{\pi}, \pi^\star)\right] = \mathbb{E}_D\left[r_t^{\mathrm{traj}}(\hat{\pi}, \pi^\star)\mathbf{1}_{\mathcal{E}_t}\right] + \mathbb{E}_D\left[r_t^{\mathrm{traj}}(\hat{\pi}, \pi^\star)\mathbf{1}_{\mathcal{E}_t^c}\right]$$

$$\leq \mathbb{E}_D\left[r_t^{\mathrm{traj}}(\hat{\pi}, \pi^\star)\mathbf{1}_{\mathcal{E}_t}\right] + \mathbb{E}_D\left[\mathbf{1}_{\mathcal{E}_t^c}\right]$$

$$\leq \overline{C}\sum_{s=1}^{t-1}\bar{\rho}^{t-1-s}\mathbb{E}_D\left[r_t^{\mathrm{est}}(\hat{\pi}; \pi^\star)\right] + \mathbb{P}[\mathcal{E}_t],$$

where we applied Lemma E.9 to yield the last line, recalling $\overline{C} \triangleq \frac{C_{\mathrm{ISS}}(1+C_{\mathrm{ISS}}L_\pi)}{1-\rho}$, $\bar{\rho} \triangleq \frac{1+\rho}{2}$. To bound $\mathbb{P}[\mathcal{E}_t]$, we have via the union bound:

$$\mathbb{P}[\mathcal{E}_t] \leq \sum_{s=1}^{t-1}\mathbb{P}\left[\|(\hat{\pi} - \pi^\star)(\mathbf{x}_s^{\pi^\star})\| \gtrsim \frac{(1-\rho)^3}{C_{\mathrm{reg}}(1 + L_\pi^2 + C_\pi^2) + C_\pi}\right] + \mathbb{P}\left[\|\mathcal{P}_s\nabla_{\mathbf{x}}(\hat{\pi} - \pi^\star)(\mathbf{x}_s^{\pi^\star})\|_{\mathrm{op}} \gtrsim \frac{1}{C_{\mathrm{ISS}}L_\pi}\right]$$

$$\leq \sum_{s=1}^{t-1}\mathbb{P}\left[r_s^{\mathrm{est}}(\hat{\pi}; \pi^\star) \gtrsim \frac{(1-\rho)^3}{C_{\mathrm{reg}}(1 + L_\pi^2 + C_\pi^2) + C_\pi}\right] + \mathbb{P}\left[\sqrt{\frac{d_u}{\lambda\sigma_{\mathbf{u}}^2}}\left(r_s^{\mathrm{est}}(\hat{\pi}; \pi^\star) + r_s^{\mathrm{est}}(\hat{\pi}; \tilde{\pi}^\star)\right) \gtrsim \frac{1}{C_{\mathrm{ISS}}L_\pi}\right],$$

where we applied Proposition E.5 and the condition on $\sigma_{\mathbf{u}}$ to yield the last line. Therefore, defining:

$$P_t^{(1)} \triangleq \mathbb{P}\left[r_t^{\mathrm{est}}(\hat{\pi}; \pi^\star) \gtrsim \frac{(1-\rho)^3}{C_{\mathrm{reg}}(1 + L_\pi^2 + C_\pi^2) + C_\pi}\right], \quad P_t^{(2)} \triangleq \mathbb{P}\left[\sqrt{\frac{d_u}{\lambda\sigma_{\mathbf{u}}^2}}\left(r_t^{\mathrm{est}}(\hat{\pi}; \pi^\star) + r_t^{\mathrm{est}}(\hat{\pi}; \tilde{\pi}^\star)\right) \gtrsim \frac{1}{C_{\mathrm{ISS}}L_\pi}\right],$$

summing up the bound on $\mathbb{E}_D\left[r_t^{\mathrm{traj}}(\hat{\pi}, \pi^\star)\right]$ over $t \in [T]$ and applying Lemma D.3, we get:

$$\mathbf{J}_{\mathrm{TRAJ},p,T}(\hat{\pi})^{1/p} \lesssim \frac{\overline{C}}{1-\bar{\rho}}\mathbf{J}_{\mathrm{DEMO},p,T}(\hat{\pi}; \mathbb{P}_{\pi^\star})^{1/p} + \left(\sum_{t=1}^{T-1}(T-t)(P_t^{(1)} + P_t^{(2)})\right)^{1/p}$$

$$\leq \frac{\overline{C}}{1-\overline{\rho}} \mathbf{J}_{\text{DEMO},p,T}(\hat{\pi}; \mathbb{P}_{\pi^\star})^{1/p} + T^{1/p} \left( \sum_{t=1}^{T-1} P_t^{(1)} + P_t^{(2)} \right)^{1/p}$$

$$\square$$

We make a few remarks. First off, setting $p = 2$ and trivially upper bounding the triangular factor $T - t \leq T$ and applying Markov's inequality on $P_t^{(1)}, P_t^{(2)}$ (squaring the arguments therein), we may recover the same scaling as in Theorem 4:

$$\mathbf{J}_{\text{TRAJ},p,T}(\hat{\pi}) \lesssim O_\star(T)\,\sigma_{\mathbf{u}}^{-2} \mathbf{J}_{\text{DEMO},p,T}(\hat{\pi}; \mathbb{P}_{\pi^\star,\sigma_{\mathbf{u}},0.5}).$$

Notably, by the statement of Proposition E.10, we now clearly see that the dependence on $\sigma_{\mathbf{u}}$ and $\mathbb{P}_{\pi^\star,\sigma_{\mathbf{u}},\alpha}$ solely comes from $P_t^{(2)}$, which from Lemma E.9 solely arises from the first-order on-expert policy estimation $\nabla_{\mathbf{x}}(\hat{\pi} - \pi^\star)(\mathbf{x}_t^{\pi^\star})$. Importantly, we observe that the horizon-factor $T^{1/p}$ only enters via the conditioning on the localization events, and in fact shrinks as $p \to \infty$ — this precisely lines up with the *horizon-free* scaling of the "max-norm to max-norm" bound $\mathbf{J}_{\text{TRAJ},\infty,T}(\hat{\pi}) \leq O_\star(1)\,\mathbf{J}_{\text{DEMO},\infty,T}(\hat{\pi}; \mathbb{P}_{\pi^\star,\sigma_{\mathbf{u}},\alpha})$ were we to directly work with the "max-to-max" statements from TaSIL-based guarantees such as Proposition E.3 and Proposition E.6, and the $T^{1/2}$ scaling by square-rooting the bound in Theorem 4.

**Shifting horizon-scaling to higher-order.** By virtue of going through the effort of refining a TaSIL-based "max-to-max" argument to the direct sum-to-sum bound of Proposition E.10, we have now isolated the error decomposition of $\mathbf{J}_{\text{TRAJ},p,T}(\hat{\pi})$ into the regression error term $\mathbf{J}_{\text{DEMO},T}(\hat{\pi}; \mathbb{P}_{\pi^\star})$ that is *horizon-free*, and the *horizon-dependent* probabilistic error from conditioning on the localization conditions (viewed alternatively, the *burn-in*) of Proposition E.10. We see that we may apply any Markov-type inequality on $P_t^{(1)}$ and $P_t^{(2)}$: for example, given a positive monotone scalar function $h$:

$$P_t^{(1)} \lesssim h(O_\star(1))\,\mathbb{E}_D[h(r_t^{\text{est}}(\hat{\pi}; \pi^\star)].$$

This necessitates controlling $\mathbb{E}_D[h(r_t^{\text{est}}(\hat{\pi}; \pi^\star)]$; without further assumption, the ability to do so is typically a property of the learning algorithm (and loss function), e.g. square-loss regression $\mathbb{E}_D[\|(\pi^\star - \hat{\pi})(\mathbf{x}_t^{\pi^\star})\|^2]$. However, certain statistical properties precisely convert between different loss functions. A prototypical example is *hypercontractivity*, such as the classic $4 - 2$ hypercontractivity (Wainwright, 2019), satisfied by various sub-Gaussian random variables.

**Definition E.2.** A scalar random variable $X$ is $4 - 2$ hypercontractive if there exists $C_{4\to2} > 0$ such that $\mathbb{E}[X^4] \leq C_{4\to2}\mathbb{E}[X^2]^2$.

Under such an assumption, we may relegate the horizon-scaling localization terms to higher-order.

**Corollary E.3.** *Consider the assumptions and definitions in Proposition E.10. Assume $r_t^{\text{est}}(\hat{\pi}; \pi^\star)$ and $r_t^{\text{est}}(\hat{\pi}; \tilde{\pi}^\star)$ satisfy $4 - 2$ hypercontractivity with constant $C_{4\to2}$ for each $t \in [T-1]$ over $\mathbb{P}_{\pi^\star}$ and $\mathbb{P}_{\pi^\star,\sigma_{\mathbf{u}}}$, respectively. Then, we have:*

$$\mathbf{J}_{\text{TRAJ},2,T}(\hat{\pi}) \leq \left( \frac{\overline{C}}{1-\overline{\rho}} \right)^2 \mathbf{J}_{\text{DEMO},2,T}(\hat{\pi}; \mathbb{P}_{\pi^\star}) + O_\star(C_{4\to2}T)\,\mathbf{J}_{\text{DEMO},2,T}(\hat{\pi}; \mathbb{P}_{\pi^\star,\sigma_{\mathbf{u}},0.5})^2.$$

We note that we may optimize over moment-equivalence conditions; we refer to Ziemann & Tu (2022) for various examples.

**How fundamental is horizon-dependence?** A natural consideration is whether horizon-dependence should be present at all. In our analysis of Proposition E.10, the horizon-dependence arises from conditioning on the on-expert errors being sufficiently small *for each time-step*. We sketch an intuitive argument why horizon-dependence may not be avoidable in general: on-expert regression necessarily only certifies that $\hat{\pi}$ matches $\pi^\star$ around expert-trajectories. Since the nominal dynamics need not be open-loop EISS, sufficiently large regression errors on $O_\star(1)$ time-steps can induce closed-loop unstable dynamics, regardless of ensuing on-expert regression errors. Given a regression oracle that only controls $\mathbf{J}_{\text{DEMO},p,T}(\pi^\star; \mathbb{P})$, and non-stationary expert trajectories, we cannot without further assumption (e.g. algorithmic stability) guarantee error is delocalized across timesteps.

### E.7 LIMITATIONS OF PRIOR APPROACHES

One may wonder what a control-oriented analysis as above buys compared to instantiating prior guarantees in the imitation learning literature. In particular, recent work in `LogLossBC` (Foster et al., 2024) reduces imitation learning to estimation in the Hellinger distance, which is achieved by regressing in the log-loss. However, as observed in Simchowitz et al. (2025), `LogLossBC` (and in the same vein earlier analyses (Ross & Bagnell, 2010; Ross et al., 2011) that rely on the $\{0,1\}$ loss) yields vacuous guarantees even for deterministic experts in continuous action spaces. Therefore, we consider fitting a noised expert and yield guarantees on the trajectory error of the resulting *noisy* rollouts. Contrast this with Theorem 4, where the trajectories used in training may be executed noisily, but the trajectory error bound is measured on rolling out the *noiseless* expert and candidate policies. As a last caveat, we note these works typically bound a cost suboptimality $\mathbf{J}(\hat{\pi}) - \mathbf{J}(\pi^\star)$; this is generally a weaker notion than the trajectory error we consider, which via the formalism of integral probability metrics (IPMs) upper bounds the cost gap (see e.g. Sec 2.3 of Simchowitz et al. (2018)). We now introduce (stochastic) policies $\boldsymbol{\pi} : \mathcal{X} \to \triangle(\mathcal{U})$, where:

$$\boldsymbol{\pi}(\mathbf{x}) = \mathcal{N}(\pi(\mathbf{x}), \sigma_{\mathbf{u}}^2 d_u^{-1} \mathbf{I}_{d_u}), \ \pi \in \Pi. \tag{E.15}$$

In other words, $\boldsymbol{\pi}$ encodes the deterministic policy $\pi$ and a $\approx \sigma_{\mathbf{u}}$-bounded noise-injection process Definition 4.1, where we specify to scaled isotropic Gaussian noise for convenient evaluation of distributional distances.[12] In particular, $\boldsymbol{\pi}^\star$ denotes the *noisy* expert policy. A key step of `LogLossBC` bounds the Hellinger error of a maximum likelihood estimator via a log-loss covering. Define an $\varepsilon$-log-loss-cover $\Pi'$ of $\Pi$: for all $\boldsymbol{\pi} \in \Pi$, there exists $\boldsymbol{\pi}' \in \Pi$ such that for all $\mathbf{x} \in \mathcal{X}$, $\mathbf{u} \in \mathcal{U}$, $\log(\mathbb{P}_{\boldsymbol{\pi}}[\mathbf{u} \mid \mathbf{x}] / \mathbb{P}_{\boldsymbol{\pi}'}[\mathbf{u} \mid \mathbf{x}]) \le \varepsilon$. Denote $N_{\log}(\Pi, \varepsilon)$ as the smallest such cover. Then, the following guarantee on an MLE policy holds Foster et al. (2024, Prop. B.1).

**Proposition E.11.** *Given $n$ trajectories of length $T$ generated by the noised expert $\boldsymbol{\pi}^\star$, define the maximum likelihood policy:*

$$\hat{\boldsymbol{\pi}} \in \arg\max_{\boldsymbol{\pi}} \sum_{i=1}^{n} \sum_{t=1}^{T} \mathbb{P}_{\boldsymbol{\pi}}[\tilde{\mathbf{u}}_t^{(i)} \mid \tilde{\mathbf{x}}_t^{(i)}].$$

*Then, with probability at least $1 - \delta$, the resulting generalization error of $\hat{\boldsymbol{\pi}}$ is bounded by*

$$D_{\mathsf{H}}(\hat{\boldsymbol{\pi}}, \boldsymbol{\pi}^\star) \le \inf_{\varepsilon > 0} \left\{ \frac{6 \log(2N_{\log}(\Pi, \varepsilon)/\delta)}{n} + 4\varepsilon \right\}.$$

Now, we observe for conditional-Gaussian policies (E.15) $\boldsymbol{\pi}, \boldsymbol{\pi}'$, the log-likelihood ratio is given by:

$$\log(\mathbb{P}_{\boldsymbol{\pi}}[\mathbf{u} \mid \mathbf{x}] / \mathbb{P}_{\boldsymbol{\pi}'}[\mathbf{u} \mid \mathbf{x}]) = \frac{d_u}{2\sigma_{\mathbf{u}}^2} \left( \|\pi'(\mathbf{x}) - \mathbf{u}\|^2 - \|\pi(\mathbf{x}) - \mathbf{u}\|^2 \right).$$

Though the log-likelihood ratio is unbounded over the support $\mathbf{u} = \mathbb{R}^{d_u}$, we may truncate the domain, wherein the scaling is similar to $\mathsf{KL}(\boldsymbol{\pi}(\mathbf{x}) \| \boldsymbol{\pi}'(\mathbf{x}))$, from which we have:

$$\mathsf{KL}(\boldsymbol{\pi}(\mathbf{x}) \| \boldsymbol{\pi}'(\mathbf{x})) = \frac{d_u}{2\sigma_{\mathbf{u}}^2} \|\pi(\mathbf{x}) - \pi'(\mathbf{x})\|^2.$$

Notably, this implies an $\varepsilon$-cover in $\max_{\mathbf{x} \in \mathcal{X}} \mathsf{KL}(\boldsymbol{\pi}(\mathbf{x}) \| \boldsymbol{\pi}'(\mathbf{x}))$ is equivalent to a $\sqrt{2\sigma_{\mathbf{u}}^2 d_u^{-1} \varepsilon}$-cover of $\Pi$ in $\mathrm{d}(\pi, \pi') \triangleq \max_{\mathbf{x}} \|\pi(\mathbf{x}) - \pi'(\mathbf{x})\|_2$. For parametric classes with parameters in $\mathbb{R}^{d_\theta}$, $\log N_{\mathrm{d}}(\Pi, \varepsilon) \approx d_\theta \log(1/\varepsilon)$, and thus converting between an $\ell^2$ and KL cover only introduces additional logarithmic factors of $\sigma_{\mathbf{u}}$. However, for non-parametric classes such as those in the lower-bound constructions in Theorem A (Simchowitz et al., 2025), $\log N_d(\Pi, \varepsilon) \approx \mathrm{poly}(1/\varepsilon)$, and thus converting to a KL cover worsens the dependence on $\varepsilon$ and introduces additional *polynomial* factors of $\sigma_{\mathbf{u}}$ and $d_u$. Contrast this with Suboptimal Proposition 4.2 or Theorem 4, where the dependence is always $\sigma_{\mathbf{u}}^{-2}$, regardless of the statistical capacity of $\hat{\pi}, \pi^\star \in \Pi$, since we are covering in $\ell^2$ over the deterministic class, rather than in KL over the conditional-Gaussian class. In either case, we recall that this route of analysis ultimately only controls the rollout cost of *noised* policies. We now establish in the sequel, as insinuated by the upper bound in Suboptimal Proposition 4.2, imitating purely on noised expert demonstrations yields an unavoidable bias scaling with $\sigma_{\mathbf{u}}$.

---

[12]This technically violates boundedness, but this is of minor concern by concentration of measure.

**Suboptimality of only regressing on noise-injected trajectories**  To underscore the importance of imitating on *both* noise-injected and noiseless expert trajectories, we show via a simple example with maximally benign expert closed-loop dynamics that even perfect imitation on noise-injected trajectories necessarily incurs an additive factor in the trajectory error scaling with the smoothness of $\pi^\star - \hat{\pi}$ and the noise-level $\sigma^2$. Consider the system $\mathbf{x}_{t+1} = \mathbf{x}_t + \mathbf{u}_t$, expert policy $\pi^\star(\mathbf{x}_t) = -\mathbf{x}_t$.

**Proposition E.12** (Full ver. of Proposition 4.1). *Let the horizon $T = 3$ and $\mathbf{x}_1^{\pi^\star}$ be fixed with $\|\mathbf{x}_1^{\pi^\star}\| = 1$. Fixing any $\sigma_{\mathbf{u}} \in (0,1)$ and $C_\pi > 0$, let $\mathbf{z} \sim \mathcal{D}_{\mathbf{z}}$ be any log-concave distribution with mean-zero and covariance satisfying $\Sigma_{\mathbf{z}} \succeq \frac{1}{2d_u}\mathbf{I}_{d_u}$, and recall the corresponding noised expert states Definition 4.1. Then, there is a class of policies $\mathcal{P}$ where any $\hat{\pi} \in \mathcal{P}$ satisfies: 1. $\frac{1}{n}\sum_{i=1}^n \|(\pi^\star - \hat{\pi})(\sigma_{\mathbf{u}}\mathbf{z}^{(i)})\| = 0$ with probability $\gtrsim 1 - n\exp(-\sqrt{d_u})$ where $\mathbf{z}^{(i)} \overset{\text{i.i.d}}{\sim} \mathcal{D}_{\mathbf{z}}$, 2. $\hat{\pi}(\mathbf{x}) = \pi^\star(\mathbf{x})$ for all $\|\mathbf{x}\| \geq \sigma_{\mathbf{u}}$, 3. $\hat{\pi}$ is $C_\pi$-smooth. However, the trajectory error induced by rolling out $\hat{\pi}$ is lower-bounded by:*

$$\mathbf{J}_{\text{TRAJ},2,T}(\hat{\pi}) \geq O(1)C_\pi^2\sigma_{\mathbf{u}}^4.$$

In other words, even when the candidate policy fits the expert *perfectly* on noise-injected expert trajectories, the trajectory error of the policies necessarily suffers a drift proportional to the smoothness budget $C_\pi$ and noise-scale $\sigma_{\mathbf{u}}^2$, i.e. policies $\hat{\pi} \in \mathcal{P}$ and $\pi^\star$ are indistinguishable under purely noise-injected trajectories. On the other hand, a single un-noised trajectory from $\hat{\pi}$ and $\pi^\star$ can distinguish between the two policies perfectly.

Noting the expert closed-loop system here satisfies $C_{\text{reg}} = 0$, $C_{\mathbf{K}} = 1$, $C_{\text{stab}} = 1$, we may compare to the key "expectation-to-uniform" step Lemma E.4 in establishing Suboptimal Proposition 4.2, where this lower bound matches the drift in the upper bound of Lemma E.4.

*Proof of Proposition E.12.*  We first write out the noiseless expert's trajectory:

$$\mathbf{x}_1^{\pi^\star} = \mathbf{x}_1, \;\; \mathbf{x}_2^{\pi^\star} = \mathbf{x}_1^{\pi^\star} - \mathbf{x}_1^{\pi^\star} = \mathbf{0}, \;\; \mathbf{x}_3^{\pi^\star} = \mathbf{0}.$$

In other words, the expert reaches $\mathbf{0}$ in one timestep and stays there. Now consider the expert under the noising process $\tilde{\mathbf{u}} = \pi^\star(\tilde{\mathbf{x}}) + \sigma_{\mathbf{u}}\mathbf{z} = -\tilde{\mathbf{x}} + \sigma_{\mathbf{u}}\mathbf{z}$, $\mathbf{z} \sim \mathcal{D}_{\mathbf{z}}$: letting $\mathbf{z}_1, \mathbf{z}_2 \overset{\text{i.i.d}}{\sim} \mathcal{D}_{\mathbf{z}}$ be two i.i.d. draws of noise, we have

$$\tilde{\mathbf{x}}_1 = \mathbf{x}_1^{\pi^\star}, \;\; \tilde{\mathbf{x}}_2 = \tilde{\mathbf{x}}_1 + (-\tilde{\mathbf{x}}_1 + \sigma_{\mathbf{u}}\mathbf{z}_1) = \sigma_{\mathbf{u}}\mathbf{z}_1, \;\; \tilde{\mathbf{x}}_3 = \tilde{\mathbf{x}}_2 + (-\tilde{\mathbf{x}}_2 + \sigma_{\mathbf{u}}\mathbf{z}_2) = \sigma_{\mathbf{u}}\mathbf{z}_2.$$

In other words, after timestep 1, since the expert policy always perfectly cancels out the previous state, the distribution of noised expert states is identical to the noise distribution $\sigma_{\mathbf{u}}\mathbf{z}$. Therefore, the intuition for the lower bound is as follows: by concentration of measure, any "usual" distribution (e.g. log-concave, subgaussian) that has non-vanishing excitation, as captured by the second moment $\Sigma_{\mathbf{z}} \succeq c\frac{1}{d_u}\mathbf{I}_{d_u}$, necessarily concentrates on the $O(1)\sigma_{\mathbf{u}}$-scaled unit sphere $\mathbb{S}^{d_u}$.[13] Therefore, given $n$ independent trajectories, i.e. $n$ independent draws $\{(\mathbf{z}_1^{(i)}, \mathbf{z}_2^{(i)})\}$, with high probability we do not see any states $\tilde{\mathbf{x}}_1^{(i)}, \tilde{\mathbf{x}}_2^{(i)}$ within an $\approx \sigma_{\mathbf{u}}$ radius of the origin. This is formalized in the following lemma (Paouris, 2006; Adamczak et al., 2014).

**Lemma E.13** (Paouris' Inequality (Paouris, 2006)). *Let $\mathbf{z}$ be a log-concave random vector that with zero-mean and identity covariance supported on $\mathbb{R}^d$. Then, there exists a universal constant $c > 0$ such that for any $\gamma \geq 1$: $\mathbb{P}\left[\|\mathbf{z}\| \geq c\gamma\sqrt{d}\right] \leq \exp(-\gamma\sqrt{d})$.*

Therefore, re-scaling $\mathbf{z}$ such that $\Sigma_{\mathbf{z}} \succeq \frac{1}{2d_u}\mathbf{I}_{d_u}$ and setting $\gamma = \frac{\sqrt{2}}{2c}$, this implies: $\mathbb{P}[\|\mathbf{z}\| \geq 1/2] \leq \exp(-\frac{\sqrt{2}}{2c}\sqrt{d_u}) \approx \exp(-\sqrt{d_u})$. Union bounding over $i = 1, \ldots, n$, we have $\mathbb{P}[\|\mathbf{z}^{(i)}\| \geq 1/2, \; \forall i \in [n]] \gtrsim 1 - n\exp(-\sqrt{d_u})$.

Given that the noised expert states concentrate $\approx \sigma_{\mathbf{u}}$ away from the origin with overwhelming probability, we now task ourselves to constructing a family of candidate policies $\hat{\pi}$ that maximally deviate from the expert policy at the origin, given its smoothness budget $C_\pi$. This can be achieved, for example, by a straightforward bump function construction.

---

[13]We note that when $\mathcal{D}_{\mathbf{z}}$ is the uniform distribution on the unit sphere $\mathbb{S}^{d_u}$, then we may interchange the high-probability guarantee with expectation $\mathbb{E}[\|(\hat{\pi} - \pi^\star)(\sigma_{\mathbf{u}}\mathbf{z})\|^2] = 0$.

**Lemma E.14** (Bump function existence, c.f. Simchowitz et al. (2025, Lemma A.15)). *For any $d \in \mathbb{N}$, we may construct a function $\text{bump}_d(\cdot) : \mathbb{R}^d \to \mathbb{R}$, $\text{bump}_d \in C^\infty$, such that the following hold:*

1. $\text{bump}_d(\mathbf{z}) = 1$ *for all* $\|\mathbf{z}\| \leq 1$.
2. $\text{bump}_d(\mathbf{z}) = 0$ *for all* $\|\mathbf{z}\| \geq 2$.
3. *For each $p \geq 1$ and $\mathbf{z} \in \mathbb{R}^d$, $\|\nabla_p \text{bump}_d(\mathbf{z})\|_{\text{op}} \leq c_p$, where $c_p > 0$ is a constant depending on $p > 0$ but independent of dimension d.*
4. $\nabla^p \text{bump}_d(\mathbf{z}) = \mathbf{0}$ *for all* $\|\mathbf{z}\| \geq 2$.

In other words, we may construct a function that always outputs $1$ in the unit sphere, and $0$ outside of the radius $2$ sphere, and has bounded-norm derivatives in between. Before proceeding with the construction, we observe that $\pi^\star(\mathbf{x}) = -\mathbf{x}$ is a linear function, and thus satisfies $\nabla^2 \pi^\star(\mathbf{x}) = \mathbf{0}$ everywhere. For a given $\sigma_\mathbf{u} > 0$ and smoothness budget $C_\pi > 0$, it therefore suffices to determine $\Delta\pi = \hat{\pi} - \pi^\star$ that satisfies the properties:

1. $\Delta\pi(\mathbf{x}) = \mathbf{0}$ for all $\|\mathbf{x}\| \geq \sigma_\mathbf{u}/2$.
2. $\|\nabla^2 \Delta\pi(\mathbf{x})\|_{\text{op}} \leq C_\pi$.

We construct $\Delta\pi$ as follows. Fix any $\mathbf{v} \in \mathbb{S}^{d_u}$, and let $\text{bump}_{d_x}(\cdot)$ be a function that satisfies the properties in Lemma E.14. We propose:

$$\Delta\pi(\mathbf{x}) \triangleq L\text{bump}_{d_x}\left(\frac{\mathbf{x}}{\sigma_\mathbf{u}/4}\right)\mathbf{v}, \tag{E.16}$$

where $L > 0$ is a constant to be determined later. We observe that by construction: $\Delta\pi = \mathbf{0}$ for all $\|\mathbf{x}\| \geq \sigma_\mathbf{u}/2$, $\|\Delta\pi(\mathbf{0})\| = L$, and $\|\nabla_\mathbf{x}^p \Delta\pi(\mathbf{x})\|_{\text{op}} = L\|\nabla_\mathbf{x}^p \text{bump}_{d_x}(4\mathbf{x}/\sigma_\mathbf{u})\|_{\text{op}} = Lc_p \left(\frac{4}{\sigma_\mathbf{u}}\right)^p$.

Therefore, to ensure $\Delta\pi$ is $C_\pi$-smooth, this informs choosing $L = \frac{\sigma_\mathbf{u}^2}{16c_2}C_\pi$. Therefore, the resulting policy $\hat{\pi} = \pi^\star + \Delta\pi$ satisfies the following properties:

1. $\hat{\pi}$ is $C_\pi$-smooth.
2. $\|\hat{\pi}(\mathbf{0})\| = C_\pi \frac{\sigma_\mathbf{u}^2}{16c_2}$.
3. $\hat{\pi}(\mathbf{x}) = \pi^\star(\mathbf{x})$ for all $\|\mathbf{x}\| \geq \sigma_\mathbf{u}/2$. In particular, by Lemma E.13 that $\frac{1}{n}\sum_{i=1}^n (\hat{\pi} - \pi^\star)(\sigma_\mathbf{u}\mathbf{z}^{(i)}) = \mathbf{0}$ with probability $\gtrsim 1 - n\exp(-\sqrt{d_u})$.

Now, we roll out $\hat{\pi}$ and $\pi^\star$ without noise injection. We have as aforementioned $\mathbf{x}_2^{\pi^\star} = \mathbf{x}_3^{\pi^\star} = \mathbf{0}$. On the other hand, since $\sigma_\mathbf{u} < 1$, we have $\hat{\pi}(\mathbf{x}_1^{\pi^\star}) = \pi^\star(\mathbf{x}_1^{\pi^\star})$ and thus $\mathbf{x}_2^{\hat{\pi}} = \mathbf{x}_2^{\pi^\star} = \mathbf{0}$. However, by our construction of $\hat{\pi}$, $\mathbf{x}_3^{\hat{\pi}} = \mathbf{x}_2^{\hat{\pi}} + \hat{\pi}(\mathbf{x}_2^{\hat{\pi}}) = \hat{\pi}(\mathbf{0}) = C_\pi\sigma_\mathbf{u}^2(16c_2)^{-1}$, and thus:

$$\max_{t=1,2,3}\|\mathbf{x}_t^{\hat{\pi}} - \mathbf{x}_t^{\pi^\star}\| = \|\mathbf{x}_3^{\hat{\pi}} - \mathbf{x}_3^{\pi^\star}\| \geq O(1)C_\pi\sigma_\mathbf{u}^2.$$

After squaring both sides, we see the left-hand side is precisely $\mathbf{J}_{\text{TRAJ},2,T}^{\mathbf{x}}$, which is trivially upper bounded by $\mathbf{J}_{\text{TRAJ},2,T}$.

$\square$

We note extending the construction above to general, possibly improper learners, follows by noting that $\hat{\pi}$ and $\pi^\star$ are constrained to generate near-indistinguishable trajectories on $\mathbb{P}_{\pi^\star,\sigma_\mathbf{u}}$; we refer to Simchowitz et al. (2025) detailed minimax formulations. This lower bound establishes the unavoidable drift from noise-injection due to nonlinearity of the expert policy, thus highlighting the necessity of imitating on a dataset consisting of *both* noise-injected and clean expert trajectories; though, as discussed in the previous section, the exact proportion of each is not necessarily important.

## F EXPERIMENTAL VALIDATION

**Action Chunking.** To validate our predictions about the **stability-theoretic** benefits of action-chunking, we propose experiments on robotic imitation tasks in the `robomimic` framework (Mandlekar et al., 2022). In particular, we pre-train a performant state-based, deterministic expert policy on `robomimic` data, which we then roll out to generate training data. We fit models of the same architecture except the final output dimension of varying prediction horizons. We then *execute* varying

numbers of the predicted actions in open-loop and evaluate the resulting success rate. We observe the findings in Figure 2; all experiment details can be found in Appendix G. In short, we find that:

- **Executing action chunks** matters more than simply predicting longer sequences of actions. This demonstrates the action-chunking is more than a simple consequence of representation learning, or a simulation of receding-horizon control.

- The merits of action-chunking remain showcased in **deterministic, state-based control**. This reveals that action-chunking still improves performance independently of partial observability or compatibility with generative control policies.

- **End-effector control** enables the benefits of action-chunking. This is because end-effector control renders the closed-loop between system state and end-effector prediction incrementally stable (Block et al., 2024). Hence, the low-level end-effector controller transforms imitating the position policy to taking place in an open-loop stable dynamical system, precisely the regime where we prescribe our AC guarantees. Accordingly, in MuJoCo tasks that lack this property, we find that naively action-chunking hurts, not helps, performance—see Figure 3.

We emphasize the above remarks are not to rule out the role of non-Markovianity and representation learning; it is likely that these contribute further, e.g. AC can demonstrably prevent "stalling" from demonstrations with pauses. Rather, our results should be understood as stating that, instead of the benefits of action-chunking existing *in tension* with control—as controls folk-knowledge typically cautions against open-loop execution—it can be *naturally explained* by a control-theoretic perspective.

**Noise Injection.** We seek to validate our hypotheses about the exploratory benefits of noise-injection, making particular note to the algorithmic suggestions that our theoretical analysis reveal. We propose experiments on MuJoCo continuous control environments, where we seek to imitate pre-trained expert policies. We observe the findings across Figure 1 and 3. To summarize:

- **Noise injection as in Practice 2 provides the exploration necessary to mitigate compounding errors**, increasing performance on par with iteratively interactive methods such as DAgger (Ross et al., 2011) and DART (Laskey et al., 2017). We note Practice 2 collects data in one shot, without ever observing learned policy rollouts.

- **Larger noise scales $\sigma_{\mathbf{u}}$ (within tolerance) improve performance**, in contrast to prior understanding (cf. Proposition 4.1, Suboptimal Proposition 4.2) which necessitates $\sigma_{\mathbf{u}}$ set proportional to $\mathbf{J}_{\mathrm{DEMO},T}(\hat{\pi}; \mathbb{P}_{\mathrm{demo}})$, i.e. very small for policies with low on-expert error.

- **A *mixture* of noise-injected and clean expert trajectories is beneficial**, and the difference is small when provided more data, as suggested by Eq. (4.1). This matches the theoretical intuition that noise-injection is necessary up until $\hat{\pi}$ is "locally stabilized" *sufficiently well* around $\mathbf{x}_t^{\pi^\star}$ (Proposition 4.3 and 4.4), and thus only enters the trajectory error as a higher-order term, i.e., we only need "a sufficient amount" of noise-injection.

## G  EXPERIMENT DETAILS

### G.1  SYNTHETIC "CHALLENGING EXAMPLE" (SIMCHOWITZ ET AL., 2025).

- **Model:** two-hidden layers of dimension 256 and GELU activations (Hendrycks & Gimpel, 2016). We remove layer biases to introduce a mild inductive bias of the model outputting $\mathbf{0}$ at the origin.

- **Optimizer and training:** we use the AdamW optimizer (Loshchilov, 2017) with a cosine decay learning rate schedule (Loshchilov & Hutter, 2016), with initial learning rate of 0.001 and other hyperparameters set as default. The models are trained for 4000 epochs with a batch size of 64. Evaluation statistics of each model are computed on an independent sample of 100 trajectories.

For the action-chunkng experiment in Figure 1, we consider a synthetic nonlinear system that is open-loop EISS, and closed-loop EISS under a deterministic expert, as constructed in Appendix E.1 and J of Simchowitz et al. (2025). In particular, we first consider matrices:

$$\mathbf{A} = \begin{bmatrix} 1+\mu & \frac{3}{2}\mu \\ -\frac{3}{2}\mu & 1-2\mu \end{bmatrix}, \quad \mathbf{K} = \begin{bmatrix} -(1+\mu) & -\frac{3}{2}\mu \\ \frac{3}{2}\mu & 0 \end{bmatrix},$$

where we set $\mu = 1/8$. We then embed these matrices into a 6-dimensional state and input space:

$$\tilde{\mathbf{A}} = \begin{bmatrix} \mathbf{A} & \mathbf{0} \\ \mathbf{0} & \mathbf{0} \end{bmatrix}, \quad \tilde{\mathbf{K}} = \begin{bmatrix} \mathbf{K} & \mathbf{0} \\ \mathbf{0} & \mathbf{0} \end{bmatrix} \in \mathbb{R}^{6 \times 6}.$$

These matrices are respectively embedded into smooth nonlinear dynamics $f$ and expert policy $\pi^\star$ as described in Construction E.1 (Simchowitz et al., 2025). For the requisite smooth function $g$ in the embedding, we generate a randomly initialized neural network with 1-hidden layer of dimension 16, with weights following the Xavier normal initialization (Glorot & Bengio, 2010) and biases sampled entrywise from $\mathrm{Unif}([-1, 1])$; note that we only generate this network *once* to complete the problem instance. Having generated a "hard instance" indexed by $(\tilde{\mathbf{A}}, \tilde{\mathbf{K}}, g)$, the training data is comprised of 100 independent trajectories of length 64 rolled out under the expert policy $\pi^\star$. For Figure 1, we train a behavior cloning agent; for each chunk length, the BC policy takes the form as described above, with the sole difference being the output dimension, which equates to $6 \times$ chunk_len. Given the training recipe above, all BC policies across chunk lengths $\in \{1, 2, 4, 8, 16\}$ achieve training error of at most $10^{-6}$, attaining near perfect imitation on the expert data.

Crucially, we note that, in contrast to our formal prescription in Practice 1, we are not enforcing that the chunked BC policies accompany a simulated dynamics that it stabilizes, and purely treat the policy as an $\ell$-step action predictor. Beyond the soft inductive biases in ensuring the policies output $\mathbf{0}$ at the origin, we make no effort to enforce "simulated" stability, yet we still see the clear stabilization benefits of action-chunking in Figure 1.

## G.2  ACTION-CHUNKING EXPERIMENTS ON ROBOMIMIC

- **Model:** for the expert and learner policies, we use a proprietary flow-matching policy parameterization being developed in concurrent work. In short, the policy backbone is a "Chi-UNet" architecture adopted from the seminal Diffusion Policy (Chi et al., 2023), which is built on top of a 1-D U-Net (Janner et al., 2022) with FiLM conditioning (Perez et al., 2018) on the observation and flow-timestep $t \in [0, 1]$. All feed-forward hidden-layer components have width 512 across the expert and learned policies. For expert trajectory collection and learned policy evaluation, **we set the flow-policy to deterministic mode**, i.e. setting the prior distribution to all-zeros.

- **Optimizer and training:** we use the AdamW optimizer (Loshchilov, 2017) with a cosine decay learning rate schedule (Loshchilov & Hutter, 2016) decaying across the training horizon, with initial learning rate of $0.001$ and other hyperparameters set as default. The horizon $\in \{4, 8, 16\}$ models are trained for 1000 epochs with a batch size of 1024, and the horizon $\in \{32, 64\}$ models are trained for 2500 epochs to account for the larger output dimension and thus more difficult prediction problem. For a given training trajectory budget (e.g. 50 or 100), we collect trajectories where the task is *successfully* executed.

- **Evaluation:** For the plots in Figure 2, we train three independent models per configuration and record success over 50 evaluation trajectories each. We then plot the percentile boostrap estimators (median, shaded 10-90 percentile) across the $3 \times 50$ evaluations as informed by Agarwal et al. (2021).

We recall prior hypotheses about action-chunking's effectiveness include: 1. robustness to non-Markovianity/partial-observability, 2. amenability to multi-modal prediction, 3. improved representation learning, and 4. simulation of receding-horizon control. We consider the following set-up: we first train a flow-matching (i.e., generative) *position control* policy on *full-state* robomimic data to yield a performant expert policy, after which we generate imitation data by rolling out the expert policy in *deterministic mode*. By construction, this ensures the imitation data comes from a *fully-observable, deterministic, Markovian* expert policy, ablating away contributions from the first two points. This leaves *improved representation learning* and *simulating receding-horizon control* as the remaining alternate hypotheses.

On the other hand, our analysis in Section 3 suggests that: 1. *executing* open-loop chunks of actions is key, 2. performant chunk-lengths can be relatively short: our theory predicts *logarithmic* in system parameters is sufficient (Theorem 1). The second point is important, as slow-growing benefits of action-chunking would come into conflict with the perils of open-loop control. We remark that imitating position control (i.e., end-effector control) as opposed to joint/torque control crucially aligns with our *key condition* for prescribing action-chunking: stability of the open-loop dynamics

(Assumption 3.1). Position control is implemented by providing mid-level position commands that are tracked by high-frequency joint/torque controllers—this low-level tracking ensures that given a plan of desired positions, reasonable differences in commanded versus realized positions do not lead to diverging trajectories (Definition 2.1). This hierarchical set-up is the backbone of modern robot learning, and failure modes of direct imitation *sans* tracking/stabilization are well-documented; see e.g., (Block et al., 2024; Mehta et al., 2025).

To test our hypotheses, given the collected expert data, we consider training multi-step imitators with the same architecture as the expert except the output dimension, that predict varying horizons of actions `horizon` $\in \{4, 8, 16, 32\}$ conditioned on the current state, and *evaluate* them at varying chunk lengths $\{1, 2, \ldots, \text{horizon}\}$ measuring task success. We display the results in Figure 2. Though there is some gain from longer prediction horizons, in line with learning-theoretic work (Venkatraman et al., 2015; Somalwar et al., 2025), it is less evident in lower-data regimes (see Figure 2, **right**) and is counteracted in longer horizons due to greater strain on a given architectural capacity. On the other hand, the greater gain regardless of `horizon` come from longer *evaluation chunk lengths* up to a point before decaying predictably due to open-loop control—evaluating 32 actions at $20\text{Hz}$ control frequency corresponds to $\sim 1.5$ seconds in open-loop.

We also note that noise-injection, while prescribed for the open-loop unstable setting, is also applicable in this setting: see Figure 2, **right**, where we add $\sigma = 0.05$-scaled $\text{Unif}(\mathbb{S}^{d_u})$ noise when executing the expert's actions for half the training trajectories. However, we remark that for long chunks, action-chunking removes supervision from intervening states, thus training long-chunk predictors on noise-injected trajectories turns beneficial local exploration into uncertainty, since the predictor must fit chunks of actions to seemingly noisy targets.

We note that though Theorem 1 pairs a candidate policy with a simulated dynamics to perform chunking, here we are simply fitting a multi-step action predictor. Despite this, we still observe the stark benefits of chunking. This hints at the role of architectural inductive bias.

### G.3 NOISE INJECTION EXPERIMENTS ON MUJOCO

- **Model:** two-hidden layers of dimension 256 and GELU activations (Hendrycks & Gimpel, 2016). We also additionally place batch-norm layers after the input and first hidden layers.

- **Optimizer and training:** we use the AdamW optimizer (Loshchilov, 2017) with a cosine decay learning rate schedule (Loshchilov & Hutter, 2016), with initial learning rate of $0.001$ and other hyperparameters set as default. The models are trained for 4000 epochs with a batch size of 100.

- **Evaluation:** For the reward $\times$ # training trajectories plots across Figure 1 and Figure 3, we train 5 independent models for each configuration, and compute evaluation statistics on an independent sample of 100 trajectories. We then compute the percentile boostrap estimators for the median and shaded 10-90 percentile.

For the noise injection experiments depicted in Figure 1 (**left**) and Figure 3, we used the HalfCheetah-v5 and Humaniod-v5 environments through the Gymnasium library (Towers et al., 2024). The expert policy for HalfCheetah is a Soft Actor-Critic (Haarnoja et al., 2018) RL policy pre-trained using the StableBaselines3 library (Raffin et al., 2021), downloaded from Huggingface [url], and the Humanoid expert is a Truncated Quantile Critic (Kuznetsov et al., 2020) RL policy obtained similarly. When collecting expert demonstrations, we set `deterministic=True`. Given noise-scale $\sigma_{\mathbf{u}}$, we use scaled spherical noise $\sigma_{\mathbf{u}} \cdot \text{Unif}(\mathbb{S}^{d_u})$ as the noise-injection distribution. We set the trajectory horizons at $T = 300$ timesteps. For each figure specifically:

- Figure 1 (**center + right**): We sweep over noise-levels $\sigma_{\mathbf{u}} \in \{0.0, 0.01, 0.1, 0.5, 1.0\}$ for HalfCheetah and $\sigma_{\mathbf{u}} \in \{0.05, 0.1, 0.25\}$ for Humanoid, fixing the proportion of clean trajectories at 50%, equivalent to imitating over $\mathbb{P}_{\pi^\star, \sigma_{\mathbf{u}}, 0.5}$ from Practice 2. We note that noise-level $0.0$ corresponds to vanilla behavior cloning. Since the Humanoid environment terminates early when the agent falls over, we crudely pick the upper noise-limit for Humanoid by setting it such that the total collected timesteps is $80\%$ of the maximum possible #traj $\times 300$. We similarly run DAGGER (Ross & Bagnell, 2010) and DART (Laskey et al., 2017), where we split a given training trajectory budget into 5 equal rounds of expert trajectory collection model updates. We found a performant mixing parameter for DAGGER to be $\beta = 0.5$.

- Figure 3 (**left**): We consider for $\sigma_{\mathbf{u}} \in \{0.5, 1.0\}$ the effect of recording clean versus noisy action labels. Recall that Practice 2 prescribes *executing* expert actions noisily $\tilde{\mathbf{x}}_{t+1} = f(\tilde{\mathbf{x}}_t, \pi^\star(\tilde{\mathbf{x}}_t) + \sigma_{\mathbf{u}}\mathbf{z}_t)$, but records the clean action label $\tilde{\mathbf{u}}_t = \pi^\star(\tilde{\mathbf{x}}_t)$. On the other hand, the "RL-theoretic" approach (Appendix E.7), in order to achieve density, also requires recording the noisy label $\tilde{\mathbf{u}}_t = \pi^\star(\tilde{\mathbf{x}}_t) + \sigma_{\mathbf{u}}\mathbf{z}_t$. We fix the proportion of clean trajectories to 0.0 for both set-ups for fair comparison.

- Figure 3 (**center**): We sweep over proportion of clean trajectories $\alpha \in \{0.0, 0.2, 0.5, 0.8, 1.0\}$, holding noise-level $\sigma_{\mathbf{u}} = 0.5$ fixed. We note that $\alpha = 1.0$ (no noise-injection) corresponds to vanilla behavior cloning, and $\alpha = 0.0$ corresponds to pure noise-injection $\mathbb{P}_{\pi^\star, \sigma_{\mathbf{u}}}$ (see Proposition 4.1).

- Figure 3 (**right**): We consider fitting multi-step chunking policies. We naively extend the output dimension to `chunk_length` $\times \, d_u$ and play the full chunk open-loop. We note this does not necessarily rule out some form of advanced action-chunking recipe from enabling performance; however, where naive multi-step predictors benefit in the `robomimic` set-up, they do not appear to do so here, likely due to the open-loop instability of the environments (e.g. lacking low-level stabilizing controllers).

