# OpenReview forum: "Action Chunking and Data Augmentation Yield Exponential Improvements in Behavior Cloning for Continuous Spaces"
_ICLR.cc/2026/Conference — ICLR 2026 Poster_

### Official Review · Reviewer_Y1gT · 2025-10-27

**Soundness:** 1
**Presentation:** 1
**Contribution:** 1
**Rating:** 0
**Confidence:** 3

**Summary:**

The main point of the work seems to be that predicting a suitable sequence - chunk -  of actions ( 2, 4, 8, ...) instead of a single action can be beneficial for imitation learning of certain tasks. The same finding holds for adding noise to training data. This are related to the compounding error problem of offline learning (small deviations in the real world accumulate). In online learning these problems are solved by using the current policy in sampling new data (DaGGER).

I believe these findings are somewhat known in the scientific community. The authors claim to derive "theoretical guarantees" (lines 46-47), but their writing is so difficult to follow that I cannot understand what the authors mean by those. Moreover, Sections 5 and 6 do not anymore discuss about the theoretical guarantees and their consequences which makes it confusing what really are the main contributions of this work.

**Strengths:**

Action chunking is still a rather new methodology in imitation learning and therefore makes it an interesting topic to study and investigate its properties and reasons why it is sometimes beneficial.

**Weaknesses:**

**Major weaknesses:**

 - This paper writing style is very complicated and therefore I cannot really follow what are the main research questions, contributions, and the main findings. The authors should simplify the text and clarify the contributions. In this form I cannot accept the work.

**Moderate weaknesses:**

 - The main contribution is rather complicated to be easily understandable: "We provide the first theoretical guarantees in continuous state-action IL for interventions that provably prevent compounding error without iterative expert feedback." This contribution leads to *two practices* that result two "surprising" takeaways. So contribution is limited to practices that result to surprising takeaways. Could this line of thinking be simplified to something easier to understand and follow?

 - Text is partially difficult to follow (perhaps too extensive use of AI language correction). For example, the second paragraph of Introduction: first sentence is 5 plus lines and the last sentence is more than 4 lines. Such monster sentences appear all over the manuscript.

 - Usage of figures is not good. For example, Figure 1 on the page 3 is not referred until page 9. Figures should flow with the text and not be randomly around the manuscript.

 - Main findings from the experimental part (page 9) are difficult to comprehend. For example, in Figure 2 it seems that prediction horizon and action chunk 16 result to the best result (should not come as surprise that the both are them same). By adding noise, some of the prediction horizons improve (e.g. 8), but on the other hand the best performing 16 performance degrades. So the effects of noise and chunking are not complementary always. [Btw. "left" and "right" in a figure with three images is confusing, why not (a), (b), and (c)? Moreover, marking 2³ instead of 8 is odd until the purpose really is to confuse the reader from the obvious]

**Minor weaknesses:**

 - Citations should be updated. Some of them are listed as ArXiv papers despite that the paper have already been published in a conference or journal (e.g. Chi et al. 2023)

 - Use of colors to emphasize parts of the text is already a bit exhaustive. On page two there are normal text, bold text, blue text, and red text.

**Questions:**

If the authors can restate their research question(s), main contributions, and findings in more concise manner

---

### Official Review · Reviewer_zW76 · 2025-11-01

**Soundness:** 3
**Presentation:** 2
**Contribution:** 3
**Rating:** 6
**Confidence:** 2

**Summary:**

This paper provides theoretical justification for two major ways that people have done imitation learning in robotics: action chunking (where an agent chooses K > 1 actions in parallel to execute) and data augmentation (where one can learn a generative model to generate more useful synthetic expert data to imitate), particularly in a state-based continuous control setting with deterministic dynamics.

Their theory relies on the notion of EISS, in which perturbations to actions and next states have a bounded effect over future state distributions under the dynamics across time. In particular, there is a clear upper bound on the deviation between states far away in time, depending on the initial perturbation amount. This, combined with Lipschitzness of the policy class, yield upper bounds for compunding error that scale polynomially with respect to the Lipschitz coefficient of the policy and the EISS coefficient of the dynamics. Furthermore, their theory shows that one can achieve a strong compounding error bound with a chunked policy, where the compounding error coefficient is independent of the horizon of the problem, which is a strong result.

In the case where the dynamics are not open-loop EISS, then the authors propose noise injection, where one injects uniform noise to the actions selected by the learner at each step. In this particular case, the imitation learning procedure is to then imitate both over clean expert trajectories and "noised" expert trajectories.

**Strengths:**

I am not a theoretical expert nor a control theory expert, but it seems that the theory is decently sound, at least from the control theory perspective. This was corroborated by looking into Simchowitz et. al. (2025), the closest work to this.

The content is also quite novel, as no one (at least in robotics) has looking into an analysis of data augmentation and action chunking (aside from Simchowitz et al. (2025) partly), as it seemed to be more of a practical trick than a deeply analyzed method. Moreover, the authors analyze different components of learning that of Simchowitz et. al. (2025), notably the policy parameterization (chunking) and data collection (e.g. noisy expert), different from the learning algorithm perspective that the prior paper takes.

The empirical results were also quite compelling, and showed that the combination of both chunking horizon (e.g. Figure 2, middle & right) and noising of both actions (e.g. Figure 2, right) and trajectory data (e.g. Figure 3) yield improvements in imitation performance beyond that of standard BC.

**Weaknesses:**

While I am not a control theory person, it was quite difficult for me to follow along with the theory in this work. For instance, I didn't know what the difference between "closed-loop" and "open-loop" was, particularly when it came to defining the EISS notion (e.g. Theorem A). As such, while I was able to semi-follow the theory on action chunking reasonably well, I still had some questions regarding definitions. The same went for the noising analysis, which I found difficult to read through. I think that presentation could be fixed there, and that would clear things up for anyone who reads the paper, theoretician or not.

There is a typo in Eqn. (2.2) -- need a T in the subscript on the LHS. Small nit.

**Questions:**

See my thoughts in the Weaknesses section -- I feel like this is more of a misunderstanding than anything.

---

> ### Author Response · Authors · 2025-11-22
> **Author response to Reviewer zW76**
>
> We thank the reviewer for their thoughtful comments. We would like to refer the reviewer to our overall Official Comment for some general remarks, as well as our updated manuscript, which contains multiple visual additions. In particular, we direct the reviewer to the boxed elements throughout, which contain the vital takeaways of our paper that may address some of the reviewer's overall comments and concerns, and newly added visualizations.
>
> Regarding the reviewer's specific concerns: we have fixed a few typos throughout the manuscript, thank you. As for the **control-theoretic terminology** in the manuscript, we fully understand the reviewer's concern regarding presentation. We believe that Simchowitz et al. (2025) demonstrates that the underlying reason why continuous-action imitation learning is fundamentally control-theoretic in nature. Thus it is unavoidable that any work which yields meaningful guarantees in continuous-action spaces requires significant control theoretic terminology. However, we understand that many of these concepts are not native to the machine-learning audience which this conference targets. We have taken steps in our revised manuscript to introduce control theoretic concepts more gently, including additional content in Section 4 and a control-theory primer in Appendix C, and enhanced the presentation with explanatory figures (Fig 4-7).
>
> Our work attempts to push a nascent front that demonstrates familiar discrete/online learning frameworks alone do not suffice to describe (or prescribe) empirically observed phenomena, and *neither do naive applications of ideas from traditional control*. We think our work is especially timely because up to this point *neither the control or machine learning thoeretics have yet realized the potential in deeply analyzing the fundamental connections between robot (i.e. physical!) learning practice and control-theoretic tools*. On the technical front, our overarching goal is to introduce:
> 1. A sufficiently general framework (**nonlinear** dynamics/policies, incremental stability for describing expert robustness) where phenomena in robot learning can be uniquely described, e.g. exponential compounding error.
> 2. A no-frills instantiation of said framework (**deterministic** dynamics/policies, **global, exponential** stability) to capture the essence of the problems while avoiding notational distractions from technical extensions that do not add new conceptual takeaways.
> 3. Surprisingly simple and clean prescriptions that both explain why certain components of existing practices work, but also contain crucial subtleties that are not captured by prior viewpoints.

---

### Official Review · Reviewer_Xggu · 2025-11-02

**Soundness:** 3
**Presentation:** 2
**Contribution:** 4
**Rating:** 6
**Confidence:** 3

**Summary:**

This paper gives a novel and rigorous analysis connecting two widely-used heuristics in continuous imitation learning, (1) action-chunking and (2) expert action noise-injection, to control-theoretic stability and compounding-error guarantees.
Under the notion of an open-loop exponential incremental input-to-state stability (EISS) assumption on the true dynamics,
the authors prove that sufficiently long action chunks yield horizon-free trajectory error bounds (Theorem 1).
When the true dynamics are not open-loop EISS, they advocate Practice 2: collect a mixture $P_{\pi^\star,\sigma_u,\alpha}=\alpha P_{\pi^\star}+(1-\alpha)P_{\pi^\star,\sigma_u}$ (noise injected at execution but clean action labels).
This paper also show that this mixture certifies first-order matching on the controllable subspace, producing a trajectory bound that removes the worst-case exponential blowup (Theorem 2).
Empirical validation on robomimic and MuJoCo tasks supports the theory and compares the offline noise-injection approach to baselines.

**Strengths:**

**1. A novel control-theoretic perspective on imitation learning.**

The paper reframes two widely used practical heuristics (1) **action-chunking** and (2) **expert action noise-injection,** in control-theoretic perspectives, centered on exponential incremental input-to-state stability (EISS).
By reducing compounding error to a stability property of the true dynamics $f$ and the policy–environment closed loop, the work gives precise conditions under which each heuristic is principled.
This perspective clarifies when a practitioner should prefer chunking (longer chunks $\ell$ when $f$ is open-loop EISS) versus noisy expert rollouts, and it interestingly connects IL practice to established stability concepts rather than treating the heuristics as ad-hoc.

**2. Mutually aligned theoretical and experimental results**

Theoretical predictions, (i) action-chunking yields horizon-free trajectory error under open-loop EISS, and (ii) noise-injection facilitates first-order matching in controllable subspaces when open-loop stability fails—are supported by empirical results on Robomimic and MuJoCo tasks.

**Weaknesses:**

**1. Narrow applicability of action-chunking under EISS assumption**

The primary theoretical guarantee for action-chunking (Theorem 1 / Propositions 3.1, 3.2) is stated under the strong assumption that the **true dynamics $f$ are open-loop EISS**.
This crucial scope condition is only briefly noted in the text, but the paper lacks a clear characterization of the method’s behavior when that assumption is violated.
To be more useful to practitioners and convincing to readers, the manuscript should clarify the conditions where chunking is recommended (e.g., global vs. local EISS), show how the bound degrades when stability weakens, and provide either a weakened theoretical statement for local/probabilistic EISS or empirical sweeps that identify where chunking helps versus hurts.

**2. Limited empirical scope and reproducibility concerns**

The experimental validation is limited to a small set of tasks, which makes it difficult to assess the generality of the claims. In addition, the authors have not released the code or full data-collection scripts, which prevents independent reproduction and detailed ablation.
The paper also omits results and practical guidelines for applying the two practices jointly; reporting combined-case experiments and decision rules would be valuable.

**3. Overbroad use of the term “data augmentation”**

The use of the generic term **data augmentation** in the title and abstract overstates the scope of the empirical/theoretical contribution: the manuscript studies a specific augmentation (additive action noise executed during expert rollouts with *clean* action labels) rather than arbitrary augmentation families (state perturbations, visual augmentations, domain randomization, etc.). This wording risks misleading readers and reviewers. To avoid overclaiming, replace or qualify “data augmentation” in the title/abstract/introduction with a precise phrase (e.g., “expert action noise-injection”, etc.), and if broader augmentation types are intended, either provide evidence for them or explicitly mark such extensions as future work.

**Minor comments**

- The paper’s presentation is a quite dense and often assumes control-theory familiarity that may not be shared by the broader RL/IL audience (who is more common ICLR reader); this reduces accessibility and reproducibility.
Additionally, Figures 1,2,3, whose experiments are discussed on page 9, are currently shown much earlier (pages 5 and 7) and therefore disrupt the narrative; these figures should be relocated.

- In Theorem A, $\mathcal{P}^\star_{\mathrm{stab}}$
and $\mathcal{P}^\star_{\mathrm{unst}}$ are defined in a circular way, which does not seem a proper definition.

**Questions:**

Q1. Could the authors recommend a practical criterion (estimable stability indicator, threshold) to decide when to apply action-chunking?

Q2. Could you justify the specific choice $\alpha=0.5$ used in Proposition 4.4 and Theorem 2?

Q3. Line 463 states “Practice 2 collects data in one shot, without ever observing learned policy rollouts.” By “one shot” do you mean a single interaction of many trajectories (not literally one trajectory)?

Q4. Beyond the offline protocol, Practice 2 could plausibly be applied interactively (i.e., inject noise into expert actions when the expert is queried during training, similar to DAgger-style interaction). Do the authors see this interactive variant as theoretically or practically different from the offline mixture?

---

> ### Author Response · Authors · 2025-11-22
> **Author response to Reviewer Xggu**
>
> We thank the reviewer for their thoughtful comments. We would like to refer the reviewer to our overall Official Comment for some general remarks, as well as our updated manuscript, which contains multiple visual additions. In particular, we direct the reviewer to the boxed elements throughout, which contain the vital takeaways of our paper that may address some of the reviewer's overall comments, and newly added visualizations (see e.g. Figures 4-7).
>
> To address some of the reviewer's specific comments:
>
> 1. **Applicability of action-chunking, EISS, and underlying assumptions.** Our paper addresses two different scenarios: (1) when the system dynamics are naturally EISS (informally *stabilizing*) in Section 3 and (2) when the system dynamics are not EISS in Section 4.
>
>    We analyze these separately for convenience but in the context of robotics these settings naturally correspond to aspects of the system state which (1) can be directly controlled via stabilizing low-level position based control (e.g. robot end effectors) or (2) cannot be directly be actuated (e.g. the manipulated objects) but where the expert can be expected to provide stability as a precondition of completing the task.
>
>    Our analysis proves that action-chunking guarantees stability of the learned policy for (1) while including noise-based exploration guarantees stability in the context of (2). Both of our analyses explicitly state their dependency on the EISS stability parameters $(C_{ISS}, \rho)$. In particular, as $\rho \to 1$, the system transitions from EISS to non-EISS.
>
>    Our results are tight in the sense that without either the interventions and assumptions in (1) or (2), per the hardness bounds in Simchowitz 2025, exponentially compounding error is unavoidable. In this regard, the EISS assumptions we make are essentially the minimal set where IL can be expected to work in any sense.
>
>    As such, **action-chunking requiring stability of the dynamics is in fact a primary takeaway of our results**: it is not a silver bullet, and sees use primarily in conjuction with **end-effector control** at a lower level, which enforces the necessary EISS on the open-loop dynamics from the learned policy's perspective. The relation between end-effector control and open-loop stable dynamics is remarked in Section 5, but we have added discussion following Definition 2.1 to clearly tie "open-loop stable dynamics" to the subset of robot learning applications that have seen the greatest uptake of action-chunking.
>
>    The only assumptions we make which could be relaxed are the assumptions on system regularity/smoothness; however, without these it is difficult to make any quantitative statements regarding behavior under error.
>
>    We have reworked the rebuttal draft of the paper to more explicitly highlight the above and the relation of our assumptions to practical applications in robotics.
>
> 2. **Evaluation Scope.** We understand the reviewers concerns regarding limited evaluation scope. We use the terminology of *Practice* to highlight that our analyzed interventions are currently common practice in large-scale foundation models for robotics [1-4]. In particular, we used the original Diffusion Policy codebase. As these interventions are already common, our paper focuses on a unified control-theoretic foundation for understanding how these practices work, with the experiments serving to better illustrate our results. Our robomimic experiments use a near vanilla CleanDiffuser codebase [5]. We will include the code for our mujoco experiments in the coming weeks.
>
> 3. **Use of "data augmentation."** We came to a similar conclusion as the reviewer, and have modified our terming to "exploratory data collection" in the current working draft of the paper as this is more precise and less overloaded.

---

> ### Author Response · Authors · 2025-11-22
> **Author response to Reviewer Xggu (cont.)**
>
> Regarding the reviewer's Questions:
>
>  1. The practical criterion we suggest for whether to use action chunking is whenever the policy's generated actions are fed into a lower-level stabilizing controller, e.g. a PID or end-effector control loop.
>
>     Our results intuitively link the stability of the lower-level controller to a minimum chunk length required for stable imitation learning. While one could estimate $(C_{ISS}, \rho)$ of the lower-level controller, and their logarithmic dependence in our bounds means these estimates can afford to be conservative, in practice it is more straightforward to treat the chunk length as a hyperparameter, with the understanding that changes in the lower-level control algorithm or sampling frequency will require adjustment of the chunk length.
>
> 2. We pick $\alpha = 0.5$ to reduce notational baggage. Any choice of $\alpha \in (0,1)$ suffices theoretically to derive guarantees, as long as enough noised trajectories are collected to satisfy the "burn-in"/sufficiency requirements, which is supported empirically in Figure 3 (center). In the revision, we will highlight the fact that our analysis reveals that the requisite proportion noise-injected trajectories can be **vanishing** in the large #traj regime, as noise-injection is only necessary for controlling a higher-order error term.
>
> 3. By "one-shot" we indeed mean a single round of data collection, consisting of multiple trajectories rather than one trajectory. This is opposed to DAgger or DART, which alternate between learning a policy and collecting more demonstrations. We inject a fixed noise-level, meaning all augmented trajectories can be collected in one sitting before any training is performed.
>
> 4. **Beyond offline protocol.** Noise-injection is not strictly offline, as it requires trajectory collection *after* specifying interaction parameters (e.g. noise level $\sigma_{\mathbf{u}}$ and ratio $\alpha$). Our analysis suggests that there is no benefit to using anything other than a fixed noise level for all noised trajectories, even given the opportunity for multiple rounds of data collection. An "online" version would thus essentially be DAgger-esque, where the optimal $\sigma_u$ is unknown and is estimated adaptively over multiple collection rounds. We believe this is an interesting route for further work.
>
> [1] Ke, L. et al. "Grasping with chopsticks: Combating covariate shift in model-free imitation learning for fine manipulation, " 2021.
>
> [2] Torne, Marcel, et al. "Learning Long-Context Diffusion Policies via Past-Token Prediction," 2025.
>
> [3] Høeg, S. H. et al. "Streaming diffusion policy: Fast policy synthesis with variable noise diffusion models," 2024.
>
> [4] Chi, C. et al "Diffusion policy: Visuomotor policy learning via action diffusion," 2023.
>
> [5] Dong, Z., et al. "Cleandiffuser: An easy-to-use modularized library for diffusion models in decision making," 2024.

---

> > ### Comment · Reviewer_Xggu · 2025-11-27
> > **Response to Rebuttal**
> >
> > Thank you for the response and the authors' efforts. I think most of my concerns have been addressed by the author's responses.
> >
> > The presentation has improved substantially, and the added clarifications make the paper easier to follow.
> > However, my concerns regarding limited reproducibility and applicability remain only partially resolved, and these aspects are central to the values of the ICLR community.
> >
> > Nonetheless, I continue to view the theoretical contribution, providing a principled justification for widely used IL heuristics and control-theoretic perspectives, as valid and academically valuable. I therefore keep my original rating.

---

### Author Response · Authors · 2025-11-22
**Overall Rebuttal-phase Revisions and Comments**

We thank the reviewers for their effort in reviewing our paper. We provide here general clarifications on the goals of our paper and mention some updates to the manuscript.

1. We have uploaded an **updated manuscript**, which contains multiple visual additions. This includes **boxed elements** that contain all the vital takeaways of our work, and **visualization figures** that illustrate core concepts and phenomena described in the paper (Figures 4-7). We hope this helps emphasize and clarify the crucial intuitions and messages in our work. We also added some additional technical background to Section 4, as well as a primer on control-theory in Appendix C.

2. We have moved the experiment details into the appendices in order to make space for additional discussion and more presentation regarding control theoretic preliminaries, which based on reviewer feedback seems a more effective use of constrained page budget. As such, we have kept the experiment figures (Fig 1-3) in place, as their messages and set-up are self-contained.

To reaffirm the **scope of paper**: our goal is foremost to provide the first solid mathematical grounding of disparate-but-powerful techniques in robot learning practice. This requires significant departures from existing paradigms (e.g. online learning, discrete IL/RL) in order to even **describe** unique pathologies that arise when applying learned models in dynamical systems. Rather, the key upshot of our fine-grained analyses is that they unlock novel interpretations of how these techniques **crucially enable stable imitation in continuous domains**. This has a few consquences. Most tangibly, our analyses exposes subtle but important algorithmic considerations, for example:

- **Short executed chunk lengths** suffice to stabilize in "open-loop stable" settings, matching prior evidence that chunk length is a robust hyperparameter (see e.g. Table 7 & 8 in Chi et al. 2023).
- **Collecting clean action labels** in noise-injection is critical for enjoying the benefits of local exploration, rather than noising policy the itself as RL-esque coverage suggests.
- **A mixture of clean and exploratory data** is optimal, where asymptotically the proportion of clean data dominates. This stands in stark contrast to how online/exploratory IL algorithms such as DART and DAgger are formulated, where all data collection beyond the initial round is exploratory.

More broadly, our results also strike up reconsiderations of prior "folk" understanding, for example:
- Though our analysis shows action-chunking is extremely powerful in "open-loop stable" settings, **it is not a silver bullet**. This draws a new considerations for when AC is applicable, both matching current practice (where AC is most successful in settings with **end-effector control**) and shaping expectations for unstable/dexterous settings.
- Our analysis of noise-injection reveals that the necessary "exploration" to stabilize IL in general settings is local in a precise geometric sense around expert trajectories, which can be achieved by a **fixed** exploratory signal. This stands in contrast to prior understanding that stable IL is achieved by online learning, where exploration is adaptive, e.g. expert labels on learned policy states (DAgger), or learned-policy-shaped noise (DART).

All the above points can also be found in the boxed elements in our revision. Finally, taking a longer-term view, we see our work as a stride toward establishing control theory as a mathematical **language** for understanding how **learning systems interact with the physical world**. This leads to our next point.

3. Regarding the technical **control-theory learning curve**, we fully understand the reviewers' concern, as a core motivation in the paper is precisely to introduce these concepts to an area that isn't typically described using this language. We have added additional discussion for the more technical objects studied in Section 4, and a primer to Appendix C.

On the other hand, as Reviewer Xggu points out, we make modeling assumptions that may not be as general as they could be, including regularity of the underlying dynamics. Our philosophy on the mathematical generality of our results is as follows:

1. Provide a sufficiently general framework (**nonlinear** dynamics/policies, incremental stability for describing expert robustness) where phenomena in robot learning can be uniquely described, e.g. exponential compounding error.
2. A straightforward instantiation of said framework (**deterministic** dynamics/policies, **global, exponential** stability) to capture the essence of the problems while avoiding baggage from extensions that do not add new conceptual takeaways.
3. Surprisingly simple prescriptions that both explain why certain components of existing practices work, but also contain crucial subtleties not captured by prior viewpoints.

**References**

Chi et al. "Diffusion policy: Visuomotor policy learning via action diffusion," 2023.

---

### Meta-Review · Area_Chair_tjfL · 2025-12-30

**Summary:**

The paper reframes two  practical heuristics, namely  (1) action-chunking and (2) expert action noise-injection, in control-theoretic perspectives. This paper proves in Theorem 1 that under the notion of an open-loop exponential incremental input-to-state stability (EISS) assumption on the system dynamics, using sufficiently long action chunks yields horizon-free trajectory error bounds. When the environment is less benign,  the authors demonstrate noise injection, i.e., adding noise while executing expert actions, is a simple and practical tool for avoiding compounding errors; it shows in Theorem 2 that the prosed mixture certifies first-order matching on the controllable subspace, producing a trajectory bound that removes the worst-case exponential.

Overall, I find the results are strong, and offers insight on the conditions when action chunking can be powerful, and also shed lights on possible solutions when these conditions do not hold.

**Reviewer Concerns:**

The reviewers raised two main concerns: 1) the presentation leaves it a bit difficult to understand; 2)  limited experimental scope and reproducibility.  In the revision, the authors re-organized the presentation a bit and  made effort to improve its readability  using the language of continuous-space dynamical systems.  The key takeways are now presented in boxed text, which makes it much easier to follow.

**Reviewer Scores:**

The review scores are 6/6/0

The reviewer giving 0 did not participate in the discussion.  This review contains no constructive criticism.

---

### Decision · Program_Chairs · 2026-01-26

Accept (Poster)